# DSH-Bench: A Difficulty- and Scenario-Aware Benchmark with Hierarchical Subject Taxonomy for Subject-Driven Text-to-Image Generation

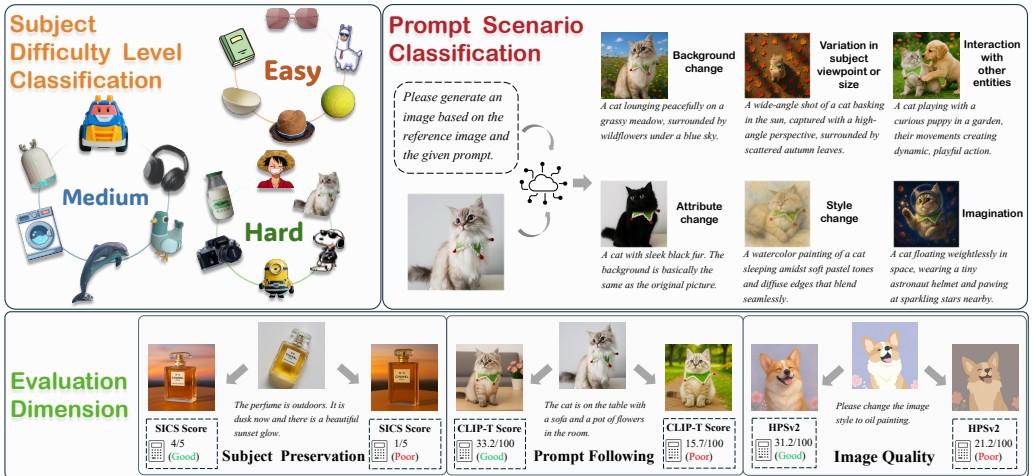

Figure 1: **Overview of DSH-Bench**. We curate a diverse dataset of subject images and categorize them into three difficulty levels—**easy**, **medium**, and **hard**—based on the complexity of preserving subject details. Leveraging GPT-4o's capabilities, we systematically generate contextually appropriate prompts for various scenarios. The generated images are then rigorously evaluated across three key dimensions: **Subject Preservation**, **Prompt Following**, and **Image Quality**.

## ABSTRACT

Significant progress has been achieved in subject-driven text-to-image (T2I) generation, which aims to synthesize new images depicting target subjects according to user instructions. However, evaluating these models remains a significant challenge. Existing benchmarks exhibit critical limitations: 1) insufficient diversity and comprehensiveness in subject images, and 2) inadequate granularity in assessing model performance across different subject difficulty levels and prompt scenarios. To address these limitations, we propose DSH-Bench, a comprehensive benchmark that enables systematic multi-perspective analysis of subject-driven T2I models through three principal innovations: 1) a hierarchical taxonomy sampling mechanism ensuring comprehensive subject representation across 58 fine-grained categories, 2) an innovative classification scheme categorizing both subject difficulty level and prompt scenario for granular model capability assessment, and 3) a novel Subject Identity Consistency Score (SICS) metric demonstrating 9.4% higher correlation with human evaluation compared to existing measures in quantifying subject preservation. Through empirical evaluation of 15 subject-driven T2I models, DSH-Bench uncovers previously obscured limitations in current approaches while establishing concrete directions for future research.

# 1 INTRODUCTION

Subject-driven text-to-image (T2I) generation aims to generate images conditioned on both textual prompts and specific reference images. It has become feasible due to significant advancements in large-scale T2I generative models (Ding et al., 2021; Gafni et al., 2022; Saharia et al., 2022; Rombach et al., 2022a; Balaji et al., 2022; Chang et al., 2023; Kang et al., 2023; Dong et al., 2024). In subject-driven T2I generation, aside from image quality considerations, two other fundamental criteria must be satisfied: Subject Preservation and Prompt Following. Subject Preservation requires that the generated image maintain the details of the reference subject. Prompt Following demands that the generated image consistently reflects the content in the prompt. For example, a user might request an image of "his dog traveling around the world". In this scenario, the generated image must depict a dog identical to the reference image while illustrating the act of traveling as described.

Significant progress has been made in subject-driven T2I generation in recent years (Ruiz et al., 2023; Gal et al., 2022; Kumari et al., 2023; Wang et al., 2024a; Li et al., 2023a; Ye et al., 2023; Gal et al., 2023b; Wei et al., 2023; Hu et al., 2024b; Qiu et al., 2023). One approach involves fine-tuning general T2I models to create specialized models that reproduce specific subjects present in the training datasets. Alternatively, encoder-based methods achieve subject preservation by adapting features to incorporate reference subject into a general T2I model. Despite these advancements, challenges remain in comprehensively and effectively evaluating the actual performance of these models. An effective evaluation method should not only provide a comprehensive and unbiased assessment, but also align with human perception to ensure reliable measurement. Furthermore, the evaluation method is expected to provide valuable insights for future research. However, current benchmarks (Ruiz et al., 2023; Kumari et al., 2023; Chen et al., 2023a; Wang et al., 2024b; Peng et al., 2025) are limited by insufficient diversity and comprehensiveness in subject image collection, which restricts the thoroughness of model evaluation. In addition, they do not facilitate a detailed understanding of subject difficulty and prompt scenarios, thus constraining the depth of insights obtainable from the evaluation. As shown in Figure 3, our analysis of numerous model-generated instances reveals that different subject images and prompts place varying demands on a model's ability. For example, although subject-driven T2I models are capable of effectively preserving the details of relatively simple objects (e.g., a tennis ball), they often struggle to accurately reproduce objects with more intricate features (e.g., a camera). This observation highlights the importance of categorizing the subject difficulty and prompt scenario to better assess model performance. To address these requirements, we introduce DSH-Bench, a novel benchmark offers three notable advantages:

1. ***The diversity of subject images in DSH-Bench is substantially greater*** To mitigate evaluation bias caused by low diversity of subject images, we employ a hierarchical taxonomy in image collection. We referenced COCO (Lin et al., 2014) and ImageNet (Deng et al., 2009) in the hierarchical taxonomy construction. As shown in Figure 4(a), the widely used DreamBench includes only 6 categories and 30 subjects. In contrast, our benchmark expands the dataset to 48 categories and 459 subjects—representing an increase of **8**× and **15**×, respectively. Although DreamBench++ (Peng et al., 2025) offers 150 subjects, its diversity is constrained by its image collection. Notably, **33%** of our categories are not represented in DreamBench++. Therefore, benefiting from DSH-Bench's greater subject diversity, we enable more comprehensive evaluation of models.

2. ***An innovative classification scheme for subject difficulty level and prompt scenario*** Figure 3 shows the model's performance varies significantly with different samples, highlighting the necessity for a classification of both subject image and prompt. Although DreamBench++ categorizes prompts based on their perceived difficulty, the criteria underlying this classification are not clearly defined. Additionally, DreamBench++ does not analyze the difficulty levels associated with different subjects. To address these limitations, we categorize subjects into three difficulty levels (easy, medium, and hard) according to the difficulty of preserving visual appearance and classify prompts into six scenarios (background change, variation in subject viewpoint or size, interaction with other entities, attribute change, style change, imagination). As a result, our approach enables a more comprehensive and granular analysis of the challenges faced by current models.

3. ***A human-aligned and more efficient metric for subject preservation*** DreamBench++ replaces CLIP (Radford et al., 2021) and DINO (Caron et al., 2021) with GPT-4o (OpenAI, 2024) for evaluation, resulting in improved alignment with human evaluation. However, our benchmark reveals that per-model evaluation under this paradigm requires approximately 20,000 API calls to

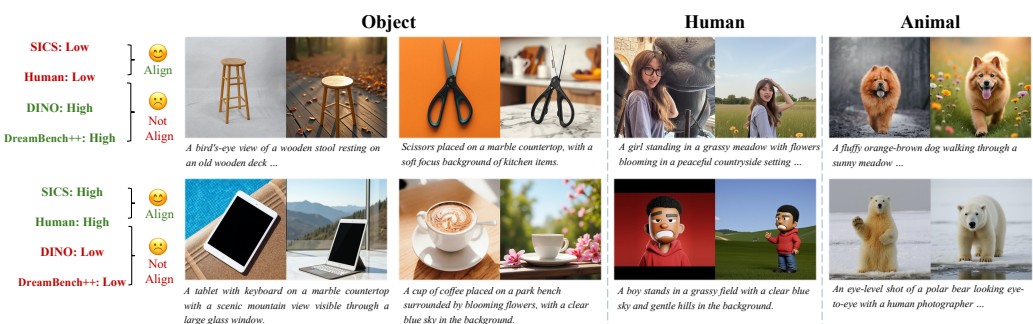

Figure 2: Qualitative comparison of subject preservation between SICS and the other methods.

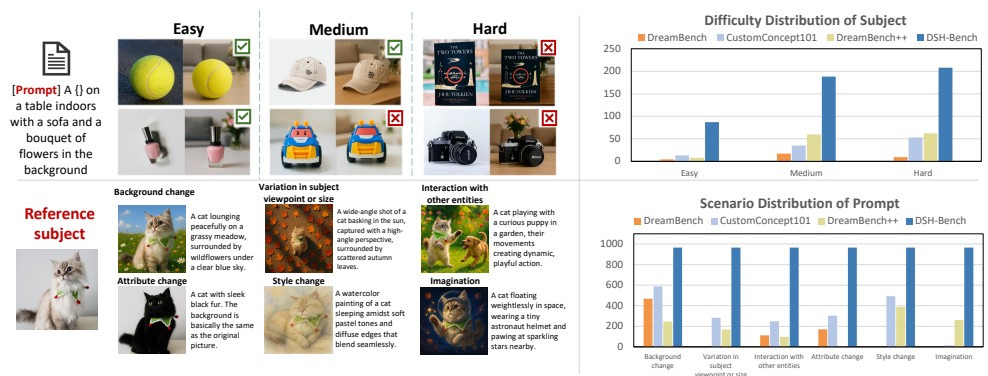

Figure 3: Qualitative comparison of generated images under different difficulty levels and scenarios.

GPT-4o, incurring prohibitive computational costs exceeding $400 for each evaluation. To address the limitation, we introduce **Subject Identity Consistency Score** (SICS), which innovatively focuses on subject-level consistency rather than merely relying on embedding comparisons. Firstly, five annotators label a training dataset containing 5,000 image-text pairs, focusing on subject preservation evaluation. We then fine-tune Qwen2.5-VL-7B (Bai et al., 2025) on this dataset, which leads the model to focus on core visual attributes rather than high-level semantics. Finally, we use Kendall's $\tau$ value to quantify the alignment between model outputs and human evaluation. Experimental results demonstrate that SICS achieves a statistically significant improvement, outperforming Dreambench++ by 9.4% in human evaluation correlation metrics. Figure 2 presents a partial qualitative comparison of concept preservation between SICS and the other assessment methods.

**Takeaways** We present some insightful findings from evaluating fifteen methods: i) Our evaluation reveals that no single method demonstrates consistently robust performance across all categories. Therefore, implementing hierarchical taxonomy sampling of subject images is critical for mitigating potential evaluation biases. ii) All methods exhibit degraded performance on hard subject images. It is crucial to enhance models' ability to encode and reconstruct complex subject details more effectively in future research. iii) The subject-driven T2I model's capability for different prompt scenarios is not robust. Future research on subject-driven T2I generation should focus on optimizing for adaptation to a variety of prompt scenarios.

In summary, our contributions are as follows: 1) We employ a hierarchical taxonomy in image collection to ensure both the diversity and comprehensiveness of subject images. 2) We propose an innovative classification scheme to categorize subject difficulty levels and prompt scenarios. This scheme enables us to obtain valuable insights. 3) We propose a human-aligned metric to evaluate subject preservation, which offers greater efficiency compared to DreamBench++. We are open-sourcing DSH-Bench, including subject images, prompts, generated images and related code.

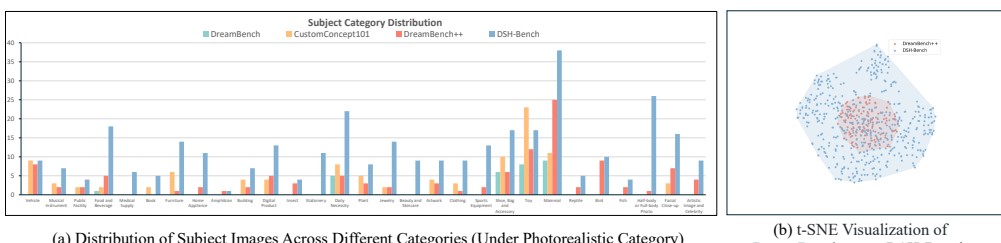

(a) Distribution of Subject Images Across Different Categories (Under Photorealistic Category)

(b) t-SNE Visualization of DreamBench++ vs. DSH-Bench

Figure 4: **Distribution of subject images.** (a) Category-wise image distribution for our benchmark versus prior benchmarks. (b) t-SNE comparison of images between DSH-Bench and DreamBench++.

## 2 DSH-BENCH

This section provides an overview of the primary components of DSH-Bench. Section 2.1 outlines the data construction process. Section 2.2 introduces the definitions and evaluation methods for three evaluation dimensions. *A detailed explanation is available in the supplementary materials.*

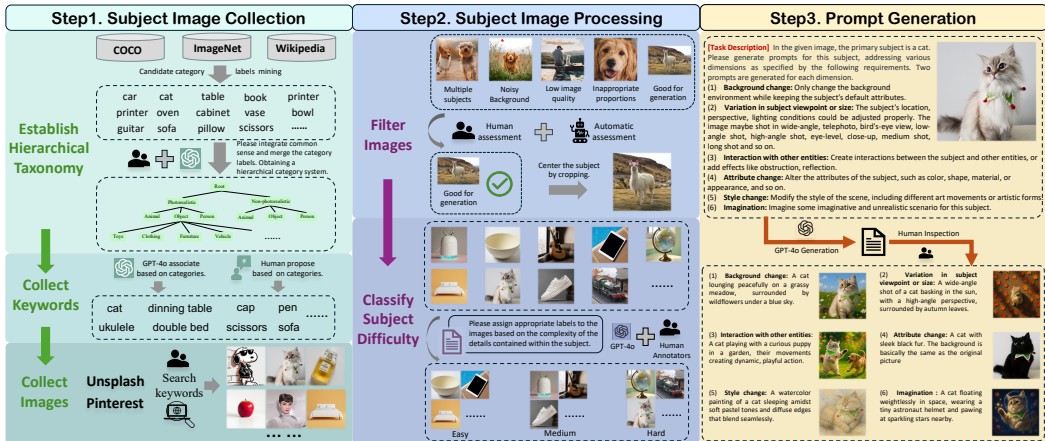

Figure 5: **Dataset construction process of DSH-Bench.** We construct a hierarchical taxonomy to obtain a comprehensive set of keywords. Then we collect web images using these keywords. After performing both manual review and automated filtering of the images, we classify the difficulty of subject images and use GPT-4o to generate prompts for each subject image.

### 2.1 BENCHMARK DATASET CONSTRUCTION

#### 2.1.1 SUBJECT IMAGE COLLECTION

**Hierarchical Taxonomy Establishment** As shown in Figure 5, we establish a hierarchical taxonomy. For the first- and second-level categories, we primarily refer to existing benchmarks from prior studies (Ruiz et al., 2023; Kumari et al., 2023; Peng et al., 2025), resulting in two first-level categories and six second-level categories. For the third-level categories, we first reference COCO and ImageNet to compile a list of candidate category labels, then utilize GPT-4o to consolidate them into 58 refined categories. The final hierarchical taxonomy is confirmed and refined through co-authors' discussion. The detailed process and the category contents are provided in Appendix A.

**Keyword Collection & Internet Image Collection** In DreamBench++, keywords collection relies on GPT-4o and human input. The approach does not adequately ensure the diversity of the obtained keywords, potentially introducing bias during the image collection process. In contrast, DSH-Bench derives keywords from a hierarchical taxonomy. For each third-level category, we use GPT-4o to generate associated keywords, which are further supplemented by humans. All keywords are

then consolidated and deduplicated, resulting in a final set of **400** unique keywords—surpassing DreamBench++'s 300. The specific keywords are provided in the Appendix B. Given a set of selected keywords, we retrieve images from Unsplash (uns) and Pinterest (pin). Keywords without suitable images are discarded. *Each image's copyright status has been verified for academic suitability.*

### 2.1.2 SUBJECT IMAGE PROCESSING

**Image Filtering** To filter unsuitable images, human annotators remove images with multiple subjects and noisy backgrounds. We use aesthetic score (Xu et al., 2024) and SAM (Kirillov et al., 2023) to filter images with low image quality and inappropriate proportions of subject regions. The curated images are subsequently cropped to centralize the reference subject.

**Subject Difficulty Level Classification** As illustrated in Figure 3, the model's performance varies considerably across different samples. To derive meaningful insights, we classify the subject images according to the difficulty level that the model experiences in preserving details of the reference subject. We define three subject difficulty levels, including (1) **Easy:** Subjects characterized by minimal surface complexity and homogeneous textural properties, exemplified by smooth-surfaced objects such as a ceramic mug with uniform coloration. These cases present negligible challenges for detail preservation due to their structural regularity. (2) **Medium:** Subjects containing discernible high-frequency features while maintaining global structural coherence, such as cylindrical containers with legible typographic elements. These cases require intermediate detail preservation capabilities. (3) **Hard:** Subjects exhibiting non-uniform texture distributions and multi-scale geometric details, typified by objects like book covers containing fine-grained calligraphic elements. Such cases expose model limitations in maintaining structural fidelity and textural granularity under complex topological constraints. We utilize GPT-4o to classify the subject images according to the aforementioned criteria. Subsequently, all images are reviewed by five human annotators to ensure accuracy and consistency.

### 2.1.3 PROMPT GENERATION

Although DreamBench++ categorizes prompts based on their perceived difficulty, it does not provide empirical evidence to substantiate the criterion. To address this limitation, we organize the prompts according to specific application scenarios, dividing them into six categories, including (1) **Background change (BC):** scenarios involving changes in background elements. (2) **Variation in subject viewpoint or size (VS):** scenarios that entail changes in camera angle, which may include variations in subject size, lighting, or shadows. (3) **Interaction with other entities (IE):** scenarios requiring complex interactions with additional entities, potentially resulting in occlusion and necessitating adherence to physical plausibility. (4) **Attribute change (AC):** scenarios involving modifications to certain attributes of the subject, such as color or shape. (5) **Style change (SC):** scenarios involving alterations in the artistic or visual style of the subject. (6) **Imagination (IM):** scenarios where the target image depicts an imagined or fictional scene. We generate two prompts for each scenario. The specific instructions are depicted in Figure 5. All prompts are reviewed by five human annotators to ensure they are ethical and free from defects. For the specific verification procedure, please refer to the Appendix E.3. Finally, we obtain a total of **459** high-quality images and **5,508** prompts. Figure 3 shows the distribution of subject image difficulty levels and prompt scenarios. We visualize the t-SNE of images from our benchmark and DreamBench++ in Figure 4(b). The results indicate that our benchmark achieves superior diversity.

## 2.2 EVALUATION DIMENSION

Previous notable works (Ruiz et al., 2023; Gal et al., 2022; Kumari et al., 2023; Wang et al., 2024a) evaluate the performance of subject-driven T2I models from two perspectives: Subject Preservation and Prompt Following. Mao et al. (2024) also uses ImageReward (Xu et al., 2023) to evaluate image quality. Therefore, DSH-Bench evaluates from the three aforementioned dimensions.

**Subject Preservation** DreamBench++ utilizes GPT-4o for evaluation to improve alignment with human assessments. However, the GPT-4o-based method is prohibitively expensive. To address this limitation, we propose a novel metric—**Subject Identity Consistency Score** (SICS). Firstly, we establish a scoring criterion for assessing subject preservation, the details are provided in Appendix E.2. Five annotators label the collected image pairs according to the criterion. During the annotation process, each image pair is not only assigned a score but also accompanied by an explanation. Previous

work (Wei et al., 2022) has indicated that labeled data with explanatory reasoning can help models better understand the underlying logic and reasoning behind the labels. We then perform meticulous fine-tuning of the model using this annotated dataset. During fine-tuning, SICS leverages prompts to explicitly prioritize subject consistency rather than global semantics, mitigating background and style artifacts that commonly bias CLIP-based approaches and yielding closer alignment with the goals of subject-consistency evaluation. Although GPT-4o demonstrates outstanding performance across a wide range of tasks, it has not been specifically optimized for subject preservation evaluation. More details of the SICS metric can be found in Appendix E.2.

**Prompt Following** Prompt following primarily evaluates whether a model can generate images that accurately correspond to textual prompts. DreamBench++ has demonstrated that the CLIP-T score is highly consistent with human annotations. Therefore, we also adopt CLIP-T score as the evaluation metric for prompt following.

**Image Quality** HPSv2 (Wu et al., 2023) utilizes professionally annotated data to more accurately reflect human aesthetic preferences for generated images. Previous studies (Sun et al., 2025) demonstrate that models optimized with HPSv2 achieve superior performance in image quality assessment compared to existing approaches. Therefore, we adopt HPSv2 for image quality evaluation.

## 3 EXPERIMENT

### 3.1 EXPERIMENT SETUP

**Implementation Details** We conduct experiments on two mainstream approaches: *i) Finetuning-based:* 1) Textual Inversion(TI) (Gal et al., 2023a), 2) DreamBooth, 3) Custom Diffusion, 4) Hiper (Han et al., 2023), 5) NeTI (Alaluf et al., 2023). *ii) Encoder-based:* 1) BLIP-Diffusion (Li et al., 2023a), 2) IP-Adapter (Ye et al., 2023), 3) MS-Diffusion (Wang et al., 2024b), 4) Emu2 (Sun et al., 2024), 5) OminiControl (Tan et al., 2024), 6) SSR-Encoder (Zhang et al., 2024), 7) RealCustom++ (Mao et al., 2024), 8) OmniGen (Xiao et al., 2024), 9) $\lambda$-Eclipse (Patel et al., 2024), 10) UNO (Wu et al., 2025). Our experiments are conducted using the official implementations to guarantee reliability and fairness. More details can be found in Appendix E.

**Human Annotation** All annotation tasks, including labeling of the SICS training datasets, were conducted by the same five human annotators. We provide the annotators with detailed labeling guidelines and sufficient training to ensure they fully understand the subject-driven T2I generation task and could provide unbiased and discriminative scores. For additional details regarding the human annotation process, please see the Appendix E.4.

Table 1: The human alignment degree among different metrics, measured by **Kendall's $\tau$ value** and **Spearman correlation coefficient value**. H: Human, G: GPT-4o, D: DINO, Dv2: DINOv2, CB: CLIP-B, CL: CLIP-L, S: SICS. Bold font is used to denote the maximum value in a row.

| Method | Kendall↑ | | | | | | Spearman↑ | | | | | |
|---|---|---|---|---|---|---|---|---|---|---|---|---|
| | H-CB | H-CL | H-D | H-Dv2 | H-G | H-S | H-CB | H-CL | H-D | H-Dv2 | H-G | H-S |
| BLIP-Diffusion | 0.228 | 0.176 | 0.285 | 0.167 | 0.354 | **0.531** | 0.285 | 0.215 | 0.350 | 0.206 | 0.383 | **0.554** |
| IP-Adapter | 0.294 | 0.296 | 0.258 | 0.290 | 0.419 | **0.622** | 0.364 | 0.371 | 0.325 | 0.364 | 0.459 | **0.657** |
| MS-Diffusion | 0.158 | 0.090 | 0.116 | 0.122 | 0.119 | **0.178** | 0.194 | 0.109 | 0.144 | 0.156 | 0.131 | **0.189** |
| OminiControl | 0.375 | 0.371 | 0.337 | 0.348 | 0.650 | **0.713** | 0.490 | 0.486 | 0.441 | 0.453 | 0.729 | **0.764** |
| SSR-Encoder | 0.264 | 0.338 | 0.295 | 0.348 | 0.504 | **0.664** | 0.328 | 0.421 | 0.368 | 0.434 | 0.549 | **0.697** |
| UNO | 0.249 | 0.218 | 0.299 | 0.240 | 0.236 | **0.385** | 0.340 | 0.297 | 0.390 | 0.312 | 0.268 | **0.426** |
| RealCustom++ | 0.181 | 0.128 | 0.206 | 0.241 | 0.291 | **0.464** | 0.229 | 0.162 | 0.266 | 0.303 | 0.325 | **0.511** |
| OmniGen | 0.465 | 0.396 | 0.344 | 0.349 | 0.617 | **0.621** | 0.579 | 0.497 | 0.440 | 0.456 | 0.697 | **0.667** |
| $\lambda$-Eclipse | 0.143 | 0.233 | 0.084 | 0.103 | 0.325 | **0.375** | 0.176 | 0.287 | 0.103 | 0.127 | 0.352 | **0.393** |
| Custom Diffusion | 0.316 | 0.336 | 0.382 | 0.425 | 0.487 | **0.642** | 0.388 | 0.409 | 0.470 | 0.519 | 0.512 | **0.654** |
| DreamBooth | 0.639 | 0.591 | 0.537 | 0.429 | 0.647 | **0.692** | 0.733 | 0.721 | 0.661 | 0.537 | 0.705 | **0.740** |
| Textual Inversion | 0.482 | 0.459 | 0.447 | 0.438 | 0.541 | **0.568** | 0.587 | 0.559 | 0.545 | 0.534 | 0.582 | **0.590** |
| HiPer | 0.338 | 0.387 | 0.351 | 0.404 | 0.584 | **0.625** | 0.417 | 0.469 | 0.430 | 0.496 | 0.629 | **0.655** |
| NeTI | 0.469 | 0.456 | 0.431 | 0.417 | 0.617 | **0.728** | 0.573 | 0.561 | 0.529 | 0.512 | 0.682 | **0.778** |
| ALL | 0.416 | 0.411 | 0.350 | 0.376 | 0.619 | **0.677** | 0.529 | 0.522 | 0.451 | 0.483 | 0.697 | **0.734** |

### 3.2 MAIN RESULTS

**SICS Results** Table 1 presents a rigorous study of human alignment using *Kendall's $\tau$ value* (KDV) and *Spearman correlation coefficient value* (SCV) (metric selection rationale in Appendix E.2). Our experimental results demonstrate that **SICS achieves superior alignment with human evaluations**

Table 2: **Evaluation of Subject-driven T2I generation.** DB: DreamBench, DB++: DreamBench++, HB: DSH-Bench. All scores are normalized to 0-1. Bold indicates the minimum value in each row for a given evaluation dimension..

| Method | Subject Preservation | | | Prompt Following | | | Image Quality | | |
|--------|------|------|------|------|------|------|------|------|------|
| | DB | DB++ | HB | DB | DB++ | HB | DB | DB++ | HB |
| BLIP-Diffusion | 0.229 | 0.216 | **0.204** | 0.291 | 0.278 | **0.277** | 0.267 | 0.254 | **0.223** |
| IP-Adapter | 0.230 | 0.244 | **0.229** | 0.321 | 0.318 | **0.315** | 0.291 | 0.296 | **0.266** |
| MS-Diffusion | **0.316** | 0.346 | 0.352 | **0.332** | 0.339 | 0.338 | 0.311 | 0.314 | **0.294** |
| OminiControl | 0.279 | 0.268 | **0.258** | 0.325 | 0.337 | 0.334 | 0.312 | 0.308 | **0.290** |
| SSR-Encoder | 0.231 | **0.202** | **0.202** | 0.290 | **0.287** | 0.295 | 0.273 | 0.270 | **0.247** |
| UNO | **0.409** | 0.410 | **0.409** | 0.317 | 0.322 | 0.323 | 0.304 | 0.297 | **0.278** |
| Emu2 | 0.360 | 0.343 | **0.341** | **0.291** | 0.309 | 0.304 | 0.272 | 0.278 | **0.260** |
| RealCustom++ | 0.377 | 0.380 | **0.375** | **0.325** | 0.329 | 0.332 | 0.316 | 0.314 | **0.298** |

Table 3: **DSH-Bench leaderboard.** The models are ranked by the final score $S_h$. We only present the top models; the complete ranking can be found in the Appendix D.2.

| Method | T2I Model | Subject Preservation | Prompt Following | Image Quality | $S_h\uparrow$ |
|--------|-----------|----------|----------|----------|------|
| UNO | FLUX.1-dev | **0.409** | 0.323 | 0.278 | **0.252** |
| RealCustom++ | SDXL | 0.375 | 0.332 | **0.294** | 0.251 |
| MS-Diffusion | SDXL | 0.352 | **0.338** | 0.294 | 0.248 |
| Emu2 | SDXL | 0.341 | 0.304 | 0.260 | 0.228 |
| OminiControl | FLUX.1-schnell | 0.258 | 0.334 | 0.290 | 0.218 |
| IP-Adapter | SDXL | 0.256 | 0.292 | 0.266 | 0.199 |
| $\lambda$-Eclipse | SDXL | 0.229 | 0.315 | 0.242 | 0.198 |
| OmniGen | SD v1.5 | 0.202 | 0.295 | 0.265 | 0.183 |
| SSR-Encoder | SDXL | 0.188 | 0.322 | 0.247 | 0.181 |
| NeTI | SD v1.4 | 0.192 | 0.301 | 0.234 | 0.176 |
| BLIP-Diffusion | SD v1.5 | 0.204 | 0.277 | 0.223 | 0.174 |
| DreamBooth | SD v1.5 | 0.158 | 0.321 | 0.245 | 0.164 |
| HiPer | SD v1.4 | 0.135 | 0.318 | 0.247 | 0.151 |
| Textual Inversion | SD v1.5 | 0.109 | 0.299 | 0.225 | 0.129 |
| Custom Diffusion | SD v1.4 | 0.062 | 0.323 | 0.240 | 0.091 |

**compared to existing methods**, showing consistently higher agreement across both correlation metrics in most experimental settings. Although SICS attains second-highest correlation scores in MS-Diffusion and OmniGen, it significantly outperforms GPT-4o (*GPT-4o refers to the evaluation method used in DreamBench++*) by **9.37%** (KDV) and **5.31%** (SCV). This performance gap strongly suggests SICS's enhanced capability in modeling human evaluation. Notably, GPT-4o exhibits greater consistency with human evaluation than CLIP and DINO, aligning with DreamBench++ findings. Importantly, our proposed SICS metric surpasses all existing metrics in human judgment consistency.

**Quantitative & Qualitative Results**    Table 2 shows overall evaluation results. The results show that: **i) DSH-Bench poses more significant challenges than existing benchmarks.** For subject preservation and image quality, the majority of methods consistently yield lower scores on DSH-Bench. The result can be attributed to the hierarchical taxonomy sampling method employed, which allows our dataset to more accurately represent the true data distribution. Moreover, it highlights that benchmarks derived from true distributions present greater challenges. **ii)** For prompt following, DreamBench yields slightly lower scores than DSH-Bench for certain methods. In DreamBench, prompts requiring attribute change constitute 22.7%, which is higher than the 16.7% observed in DSH-Bench. Figure 7(b) indicates that all methods exhibit relatively poor average performance on prompts involving attribute change. **iii)** Table 3 shows that there exists a trade-off between subject preservation and prompt following. We plot the Pareto frontier (see in Appendix D.1) using the data presented in Table 3. The primary objective is to identify a Pareto optimal solution that effectively balances the two objectives. *Additional results and discussions can be found in Appendix D.2*.

**Leaderboard**    In order to assess a model's overall capability, we define the final score as:

$$S_h = \frac{3}{\frac{\lambda}{\text{SP}} + \frac{\gamma}{\text{PF}} + \frac{\mu}{\text{IQ}}} \tag{1}$$

SP, PF, and IQ represent the scores for Subject Preservation, Prompt Following, and Image Quality, respectively. $\lambda, \gamma, \mu$ are the weights assigned to the importance of each corresponding dimension. In this study, we set $\lambda = 1.5, \gamma = 1.5, \mu = 1$, as subject preservation and prompt following are of paramount importance in subject-driven T2I generation. The harmonic mean requires strong

performance across all dimensions to yield a high overall score. We rank models by $S_h$ scores in Table 3. UNO exhibits relatively strong overall performance. We attribute this improvement to the novel architectural design of UNO and the minimal yet effective modifications implemented in DiT.

## 4 ANALYSIS

In this section, we conduct a detailed analysis of the performance of all methods based on the hierarchical category, the subject difficulty level classification, and the prompt scenario classification:

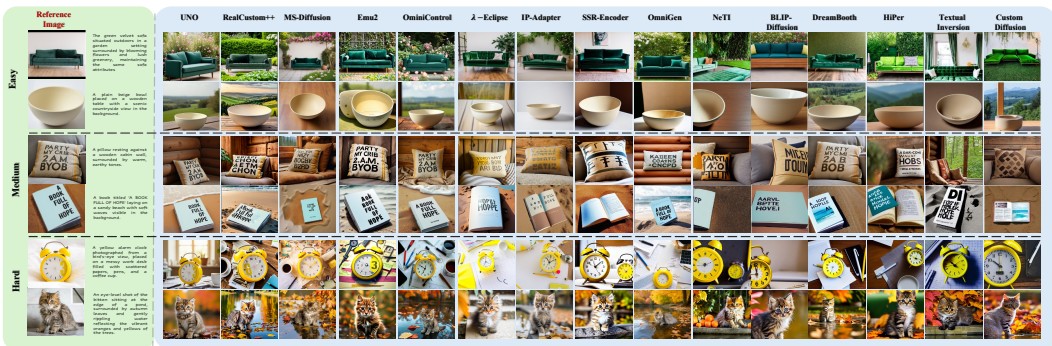

Figure 6: Examples generated by methods listed in the leaderboard. Best viewed when zoomed in.

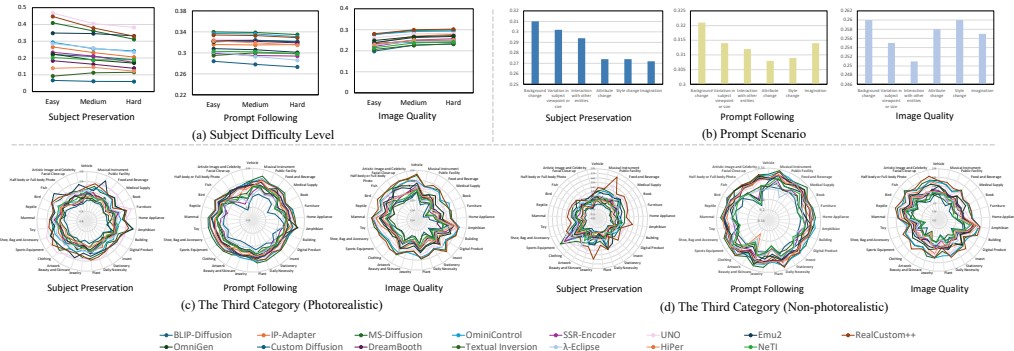

Figure 7: Comparison for DSH-Bench scores in different evaluation dimensions. The specific metric values are provided in the Appendix D.2. Best viewed when zoomed in.

**A scientific and comprehensive subject image sampling method is necessary** Figure 7(c) and Figure 7(d) present the performance of various methods in the third-level categories. The results reveal that model robustness varies considerably among categories. For example, performance in categories "*artwork*" (both photorealistic and non-photorealistic) is substantially lower. This disparity suggests that the absence of subject images from specific categories can lead to biased evaluation results, highlighting the importance of data diversity. Furthermore, Figure 7 also demonstrates that none of the current models perform well across all categories. We hypothesize that this may be related to the varying complexity of the subjects within different categories. A more detailed analysis of model performance in different categories can be found in Appendix D.1.

**Current subject-driven T2I models exhibit performance degradation on hard level subjects** As illustrated in Figure 7(a), the model exhibits substantial variation in performance across different difficulty levels: 1) For subject preservation, there is a pronounced decline in performance as the difficulty of the subject images increases. The model achieves significantly better results on images classified as simple compared to those categorized as hard. This observation supports the validity of our image difficulty classification scheme. 2) For prompt following, Figure 7(a) shows that the capability of the models is minimally influenced by the subject difficulty level. This could be explained by the fact that CLIP-T primarily emphasizes overall semantic information. Consequently,

as long as the generated image correctly represents the general category and overall shape, the evaluation score is unlikely to be substantially reduced, even if finer details are not perfectly captured. *Given these findings, it is crucial to enhance models' ability to encode and reconstruct complex subject details more effectively in future research endeavors.*

**The subject-driven T2I capability for different prompt scenarios is not robust** Figure 7(b) shows the average performance of all models across six prompt scenarios. The results show that: 1) In BC, VS, and IE scenarios, the model's performance consistently declines across all evaluation dimensions. This trend suggests that the difficulty of the scenarios increases progressively from BC to IE. Notably, the finding that the IE scenario is more challenging than the BC scenario aligns with intuitive expectations. 2) For subject preservation, the model's average performance across the AC, SC, and IM prompt scenarios remains relatively low. This could be because the generated subjects undergo partial modifications relative to the original subjects in these three scenarios. *Given these findings, more emphasis should be placed on enhancing methods for IE prompt scenario. For instance, increasing the volume of training data tailored to these specific contexts.*

## 5 RELATED WORK

### 5.1 SUBJECT-DRIVEN TEXT-TO-IMAGE GENERATION

In recent years, subject-driven T2I generation has attracted significant research attention (Ruiz et al., 2023; Gal et al., 2022; Kumari et al., 2023; Wang et al., 2024a; Li et al., 2023a; Gal et al., 2023b;a; Wei et al., 2023; Hu et al., 2024b; Qiu et al., 2023). Within the context of diffusion models, optimization-based model (Voynov et al., 2023; Liu et al., 2023; Hua et al., 2023; Hao et al., 2023) enables subject-driven generation by introducing lightweight parameters and performs parameter-efficient fine-tuning for each subject. In contrast, the encoder-based methods (Shi et al., 2023; Ma et al., 2024; Chen et al., 2023b; Li et al., 2023b; Le et al., 2024; Rowles et al., 2024; Zeng et al., 2024; Hu et al., 2024a; Huang et al., 2025a; Xiong et al., 2025; Patashnik et al., 2025; Wu et al., 2025; Huang et al., 2025b; He et al., 2025) leverage additional image encoders and network layers to encode the reference image of the subject. IP-Adapter (Ye et al., 2023) introduces cross-attention through an additional image encoder to incorporate control signals. Furthermore, SSR-Encoder (Zhang et al., 2024) enhances identity preservation without necessitating further fine-tuning when introducing new concepts. The Diffusion Transformers (Peebles & Xie, 2023; Podell et al., 2023; Rombach et al., 2022b) uses transformer as a denoising network to iteratively refine noisy image tokens. Based on these foundation models, approaches like OminiControl (Tan et al., 2024) and UNO (Wu et al., 2025) explore the inherent image reference capabilities of transformers.

### 5.2 SUBJECT-DRIVEN T2I GENERATION BENCHMARK

Evaluation of subject-driven T2I relies on diverse metrics. For image quality, several notable studies (Xu et al., 2023; Kirstain et al., 2023; Wu et al., 2023; Alaluf et al., 2023; Xu et al., 2024; Wang et al., 2025) have been proposed. Subject preservation is typically measured by learning-based metrics that compare deep feature distances, often using embeddings from large vision models such as CLIP (Radford et al., 2021) and DINO (Caron et al., 2021), as well as image-retrieval scores (Liu et al., 2021). To better align with human perception, DreamSim (Fu et al., 2023) emphasizes foreground objects when assessing image similarity. Semantic consistency is commonly measured with the CLIP score. DreamBench lacks diversity in subjects and prompts, and although DreamBench++ expands to 150 subjects, it still lacks systematic categorization, hindering meaningful analysis.

## 6 CONCLUSION

This paper introduces a novel benchmark called DSH-Bench, designed specifically for subject-driven T2I generation. Key features include: 1) a hierarchical category system in image collection to ensure both the diversity and comprehensiveness of subject images; 2) an innovative classification scheme for categorizing subject difficulty levels and prompt scenarios to obtain valuable insights; and 3) a human-aligned and more efficient metric for subject preservation. The benchmark will be publicly available to support the advancement in the subject-driven T2I generation era.

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

# A  DETAILS OF HIERARCHICAL CATEGORY ESTABLISHING

**The First-level Category**   We observed the composition of existing benchmark data. From a more abstract and higher-level perspective perspective, images in these datasets could be categorized into two types: photorealistic and non-photorealistic. Theoretically, the specific image categories represented within these two types can be identical. To maintain consistency with previous work and to ensure comprehensive data sampling, we designated photorealistic and non-photorealistic as the first-level categories. Furthermore, we ensure that the specific subcategories under both photorealistic and non-photorealistic types are fully aligned.

**The Second-level Category**   We examined both the DreamBench and DreamBench++ datasets. In DreamBench, the dataset is divided into two categories: living subjects and objects. DreamBench++ further refines this categorization by introducing three categories: living subjects, objects, and style. We construct our secondary subcategories based on them. We define our secondary categories as objects, humans, and animals. Specifically, we subdivide the "living subjects" category into "humans" and "animals," as humans exhibit significantly different visual characteristics compared to animals. For the human category, we place particular emphasis on the accuracy of facial feature reconstruction, acknowledging the existence of dedicated research domains focused on facial preservation. In contrast, animals generally display greater variability in appearance than human faces. In comparison to DreamBench++, we exclude the "style" category. This decision is motivated by the focus of our task on subject-driven T2I generation, where "style" does not constitute a tangible entity. Moreover, including the style category would complicate the calculation of subject consistency, whereas our work is primarily concerned with the customization of entities.

**The Third-level Category**   For the third-level categories, our objective was to strike a balance between granularity and generality. Categories that are too broad may result in insufficient keyword retrieval, potentially introducing bias into the final image sampling. Conversely, overly fine-grained categories may hinder subsequent experimental analysis by diluting meaningful insights. To address this, we consulted existing large-scale datasets such as COCO and ImageNet, as well as Wikipedia, to compile a list of candidate category labels. The specific labels are listed in Table 4. This comprehensive set of labels ensured broad coverage. However, many of these labels were excessively detailed, so we employ GPT-4o to merge them, followed by manual review to ensure the rationality and coherence of the final categories. The correspondence between the third-level categories and the candidate category labels is presented in Table 4. For the "human" category, we introduced a specific distinction by dividing it into "celebrities & artistic figures," "facial close-ups," and "half-body or full-body photo". We observed that models tend to perform significantly better on celebrities, which we hypothesize is due to the inclusion of celebrity data in the training sets of text-to-image foundation models. Table 14 provides empirical support for our hypothesis to some extent. The rationale for distinguishing between facial close-ups and non-facial close-ups is that the former focuses exclusively on the facial details of the individual in the reference image, whereas the latter also requires attention to the body details.

Through the aforementioned steps, we constructed a hierarchical category system. The resulting category hierarchy is presented in Figure 8.

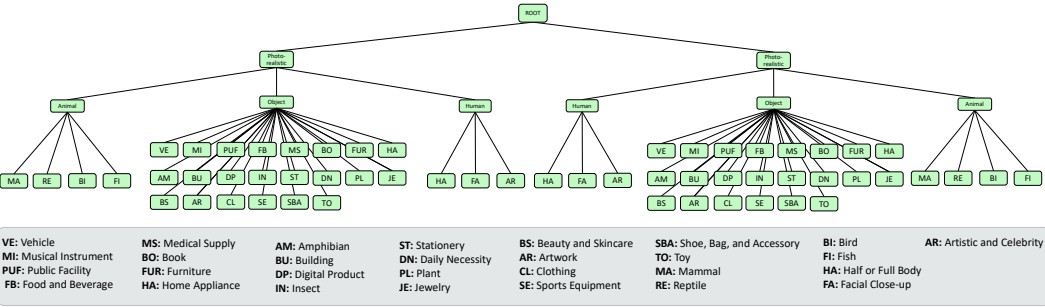

Figure 8: **The hierarchical category system.** We developed a three-level category hierarchy by integrating data from existing large-scale datasets and open-source encyclopedic resources.

Table 4: The correspondence between the third-level categories and the candidate category labels

| Candidate Category Labels | | | | | | | The Third-level Category |
|---|---|---|---|---|---|---|---|
| reptile | lizard | dinosaur | turtle | crocodile | chameleon | gecko | Reptile |
| fly | firefly | ant | butterfly | ladybug | locust | dragonfly | Insect |
| amphibian | frog | bullfrog | toad | salamander | | | Amphibian |
| fish | goldfish | seahorse | shark | tilapia | | | Fish |
| bird | chicken | duck | owl | swan | goose | rooster | Bird |
| hen | turkey | swallow | crow | pigeon | | | |
| mammal | cat | dog | horse | sheep | cow | elephant | Mammal |
| bear | squirrel | giraffe | lion | monkey | tiger | bunny | |
| goat | pig | kangaroo | rhinoceros | deer | hippo | platypus | |
| whale | aardvark | rabbit | zebra | mouse | | | |
| street | fountain | fire hydrant | traffic light | sign | parking meter | goal net | Public Facility |
| field goal post | soccer net | basketball court | bus stop sign | | | | |
| furniture | dining table | sofa | chair | couch | bed | desk | Furniture |
| table | coffee table | side table | bench | cabinet | mirror | carpet | |
| window | door | chandelier | table lamp | gate | | | |
| flower | potted plant | tree | sunflower | cactus | lavender | | Plant |
| cookie | milk | pancake | pasta | grape | cereal | bean | Food and Beverage |
| pineapple | carrot | broccoli | banana | orange | strawberry | apple | |
| bread | sandwich | cake | pizza | soup | meat | pumpkin | |
| cheese | cupcake | donut | hot dog | bacon | egg | tomato | |
| dryer | fridge | refrigerator | microwave | oven | toaster | washer | Home Appliance |
| blender | hair drier | fan (ceil/floor) | printer | fax machine | copier | | |
| necklace | bracelet | ring | pendant | brooch | anklet | | Jewelry |
| wheelchair | gauze | crutch | stethoscope | syringe | | | Medical Supply |
| pants | jacket | long sleeve shirt | short sleeve shirt | pajamas | underpants | shirt | Clothing |
| shorts | scarf | tie | super hero costume | sock | | | |
| book | magazine | textbook | dictionary | biography | | | Book |
| bat | skis | snowboard | tennis racket | basketball hoop | baseball glove | soccer ball | Sports Equipment |
| sports ball | basketball | football | tennis net | hoop | | | |
| flip flop | handbag | glove | shoe | backpack | | | Shoe, Bag, and Accessory |
| pen | pencil | fax machine | stapler | | | | Stationery |
| vehicle | car | van | truck | bus | train | boat | Vehicle |
| sailboat | raft | airplane | helicopter | hot air balloon | rocket | bicycle | |
| unicycle | motorcycle | motorbike | skateboard | | | | |
| house | building | roof | bridge | church | | | Building |
| picture frame | movie (disc) | playing cards | table cloth | | | | Artwork |
| musical instrument | guitar | drum | flute | violin | | | Musical Instrument |
| telephone | laptop | computer | tablet | ipad | iphone | cell phone | Digital Product |
| remote | mouse | keyboard | printer | desktop | copier | radio | |
| kite | toy cars | toy | legos | robot | doll | | |
| hair brush | toner | blush | serum | emulsion | sunscreen | | Beauty and Skincare |
| bottle | plate | cup | bowl | teapot | fork | knife | Daily Necessity |
| spoon | clock | toothbrush | vase | towel | candle | balloon | |
| box | chopping board | ladder | basket | pillow | power outlet | light switch | |
| person | | | | | | | Person |

# B DETAILS OF KEYWORDS COLLECTION

The keywords utilized during the image collection process are presented in Table 5. During the keyword collection process, we utilized the following prompt for GPT-4o:

*"You are a researcher with extensive knowledge of various real-world entity classifications. Given a specific category, please generate detailed, non-redundant instances relevant to this category. The category is {}.*
*The corresponding instances are as follows:"*

Table 5: Based on the categories, we employ GPT-4o to generate keyword associations and further enhanced the results by incorporating manually curated keywords.

| The Third-level Category | Keywords | | | | | | |
|---|---|---|---|---|---|---|---|
| Vehicle | van
pickup truck | steam locomotive
bicycle | car
boat | airplane
taxi | UFO
motorcycle | hot air balloon
subway | oil tanker |
| Musical Instrument | guitar pick
suona | electronic drum
saxophone | digital piano
harmonica | guitar
cello | snare drum
violin | flute
pipa | african drum
erhu |
| Public Facility | fire extinguisher | traffic sign | street lamp | street | station | | |
| Food and Beverage | edible oil
pineapple
apple
vegetable | instant noodles
milk
donut
chicken | water
orange
durian
noodles | pastries
avocado
sports drink
hamburger | coffee
can
canned health products
salad | biscuits
juice
egg
chocolate | edible salt
milk powder
rice
yogurt |
| Medical Supply | band-aid
blood glucose meter | medicine
crutch | wheelchair
stethoscope | disinfectant
syringe | first aid kit | medication | medicine bottle |
| Book | yearbook
book | almanac
notebook | workbook
magazine | comic
dictionary | encyclopedia | atlas | pamphlet |
| Furniture | shelf
barber chair
ottoman | makeup mirror
office chair
bookcase | stool
bathroom mirror
wardrobe | bathroom cabinet
chair
nightstand | cabinet
sofa
dresser | bean bag chair
dining table | children's chair
bed |
| Home Appliance | beauty device
microphone
television | kettle
refrigerator
oven | speaker
hair dryer
juicer | massage chair
humidifier
dishwasher | vacuum cleaner
washing machine | rice cooker
microwave oven | robot vacuum
curling iron |
| Amphibian | newt
frog | olm
toad | bullfrog
caecilian | wood frog
salamander | Surinam toad | alpine newt | glass frog |
| Building | house
hut | apartment building
leaning tower of pisa | duplex house
pyramid | church
statue of liberty | temple of heaven
eiffel tower | castle | golden gate bridge |
| Digital Product | smart robot
printer
smartwatch | headphones
camcorder
vintage camera | e-book reader
camera
monitor | desktop computer
smart camera
drone | roll of film
laptop
projector | router
mobile phone
fitness tracker | tablet
walkie-talkie |
| Insect | shrimp | crab | ant | grasshopper | butterfly | | |
| Stationery | glue stick
stapler | globe
crayon | calculator
ballpoint pen | floppy disk
eraser | tape measure | scissors | compass |
| Daily Necessity | hammer
birdcage
glass jar
electric saw | candle
alarm clock
vase
mop | mug
spoon
hanger
broom | teapot
bowl
soap dish
comb | berry bowl
toothbrush
frying pan | curtain
shower gel
baby bottle | pillow
clock
kitchen knife |
| Plant | cactus
mint | coconut tree
rose | tree
sunflower | potted plant
tulip | peony
cactus | willow tree
lavender | maple leaf |
| Jewelry | earrings
tiara
gold bar | ring
crown
necklace | crystal
stud
pendant | bracelet
chain
brooch | watch
gemstone
anklet | hair accessory
choker
locket | beaded bracelet
hairpin |
| Beauty and Skincare | perfume
blush | makeup brush
eye shadow | lotion
facial serum | sunscreen spray
emulsion | face cream
serum | nail polish
mascara | toner
lipstick |
| Artwork | bouquet of flowers
sculpture | clay sculpture
ceramic craft | wood carving
mural | classical bust
relief | stone carving | catstatue | mugskulls |
| Clothing | dress
pants | baby clothes
shirt | clothing
down jacket | jeans
coat | sweatshirt
skirt | T-shirt
shorts | socks
vest |
| Sports Equipment | tennis
adjustable bench
treadmill | ball
knee pad
skateboard | tent
backpack
barbell | trekking poles
soccer
dumbbell | yoga mat
sleeping bag | billiard
baseball | badminton
flamingo float |
| Shoe, Bag and Accessory | suitcase
glasses
hat | slippers
sandals
backpack | sunglasses
shoes
cap | canvas shoes
luggage purse
tie | high-top shoes
fancy boot
handbag | sports shoes
belt
sandals | scarf
sneaker |
| Toy | actionfigure
robot
minion | monster toy
motorbike toy
smart robot | car
magic cube
robot toy | egg
poop emoji
toy | duck toy
sloth plushie
wolf plushie | teddy bear
bear plushie
doll | balloon
red cartoon
Eevee figurine |
| Mammal | rabbit
panda
alpaca | fox
elephant
puppy | wolf
llama
monkey | Siamese cat
tiger
kitten | polar bear
dog
dolphin | cat
raccoon
French bulldog | deer
lion |
| Reptile | cobra
turtle | gecko
sea turtle | rattlesnake
soft-shelled turtle | crocodile
snake | chameleon
lizard | alligator | iguana |
| Bird | heron
woodpecker
peacock
bird | pigeon
nightingale
swallow
canary | toucan
duck
owl
sparrow | parrot
turkey
kingfisher
rooster | stork
chicken
hawk | flamingo
crow
dove | penguin
eagle
anchovy |
| Fish | shark
skate | tropical fish
swordfish | jellyfish
herring | goldfish
sardine | perch
carp | eel
salmon | monkfish
tuna |
| Person | person | | | | | | |

## C  DETAILS OF PROMPT GENERATION

The specific instructions used in prompt generation are detailed in Figure 5. During the actual generation process, some of the prompts produced by GPT-4o did not meet the required criteria. Therefore, we instructed GPT-4o to generate multiple prompts for each image, and then manually selected those that best matched the intended scenarios. Figure 14 presents the results generated by different methods in this study, along with their corresponding prompts.

# D  ADDITIONAL DISCUSSIONS AND DETAILS OF MODEL PERFORMANCE

## D.1  ADDITIONAL DISCUSSIONS

**Analysis of The First-Level Category**    The primary categories are divided into photorealistic and non-photorealistic. Table 6 and Figure 9 present the performance of different methods on these two categories across three evaluation dimensions. The results show that: *(1) Subject Preservation:* Almost all methods perform better on photorealistic categories than on non-photorealistic ones. We speculate that this is because, when referencing subjects from non-photorealistic categories, these methods tend to generate photorealistic images based on the prompt, which results in lower subject consistency. *(2) Prompt Following:* The performance gap between photorealistic and non-photorealistic categories is relatively small. This can be attributed to the fact that CLIP-T focuses primarily on the semantic information of the image. As long as the generated subject matches the category and general appearance described in the prompt, the CLIP-T score will not be significantly reduced. *(3) Image quality:* There is little difference in performance between photorealistic and non-photorealistic categories. This indicates that the distinction between these two categories does not affect the quality of image generation, and the HPSv2 metric does not show a preference for either category.

**Analysis of The Second-Level Category**    The secondary categories under both the realistic and non-realistic primary categories are further subdivided into objects, humans, and animals. Table 7 and Figure 10 present the performance of various methods across these three dimensions for both realistic and non-realistic categories. The results demonstrate that, irrespective of whether the primary category is realistic or non-realistic, the scores for the subject preservation dimension are consistently lower for the human category across nearly all models. As detailed in Table 8, this phenomenon can be attributed to the distribution of difficulty levels within the human category, where the proportions of simple, medium, and hard cases are 1.96%, 50.98%, and 47.06%, respectively. In contrast, the object and animal categories exhibit a higher proportion of subjects at the simple difficulty level and a lower proportion at the hard difficulty level, which likely contributes to their relatively higher subject preservation scores.

**Implications for Technical Approaches**    (1) Figure 11 shows that, as base models and model architectures are updated, the performance boundary of these models consistently expands outward. Table 9 presents all the base models used by each method. It can be observed that the top-performing methods consistently employ relatively recent text-to-image base models. For instance, UNO utilizes FLUX as its foundational model. This observation suggests that the adoption of advanced text-to-image base models is a critical factor in enhancing performance on subject-driven T2I tasks. (2) Historically, fine-tuning methods have generally outperformed encoder-based approaches in terms of subject preservation. This advantage is attributed to their ability to better retain the original text-image conditional distribution by fine-tuning on images of the specified subject. In contrast, encoder-based methods often encounter interference during feature injection, which can hinder precise prompt alignment. However, with the development of more advanced encoding techniques, the adoption of larger and more powerful base models, and the availability of extensive training datasets, encoder-based methods have demonstrated significantly improved performance. From an application standpoint, fine-tuning methods require substantial computational resources for optimization and often exhibit limited generalization capabilities. In contrast, encoder-based methods are less constrained by these limitations, making them more practical for future applications. Nevertheless, our analysis indicates that current encoder-based methods still face challenges in accurately reconstructing subjects with high-frequency details in images. This limitation may stem from the characteristics of commonly used image encoders, such as CLIP, which tend to prioritize semantic information over fine-grained details. Consequently, future research should focus on enhancing the restoration of challenging subject details.

## D.2  DETAILS OF MODEL PERFORMANCE

In this section, we present the detailed evaluation results for each metric across all models. To comprehensively evaluate the effectiveness of different metrics for assessing subject consistency, we calculated multiple metrics for each method. The detailed results are presented in Table 10, 11, 12. In section 4, we present the performance of all methods across images with different difficulty

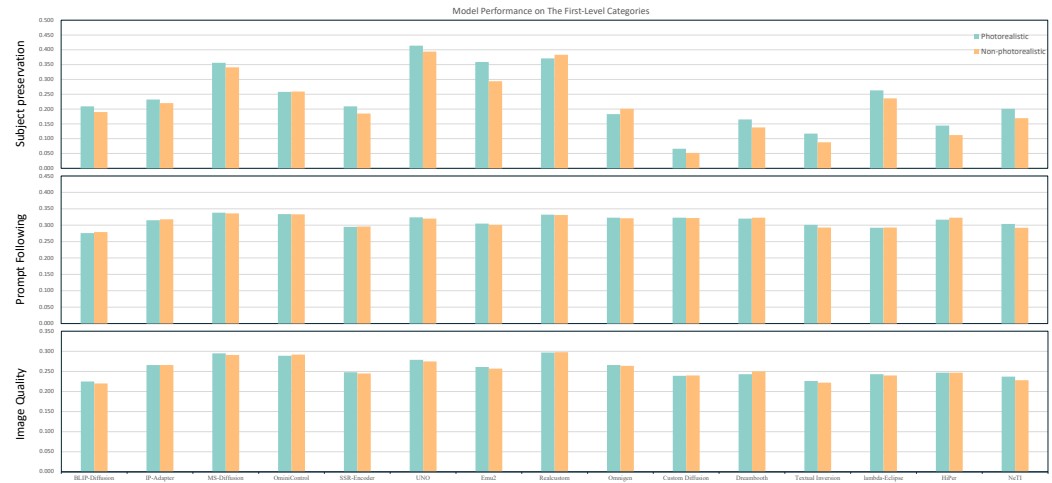

Figure 9: Comparison of bar charts for DSH-Bench scores in different first-level categories.

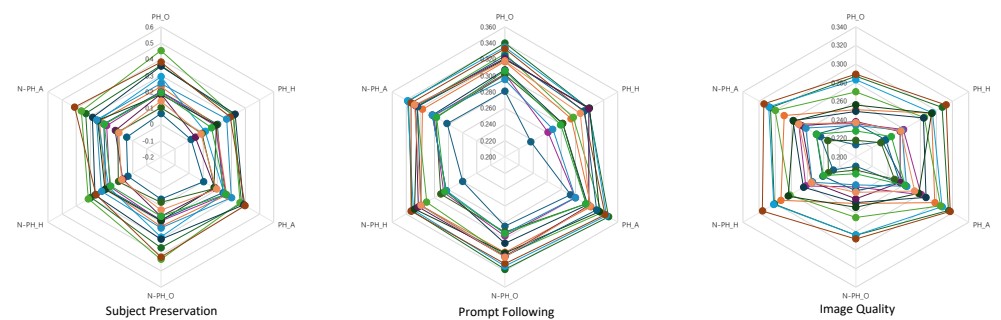

Figure 10: Comparison of radar charts for DSH-Bench scores in different second-level categories.

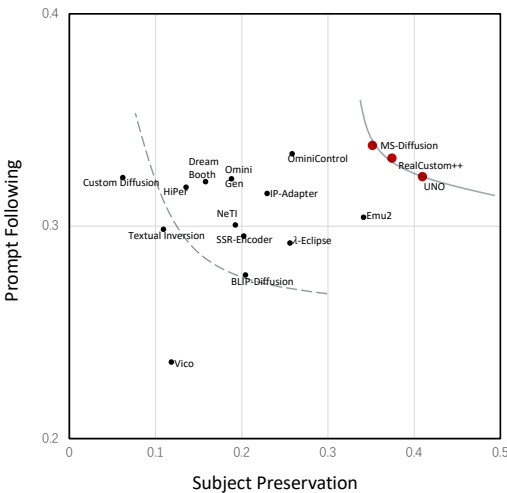

Figure 11: Pareto front diagram illustrating model performance across both subject and prompt dimensions. The red points in the diagram represent the current Pareto-optimal solutions.

levels, different prompt scenarios, and multiple categories. We show the specific metric values in Table 13, 14, 15, 16. Table 9 shows the full ranking among all methods.

Table 6: We evaluate the performance of various methods on DSH-Bench dataset, specifically analyzing their effectiveness across **the first-level categories**. PH: Photorealistic. N-PH: Non-Photorealistic.

| Method | Subject Preservation↑ | | Prompt Following↑ | | Image Quality↑ | |
|--------|------|------|------|------|------|------|
| | PH | N-PH | PH | N-PH | PH | N-PH |
| BLIP-Diffusion | 0.209 | 0.190 | 0.276 | 0.279 | 0.225 | 0.220 |
| IP-Adapter | 0.232 | 0.220 | 0.315 | 0.318 | 0.266 | 0.266 |
| MS-Diffusion | 0.356 | 0.341 | **0.338** | **0.336** | 0.295 | 0.291 |
| OminiControl | 0.258 | 0.259 | 0.334 | 0.333 | 0.289 | 0.292 |
| SSR-Encoder | 0.209 | 0.185 | 0.295 | 0.296 | 0.248 | 0.245 |
| UNO | **0.414** | **0.394** | 0.324 | 0.320 | 0.279 | 0.275 |
| Emu2 | 0.359 | 0.294 | 0.305 | 0.301 | 0.261 | 0.257 |
| RealCustom++ | 0.371 | 0.383 | 0.332 | 0.331 | **0.297** | **0.298** |
| OmniGen | 0.183 | 0.201 | 0.323 | 0.321 | 0.266 | 0.264 |
| Custom Diffusion | 0.066 | 0.052 | 0.323 | 0.322 | 0.239 | 0.240 |
| DreamBooth | 0.165 | 0.138 | 0.320 | 0.323 | 0.243 | 0.250 |
| Textual Inversion | 0.117 | 0.088 | 0.301 | 0.293 | 0.226 | 0.222 |
| $\lambda$-Eclipse | 0.263 | 0.236 | 0.292 | 0.293 | 0.243 | 0.240 |
| HiPer | 0.144 | 0.112 | 0.317 | 0.323 | 0.247 | 0.247 |
| NeTI | 0.201 | 0.169 | 0.304 | 0.292 | 0.237 | 0.228 |
| *Aver.* | 0.236 | 0.217 | 0.313 | 0.312 | 0.257 | 0.256 |

# E   IMPLEMENTATION DETAILS

## E.1   EXPERIMENTAL DETAILS OF EXISTING METHODS

The configurations for the training hyperparameters used in training-based methods on DSH-Bench are detailed in Table 17. To ensure a fair comparison in inference stage, we generated four images for each prompt of every image. The final evaluation metrics were calculated as the average score across these four images.

## E.2   DETAILS OF SICS IMPLEMENTATION

**How the SICS metric is computed**   Rather than relying on simple embedding distances, SICS instruction-tunes multimodal large language models to directly produce fine-grained subject-consistency scores (0–5) with accompanying explanations. These criteria align closely with human judgments. As illustrated in Figure 12, SICS employs prompts during fine-tuning that explicitly target subject consistency while de-emphasizing global image semantics. This design reduces background and style confounds that bias CLIP-like methods, yielding tighter alignment with the core requirements of subject-consistency evaluation. Consequently, SICS focuses on core visual attributes rather than high-level semantics. A further advantage of SICS is its scoring granularity, which mitigates the "score saturation" phenomenon commonly observed in the upper range of GPT-4o evaluations. Figure 2 shows representative cases in which SICS aligns more closely with human assessments.

**Evaluation Instruction**   Figure 12 illustrates the annotation criteria of the training dataset as well as the training process.

**Datasets**   We collected a substantial number of image pairs. To ensure data quality, we applied standardized filtering and preprocessing procedures, such as enforcing a minimum image resolution of 512 pixels. Additionally, we employed Qwen2.5-VL-72B to conduct preliminary screening. After this automated filtering, five annotators manually annotated the remaining image pairs according to the guidelines illustrated in Figure 12.

**Training Details**   We fine-tuned Qwen2.5-VL-7B on the manually annotated dataset described above. All experiments were conducted using 8 GPUs. For the learning rate, we experimented with the set 1e5. The batch size per device was set to 4, with a gradient accumulation step of 8.

Table 7: We evaluate the performance of various methods on DSH-Bench dataset, specifically analyzing their effectiveness across **the second-level categories**. PH: Photorealistic, N-PH: Non-Photorealistic, O: Object, A: Animal, H: Human.

| Method | Subject Preservation | | | | | |
|---|---|---|---|---|---|---|
| | PH_O | PH_H | PH_A | N-PH_O | N-PH_H | N-PH_A |
| BLIP-Diffusion | 0.202 | 0.201 | 0.24 | 0.186 | 0.189 | 0.206 |
| IP-Adapter | 0.232 | 0.193 | 0.267 | 0.226 | 0.188 | 0.237 |
| MS-Diffusion | 0.362 | 0.315 | 0.371 | 0.358 | 0.296 | 0.333 |
| OminiControl | 0.293 | 0.114 | 0.249 | 0.291 | 0.17 | 0.247 |
| SSR-Encoder | 0.199 | 0.186 | 0.26 | 0.193 | 0.162 | 0.185 |
| UNO | **0.453** | 0.312 | 0.361 | **0.428** | **0.315** | 0.365 |
| Emu2 | 0.358 | **0.326** | 0.387 | 0.305 | 0.266 | 0.285 |
| RealCustom++ | 0.383 | 0.291 | **0.396** | 0.415 | 0.26 | **0.412** |
| OmniGen | 0.183 | 0.194 | 0.176 | 0.19 | 0.196 | 0.249 |
| Custom Diffusion | 0.067 | 0.014 | 0.103 | 0.059 | 0.035 | 0.043 |
| DreamBooth | 0.188 | 0.044 | 0.184 | 0.164 | 0.07 | 0.124 |
| Textual Inversion | 0.104 | 0.091 | 0.184 | 0.078 | 0.101 | 0.105 |
| $\lambda$-Eclipse | 0.252 | 0.266 | 0.3 | 0.236 | 0.221 | 0.256 |
| HiPer | 0.143 | 0.083 | 0.195 | 0.126 | 0.079 | 0.098 |
| NeTI | 0.195 | 0.159 | 0.259 | 0.164 | 0.156 | 0.201 |
| *Aver.* | 0.241 | 0.186 | 0.262 | 0.228 | 0.180 | 0.223 |

| Method | Prompt Following | | | | | |
|---|---|---|---|---|---|---|
| | PH_O | PH_H | PH_A | N-PH_O | N-PH_H | N-PH_A |
| BLIP-Diffusion | 0.281 | 0.237 | 0.293 | 0.285 | 0.26 | 0.282 |
| IP-Adapter | 0.317 | 0.294 | 0.322 | 0.317 | 0.319 | 0.317 |
| MS-Diffusion | **0.340** | 0.319 | **0.347** | **0.338** | 0.332 | 0.337 |
| OminiControl | 0.335 | 0.319 | 0.344 | 0.334 | 0.33 | **0.338** |
| SSR-Encoder | 0.302 | 0.261 | 0.3 | 0.297 | 0.287 | 0.301 |
| UNO | 0.327 | 0.297 | 0.337 | 0.321 | 0.311 | 0.325 |
| Emu2 | 0.307 | 0.282 | 0.317 | 0.306 | 0.283 | 0.303 |
| RealCustom++ | 0.333 | 0.312 | 0.342 | 0.331 | **0.333** | 0.333 |
| OmniGen | 0.320 | **0.320** | 0.334 | 0.318 | 0.328 | 0.324 |
| Custom Diffusion | 0.324 | 0.313 | 0.33 | 0.322 | 0.319 | 0.324 |
| DreamBooth | 0.321 | 0.319 | 0.319 | 0.322 | 0.323 | 0.327 |
| Textual Inversion | 0.301 | 0.282 | 0.315 | 0.292 | 0.291 | 0.298 |
| $\lambda$-Eclipse | 0.295 | 0.268 | 0.3 | 0.294 | 0.283 | 0.303 |
| HiPer | 0.318 | 0.307 | 0.32 | 0.323 | 0.319 | 0.328 |
| NeTI | 0.306 | 0.279 | 0.315 | 0.294 | 0.285 | 0.297 |
| *Aver.* | 0.315 | 0.294 | 0.322 | 0.313 | 0.307 | 0.316 |

| Method | Image Quality | | | | | |
|---|---|---|---|---|---|---|
| | PH_O | PH_H | PH_A | N-PH_O | N-PH_H | N-PH_A |
| BLIP-Diffusion | 0.213 | 0.233 | 0.262 | 0.21 | 0.228 | 0.244 |
| IP-Adapter | 0.251 | 0.294 | 0.298 | 0.25 | 0.293 | 0.289 |
| MS-Diffusion | 0.287 | 0.307 | 0.315 | 0.284 | 0.301 | 0.306 |
| OminiControl | 0.283 | 0.295 | 0.307 | 0.284 | 0.302 | 0.308 |
| SSR-Encoder | 0.236 | 0.259 | 0.281 | 0.232 | 0.262 | 0.271 |
| UNO | 0.270 | 0.285 | 0.305 | 0.265 | 0.282 | 0.3 |
| Emu2 | 0.249 | 0.284 | 0.287 | 0.249 | 0.265 | 0.278 |
| RealCustom++ | **0.289** | **0.312** | **0.317** | **0.288** | **0.316** | **0.314** |
| OmniGen | 0.256 | 0.294 | 0.278 | 0.254 | 0.284 | 0.277 |
| Custom Diffusion | 0.236 | 0.237 | 0.255 | 0.236 | 0.241 | 0.249 |
| DreamBooth | 0.238 | 0.255 | 0.255 | 0.245 | 0.254 | 0.267 |
| Textual Inversion | 0.218 | 0.231 | 0.248 | 0.214 | 0.234 | 0.235 |
| $\lambda$-Eclipse | 0.234 | 0.257 | 0.263 | 0.23 | 0.254 | 0.262 |
| HiPer | 0.237 | 0.256 | 0.273 | 0.238 | 0.255 | 0.271 |
| NeTI | 0.228 | 0.244 | 0.261 | 0.218 | 0.24 | 0.249 |
| *Aver.* | 0.248 | 0.270 | 0.280 | 0.246 | 0.267 | 0.275 |

Table 8: Subject hard level distribution under the second category

| Benchmark | Photorealistic | | | | | | | | |
|---|---|---|---|---|---|---|---|---|---|
| | Object | | | Human | | | Animal | | |
| | Easy | Medium | Hard | Easy | Medium | Hard | Easy | Medium | Hard |
| DreamBench | 3 | 10 | 7 | 0 | 0 | 0 | 0 | 7 | 2 |
| DreamBench++ | 6 | 24 | 31 | 0 | 7 | 5 | 0 | 26 | 16 |
| DSH-Bench | 54 | 85 | 84 | 1 | 26 | 24 | 2 | 39 | 21 |

| Benchmark | Non-photorealistic | | | | | | | | |
|---|---|---|---|---|---|---|---|---|---|
| | Object | | | Human | | | Animal | | |
| | Easy | Medium | Hard | Easy | Medium | Hard | Easy | Medium | Hard |
| DreamBench | 1 | 0 | 0 | 0 | 0 | 0 | 0 | 0 | 0 |
| DreamBench++ | 2 | 1 | 1 | 0 | 1 | 7 | 0 | 1 | 2 |
| DSH-Bench | 28 | 32 | 15 | 1 | 4 | 20 | 3 | 11 | 8 |

Table 9: **The full DSH-Bench leaderboard.** The models are ranked by the final score $S_h$.

| Method | T2I Model | Subject Preservation | Prompt Following | Image Quality | $S_h\uparrow$ |
|---|---|---|---|---|---|
| RealCustom++ | SDXL | 0.375 | 0.332 | **0.298** | **0.110** |
| UNO | FLUX.1-dev | **0.409** | 0.323 | 0.278 | 0.109 |
| MS-Diffusion | SDXL | 0.352 | **0.338** | 0.294 | 0.107 |
| Emu2 | SDXL | 0.341 | 0.304 | 0.260 | 0.089 |
| OminiControl | FLUX.1-schnell | 0.258 | 0.334 | 0.290 | 0.085 |
| IP-Adapter | SDXL | 0.229 | 0.315 | 0.266 | 0.071 |
| $\lambda$-Eclipse | SDXL | 0.256 | 0.292 | 0.242 | 0.069 |
| OmniGen | SDXL | 0.188 | 0.322 | 0.265 | 0.062 |
| SSR-Encoder | SD v1.5 | 0.202 | 0.295 | 0.247 | 0.059 |
| NeTI | SD v1.4 | 0.192 | 0.301 | 0.234 | 0.056 |
| BLIP-Diffusion | SD v1.5 | 0.204 | 0.277 | 0.223 | 0.054 |
| DreamBooth | SD v1.5 | 0.158 | 0.321 | 0.245 | 0.052 |
| HiPer | SD v1.4 | 0.135 | 0.318 | 0.247 | 0.045 |
| Textual Inversion | SD v1.5 | 0.109 | 0.299 | 0.225 | 0.035 |
| ViCo | SD v1.4 | 0.118 | 0.236 | 0.186 | 0.029 |
| Custom Diffusion | SD v1.4 | 0.062 | 0.323 | 0.240 | 0.023 |

Table 10: Evaluation of Subject-driven T2I generation model on **DreamBench**. C, D, Img, T and I represent CLIP, DINO, Image, Text and Image, respectively.

| Method | Subject Preservation | | | | | Prompt Following | | Image Quality | | |
|---|---|---|---|---|---|---|---|---|---|---|
| | C-B-I↑ | C-L-I↑ | D-I↑ | D-v2-I↑ | SICS↑ | C-B-T↑ | C-L-T↑ | ImageReward↑ | PickScore↑ | HPSv2↑ |
| BLIP-Diffusion | 0.824 | 0.784 | 0.684 | 0.640 | 0.229 | 0.291 | 0.239 | 0.420 | 0.599 | 0.267 |
| IP-Adapter | 0.836 | **0.820** | 0.684 | 0.648 | 0.230 | 0.321 | 0.263 | 0.616 | 0.600 | 0.291 |
| MS-Diffusion | 0.814 | 0.796 | 0.732 | 0.687 | 0.316 | 0.332 | 0.279 | 0.775 | 0.600 | 0.311 |
| OminiControl | 0.784 | 0.772 | 0.614 | 0.555 | 0.279 | **0.336** | **0.284** | 0.793 | 0.593 | 0.306 |
| SSR-Encoder | 0.830 | 0.802 | 0.732 | 0.677 | 0.231 | 0.302 | 0.251 | 0.535 | 0.600 | 0.282 |
| UNO | 0.827 | 0.801 | 0.744 | **0.716** | **0.409** | 0.317 | 0.259 | 0.725 | **0.602** | 0.304 |
| Emu2 | **0.838** | 0.818 | 0.737 | 0.704 | 0.360 | 0.291 | 0.235 | 0.463 | 0.599 | 0.272 |
| RealCustom++ | 0.794 | 0.770 | **0.746** | 0.698 | 0.377 | 0.325 | 0.278 | **0.813** | 0.601 | **0.316** |

Table 11: Evaluation of Subject-driven T2I generation model on **DreamBench++**. C, D, Img, T and I represent CLIP, DINO, Image, Text and Image, respectively.

| Method | Subject Preservation | | | | | Prompt Following | | Image Quality | | |
|---|---|---|---|---|---|---|---|---|---|---|
| | C-B-I↑ | C-L-I↑ | D-I↑ | D-v2-I↑ | SICS↑ | C-B-T↑ | C-L-T↑ | ImageReward↑ | PickScore↑ | HPSv2↑ |
| BLIP-Diffusion | 0.836 | 0.809 | 0.691 | 0.664 | 0.216 | 0.279 | 0.225 | 0.260 | 0.591 | 0.249 |
| IP-Adapter | **0.846** | **0.845** | 0.659 | 0.646 | 0.244 | 0.320 | 0.266 | 0.554 | 0.593 | 0.291 |
| MS-Diffusion | 0.812 | 0.823 | 0.666 | 0.653 | 0.346 | **0.339** | **0.285** | 0.729 | 0.593 | 0.309 |
| OminiControl | 0.761 | 0.780 | 0.551 | 0.566 | 0.268 | 0.336 | 0.284 | **0.793** | 0.593 | 0.308 |
| SSR-Encoder | 0.814 | 0.815 | 0.639 | 0.611 | 0.202 | 0.302 | 0.252 | 0.455 | 0.591 | 0.276 |
| UNO | 0.828 | 0.835 | 0.694 | 0.694 | **0.410** | 0.321 | 0.263 | 0.673 | 0.592 | 0.293 |
| Emu2 | 0.833 | 0.823 | 0.665 | 0.632 | 0.343 | 0.309 | 0.255 | 0.460 | 0.593 | 0.275 |
| RealCustom++ | 0.819 | 0.810 | **0.714** | **0.706** | 0.380 | 0.330 | 0.280 | 0.710 | **0.594** | **0.314** |

Table 12: Evaluation of Subject-driven T2I generation model on **DSH_Bench**. C, D, Img, T and I represent CLIP, DINO, Image, Text and Image, respectively.

| Method | Subject Preservation | | | | | Prompt Following | | Image Quality | | |
|---|---|---|---|---|---|---|---|---|---|---|
| | C-B-I↑ | C-L-I↑ | D-I↑ | D-v2-I↑ | SICS↑ | C-B-T↑ | C-L-T↑ | ImageReward↑ | PickScore↑ | HPSv2↑ |
| BLIP-Diffusion | 0.806 | 0.770 | 0.632 | 0.573 | 0.204 | 0.277 | 0.225 | 0.239 | 0.591 | 0.223 |
| IP-Adapter | 0.824 | 0.812 | 0.610 | 0.577 | 0.229 | 0.315 | 0.263 | 0.493 | 0.594 | 0.266 |
| MS-Diffusion | 0.786 | 0.783 | 0.623 | 0.600 | 0.352 | **0.338** | 0.287 | 0.705 | **0.595** | 0.294 |
| OminiControl | 0.721 | 0.736 | 0.462 | 0.461 | 0.258 | 0.334 | **0.288** | **0.787** | 0.594 | 0.290 |
| SSR-Encoder | 0.803 | 0.787 | 0.613 | 0.554 | 0.202 | 0.295 | 0.246 | 0.369 | 0.593 | 0.247 |
| UNO | 0.781 | 0.784 | 0.607 | 0.599 | **0.409** | 0.323 | 0.272 | 0.705 | 0.594 | 0.278 |
| Emu2 | 0.815 | 0.804 | 0.631 | 0.606 | 0.341 | 0.304 | 0.256 | 0.441 | 0.594 | 0.260 |
| RealCustom++ | 0.781 | 0.769 | 0.645 | 0.624 | 0.374 | 0.332 | 0.285 | 0.695 | 0.595 | **0.298** |
| OmniGen | 0.696 | 0.678 | 0.436 | 0.326 | 0.188 | 0.322 | 0.274 | 0.586 | 0.592 | 0.265 |
| Custom Diffusion | 0.648 | 0.648 | 0.283 | 0.230 | 0.062 | 0.323 | 0.282 | 0.481 | 0.590 | 0.239 |
| DreamBooth | 0.714 | 0.713 | 0.451 | 0.420 | 0.158 | 0.321 | 0.279 | 0.489 | 0.591 | 0.245 |
| Textual Inversion | 0.689 | 0.683 | 0.372 | 0.320 | 0.109 | 0.299 | 0.253 | 0.340 | 0.590 | 0.225 |
| λ-Eclipse | 0.852 | **0.833** | **0.676** | **0.638** | 0.256 | 0.292 | 0.239 | 0.349 | 0.594 | 0.242 |
| HiPer | 0.749 | 0.734 | 0.449 | 0.431 | 0.135 | 0.318 | 0.274 | 0.410 | 0.592 | 0.247 |
| NeTI | 0.762 | 0.743 | 0.525 | 0.491 | 0.192 | 0.301 | 0.256 | 0.338 | 0.592 | 0.234 |

Table 13: We evaluated the performance of various methods on DSH-Bench dataset, specifically analyzing their effectiveness across **the third-level categories (under photorealistic)**. Subject preservation, prompt following, and image quality are evaluated using SICS, CLIP-T, and HPSv2, respectively. **VE**: Vehicle, **MI**: Musical Instrument, **PUF**: Public Facility, **FB**: Food and Beverage, **MS**: Medical Supply, **BO**: Book, **FUR**: Furniture, **HA**: Home Appliance, **AM**: Amphibian, **BU**: Building, **DP**: Digital Product, **IN**: Insect, **ST**: Stationery, **DN**: Daily Necessity, **PL**: Plant, **JE**: Jewelry, **BS**: Beauty and Skincare, **AR**: Artwork, **CL**: Clothing, **SE**: Sports Equipment, **SBA**: Shoe, Bag, and Accessory, **TO**: Toy, **MA**: Mammal, **RE**: Reptile, **BI**: Bird, **FI**: Fish, **HF**: Half or Full Body, **FA**: Facial Close-up, **AC**: Artistic and Celebrity.

**Subject Preservation**

| Method | VE | MI | PUF | FB | MS | BO | FUR | HA | AM | BU | DP | IN | ST | DN | PL | JE | BS | AR | CL | SE | SBA | TO | MA | RE | BI | FI | HF | FA | AC |
|---|---|---|---|---|---|---|---|---|---|---|---|---|---|---|---|---|---|---|---|---|---|---|---|---|---|---|---|---|---|
| BLIP-Diffusion | 0.246 | 0.181 | 0.142 | 0.182 | 0.151 | 0.187 | 0.220 | 0.195 | 0.242 | 0.285 | 0.192 | 0.185 | 0.257 | 0.164 | 0.176 | 0.183 | 0.247 | 0.225 | 0.209 | 0.205 | 0.268 | 0.192 | 0.202 | 0.194 | | | | | |
| IP-Adapter | 0.217 | 0.244 | 0.183 | 0.217 | 0.207 | 0.137 | 0.287 | 0.228 | 0.208 | 0.300 | 0.216 | 0.123 | 0.257 | 0.230 | 0.248 | 0.189 | 0.232 | 0.162 | 0.292 | 0.246 | 0.262 | 0.218 | 0.303 | 0.190 | 0.248 | 0.225 | 0.195 | 0.223 | 0.139 |
| MS-Diffusion | 0.293 | 0.368 | 0.292 | 0.313 | 0.378 | 0.290 | 0.451 | 0.361 | 0.592 | 0.327 | 0.346 | 0.244 | 0.353 | 0.353 | 0.397 | 0.386 | 0.307 | 0.348 | 0.396 | 0.433 | 0.336 | 0.374 | 0.345 | 0.415 | 0.337 | 0.337 | 0.302 | 0.273 | |
| OminiControl | 0.187 | 0.329 | 0.333 | 0.312 | 0.314 | 0.172 | 0.292 | 0.335 | 0.275 | 0.252 | 0.296 | 0.225 | 0.296 | 0.311 | 0.300 | 0.227 | 0.296 | 0.293 | 0.258 | 0.297 | 0.336 | 0.324 | 0.246 | 0.208 | 0.308 | 0.208 | 0.135 | 0.082 | 0.107 |
| SSR-Encoder | 0.187 | 0.237 | 0.156 | 0.191 | 0.201 | 0.150 | 0.315 | 0.202 | 0.192 | 0.286 | 0.183 | 0.146 | 0.158 | 0.195 | 0.293 | 0.112 | 0.158 | 0.143 | 0.204 | 0.231 | 0.185 | 0.203 | 0.301 | 0.197 | 0.209 | 0.215 | 0.159 | 0.228 | 0.188 |
| UNO | 0.387 | 0.411 | 0.446 | 0.444 | 0.461 | 0.330 | 0.580 | 0.442 | 0.442 | 0.461 | 0.442 | 0.325 | 0.402 | 0.457 | 0.427 | 0.396 | 0.418 | 0.419 | 0.515 | 0.502 | 0.524 | 0.436 | 0.359 | 0.322 | 0.420 | 0.300 | 0.367 | 0.258 | 0.247 |
| Emu2 | 0.324 | 0.281 | 0.546 | 0.326 | 0.382 | 0.302 | 0.444 | 0.394 | 0.583 | 0.379 | 0.378 | 0.344 | 0.318 | 0.319 | 0.355 | 0.290 | 0.418 | 0.348 | 0.365 | 0.385 | 0.429 | 0.300 | 0.370 | 0.310 | 0.443 | 0.510 | 0.308 | 0.311 | 0.404 |
| RealCustom++ | 0.340 | 0.421 | 0.371 | 0.340 | 0.328 | 0.228 | 0.479 | 0.395 | 0.500 | 0.406 | 0.347 | 0.327 | 0.358 | 0.399 | 0.432 | 0.349 | 0.325 | 0.347 | 0.389 | 0.488 | 0.369 | 0.417 | 0.394 | 0.342 | 0.450 | 0.390 | 0.306 | 0.270 | 0.283 |
| OmniGen | 0.111 | 0.123 | 0.094 | 0.249 | 0.161 | 0.183 | 0.196 | 0.133 | 0.058 | 0.236 | 0.192 | 0.096 | 0.157 | 0.169 | 0.229 | 0.146 | 0.251 | 0.179 | 0.202 | 0.173 | 0.203 | 0.180 | 0.197 | 0.082 | 0.203 | 0.144 | 0.209 | 0.215 | 0.113 |
| Custom Diffusion | 0.075 | 0.083 | 0.060 | 0.086 | 0.058 | 0.037 | 0.058 | 0.051 | 0.083 | 0.181 | 0.048 | 0.063 | 0.074 | 0.097 | 0.030 | 0.082 | 0.038 | 0.074 | 0.079 | 0.054 | 0.058 | 0.107 | 0.115 | 0.060 | 0.014 | 0.015 | 0.009 | | |
| DreamBooth | 0.225 | 0.180 | 0.206 | 0.206 | 0.151 | 0.077 | 0.224 | 0.232 | 0.200 | 0.242 | 0.158 | 0.190 | 0.195 | 0.220 | 0.307 | 0.135 | 0.163 | 0.105 | 0.131 | 0.187 | 0.167 | 0.186 | 0.181 | 0.170 | 0.190 | 0.208 | 0.048 | 0.035 | 0.050 |
| Textual Inversion | 0.143 | 0.081 | 0.090 | 0.116 | 0.081 | 0.022 | 0.099 | 0.085 | 0.192 | 0.225 | 0.059 | 0.117 | 0.115 | 0.089 | 0.174 | 0.115 | 0.074 | 0.101 | 0.071 | 0.114 | 0.087 | 0.134 | 0.191 | 0.133 | 0.219 | 0.160 | 0.060 | 0.114 | 0.138 |
| λ-Eclipse | 0.280 | 0.215 | 0.117 | 0.215 | 0.214 | 0.170 | 0.346 | 0.248 | 0.283 | 0.312 | 0.253 | 0.116 | 0.211 | 0.249 | 0.275 | 0.243 | 0.211 | 0.193 | 0.298 | 0.283 | 0.309 | 0.238 | 0.340 | 0.227 | 0.272 | 0.231 | 0.261 | 0.273 | 0.268 |
| HiPer | 0.148 | 0.150 | 0.165 | 0.162 | 0.108 | 0.097 | 0.135 | 0.175 | 0.183 | 0.261 | 0.132 | 0.144 | 0.132 | 0.144 | 0.183 | 0.112 | 0.119 | 0.115 | 0.119 | 0.153 | 0.107 | 0.160 | 0.201 | 0.160 | 0.235 | 0.140 | 0.092 | 0.079 | 0.065 |
| NeTI | 0.199 | 0.154 | 0.140 | 0.193 | 0.179 | 0.123 | 0.235 | 0.200 | 0.292 | 0.275 | 0.144 | 0.196 | 0.182 | 0.225 | 0.264 | 0.176 | 0.152 | 0.188 | 0.145 | 0.193 | 0.193 | 0.226 | 0.262 | 0.252 | 0.282 | 0.233 | 0.152 | 0.139 | 0.218 |

**Prompt Following**

| Method | VE | MI | PUF | FB | MS | BO | FUR | HA | AM | BU | DP | IN | ST | DN | PL | JE | BS | AR | CL | SE | SBA | TO | MA | RE | BI | FI | HF | FA | AC |
|---|---|---|---|---|---|---|---|---|---|---|---|---|---|---|---|---|---|---|---|---|---|---|---|---|---|---|---|---|---|
| BLIP-Diffusion | 0.271 | 0.283 | 0.279 | 0.282 | 0.275 | 0.244 | 0.269 | 0.286 | 0.307 | 0.285 | 0.273 | 0.299 | 0.283 | 0.286 | 0.294 | 0.280 | 0.273 | 0.272 | 0.281 | 0.296 | 0.281 | 0.294 | 0.291 | 0.287 | 0.301 | 0.287 | 0.241 | 0.223 | 0.245 |
| IP-Adapter | 0.315 | 0.332 | 0.310 | 0.318 | 0.314 | 0.301 | 0.306 | 0.324 | 0.331 | 0.312 | 0.320 | 0.319 | 0.320 | 0.317 | 0.316 | 0.306 | 0.307 | 0.310 | 0.319 | 0.325 | 0.321 | 0.333 | 0.325 | 0.303 | 0.327 | 0.306 | 0.291 | 0.291 | 0.311 |
| MS-Diffusion | 0.336 | 0.354 | 0.335 | 0.339 | 0.342 | 0.322 | 0.339 | 0.343 | 0.349 | 0.332 | 0.333 | 0.338 | 0.337 | 0.344 | 0.345 | 0.324 | 0.321 | 0.335 | 0.336 | 0.344 | 0.347 | 0.349 | 0.353 | 0.352 | 0.351 | 0.345 | 0.327 | 0.319 | 0.324 |
| OminiControl | 0.328 | 0.345 | 0.333 | 0.337 | 0.335 | 0.317 | 0.336 | 0.351 | 0.331 | 0.327 | 0.333 | 0.328 | 0.337 | 0.337 | 0.339 | 0.331 | 0.331 | 0.327 | 0.336 | 0.342 | 0.341 | 0.347 | 0.349 | 0.328 | 0.344 | 0.321 | 0.317 | 0.321 | 0.322 |
| SSR-Encoder | 0.290 | 0.315 | 0.289 | 0.301 | 0.295 | 0.291 | 0.291 | 0.311 | 0.303 | 0.294 | 0.297 | 0.291 | 0.306 | 0.304 | 0.311 | 0.300 | 0.307 | 0.298 | 0.304 | 0.307 | 0.299 | 0.309 | 0.302 | 0.281 | 0.309 | 0.293 | 0.267 | 0.249 | 0.266 |
| UNO | 0.322 | 0.343 | 0.330 | 0.328 | 0.325 | 0.312 | 0.326 | 0.331 | 0.349 | 0.322 | 0.319 | 0.331 | 0.330 | 0.330 | 0.320 | 0.320 | 0.315 | 0.324 | 0.333 | 0.340 | 0.333 | 0.340 | 0.327 | 0.306 | 0.325 | 0.299 | 0.299 | 0.297 | 0.266 |
| Emu2 | 0.310 | 0.316 | 0.307 | 0.303 | 0.309 | 0.285 | 0.308 | 0.310 | 0.316 | 0.316 | 0.297 | 0.310 | 0.310 | 0.311 | 0.319 | 0.300 | 0.296 | 0.299 | 0.305 | 0.316 | 0.308 | 0.315 | 0.318 | 0.304 | 0.326 | 0.309 | 0.292 | 0.276 | 0.266 |
| RealCustom++ | 0.327 | 0.339 | 0.332 | 0.336 | 0.333 | 0.337 | 0.352 | 0.326 | 0.336 | 0.323 | 0.338 | 0.335 | 0.312 | 0.338 | 0.335 | 0.322 | 0.323 | 0.326 | 0.340 | 0.338 | 0.335 | 0.344 | 0.346 | 0.328 | 0.343 | 0.325 | 0.312 | 0.315 | 0.309 |
| OmniGen | 0.318 | 0.318 | 0.312 | 0.333 | 0.312 | 0.307 | 0.316 | 0.313 | 0.357 | 0.316 | 0.314 | 0.316 | 0.310 | 0.322 | 0.330 | 0.316 | 0.325 | 0.317 | 0.314 | 0.327 | 0.328 | 0.340 | 0.311 | 0.331 | 0.317 | 0.324 | 0.317 | 0.315 | |
| Custom Diffusion | 0.331 | 0.330 | 0.316 | 0.322 | 0.321 | 0.313 | 0.323 | 0.321 | 0.337 | 0.327 | 0.313 | 0.316 | 0.316 | 0.325 | 0.325 | 0.322 | 0.322 | 0.326 | 0.325 | 0.329 | 0.326 | 0.332 | 0.336 | 0.317 | 0.330 | 0.310 | 0.312 | 0.314 | 0.315 |
| DreamBooth | 0.315 | 0.330 | 0.315 | 0.321 | 0.320 | 0.314 | 0.319 | 0.325 | 0.299 | 0.316 | 0.312 | 0.306 | 0.318 | 0.320 | 0.321 | 0.320 | 0.313 | 0.320 | 0.321 | 0.327 | 0.325 | 0.330 | 0.323 | 0.303 | 0.320 | 0.310 | 0.319 | 0.319 | 0.316 |
| Textual Inversion | 0.310 | 0.307 | 0.294 | 0.308 | 0.304 | 0.275 | 0.293 | 0.306 | 0.323 | 0.312 | 0.296 | 0.307 | 0.300 | 0.292 | 0.309 | 0.292 | 0.302 | 0.294 | 0.299 | 0.309 | 0.305 | 0.307 | 0.320 | 0.301 | 0.316 | 0.283 | 0.276 | 0.290 | 0.286 |
| λ-Eclipse | 0.280 | 0.302 | 0.286 | 0.296 | 0.278 | 0.299 | 0.297 | 0.312 | 0.299 | 0.303 | 0.310 | 0.305 | 0.310 | 0.305 | 0.284 | 0.289 | 0.289 | 0.292 | 0.301 | 0.290 | 0.297 | 0.301 | 0.294 | 0.297 | 0.296 | 0.272 | 0.274 | 0.248 | |
| HiPer | 0.313 | 0.335 | 0.316 | 0.316 | 0.316 | 0.304 | 0.316 | 0.316 | 0.308 | 0.312 | 0.311 | 0.307 | 0.311 | 0.319 | 0.319 | 0.313 | 0.317 | 0.301 | 0.326 | 0.331 | 0.333 | 0.324 | 0.324 | 0.313 | 0.320 | 0.305 | 0.305 | 0.312 | 0.302 |
| NeTI | 0.308 | 0.314 | 0.311 | 0.314 | 0.314 | 0.276 | 0.304 | 0.312 | 0.302 | 0.307 | 0.302 | 0.310 | 0.310 | 0.302 | 0.312 | 0.295 | 0.307 | 0.299 | 0.307 | 0.315 | 0.315 | 0.312 | 0.320 | 0.320 | 0.309 | 0.311 | 0.300 | 0.281 | 0.282 | 0.266 |

**Image Quality**

| Method | VE | MI | PUF | FB | MS | BO | FUR | HA | AM | BU | DP | IN | ST | DN | PL | JE | BS | AR | CL | SE | SBA | TO | MA | RE | BI | FI | HF | FA | AC |
|---|---|---|---|---|---|---|---|---|---|---|---|---|---|---|---|---|---|---|---|---|---|---|---|---|---|---|---|---|---|
| BLIP-Diffusion | 0.253 | 0.203 | 0.213 | 0.222 | 0.175 | 0.168 | 0.189 | 0.190 | 0.274 | 0.235 | 0.196 | 0.256 | 0.195 | 0.204 | 0.236 | 0.206 | 0.205 | 0.235 | 0.216 | 0.214 | 0.227 | 0.243 | 0.263 | 0.246 | 0.265 | 0.263 | 0.234 | 0.229 | 0.233 |
| IP-Adapter | 0.296 | 0.248 | 0.241 | 0.252 | 0.218 | 0.224 | 0.239 | 0.242 | 0.305 | 0.276 | 0.247 | 0.297 | 0.242 | 0.305 | 0.255 | 0.231 | 0.245 | 0.275 | 0.255 | 0.266 | 0.259 | 0.284 | 0.305 | 0.266 | 0.299 | 0.294 | 0.293 | 0.306 | |
| MS-Diffusion | 0.320 | 0.279 | 0.282 | 0.290 | 0.260 | 0.269 | 0.282 | 0.279 | 0.310 | 0.306 | 0.279 | 0.305 | 0.271 | 0.287 | 0.296 | 0.265 | 0.275 | 0.301 | 0.288 | 0.283 | 0.294 | 0.313 | 0.322 | 0.288 | 0.308 | 0.309 | 0.303 | 0.311 | 0.309 |
| OminiControl | 0.297 | 0.277 | 0.280 | 0.283 | 0.264 | 0.286 | 0.273 | 0.274 | 0.313 | 0.264 | 0.287 | 0.273 | 0.282 | 0.289 | 0.277 | 0.274 | 0.289 | 0.284 | 0.277 | 0.290 | 0.303 | 0.314 | 0.283 | 0.304 | 0.292 | 0.292 | 0.298 | 0.300 | |
| SSR-Encoder | 0.275 | 0.226 | 0.239 | 0.239 | 0.202 | 0.204 | 0.216 | 0.225 | 0.290 | 0.264 | 0.223 | 0.263 | 0.221 | 0.230 | 0.255 | 0.232 | 0.225 | 0.250 | 0.243 | 0.236 | 0.242 | 0.259 | 0.286 | 0.262 | 0.281 | 0.282 | 0.261 | 0.254 | 0.264 |
| UNO | 0.295 | 0.266 | 0.277 | 0.272 | 0.248 | 0.256 | 0.252 | 0.260 | 0.308 | 0.275 | 0.266 | 0.289 | 0.260 | 0.270 | 0.280 | 0.254 | 0.262 | 0.275 | 0.272 | 0.279 | 0.293 | 0.313 | 0.284 | 0.298 | 0.296 | 0.282 | 0.287 | 0.287 | |
| Emu2 | 0.289 | 0.242 | 0.252 | 0.243 | 0.228 | 0.214 | 0.238 | 0.234 | 0.288 | 0.276 | 0.234 | 0.271 | 0.246 | 0.248 | 0.265 | 0.235 | 0.236 | 0.272 | 0.249 | 0.255 | 0.272 | 0.291 | 0.271 | 0.291 | 0.272 | 0.283 | 0.283 | 0.270 | |
| RealCustom++ | 0.325 | 0.279 | 0.294 | 0.291 | 0.277 | 0.275 | 0.277 | 0.275 | 0.307 | 0.308 | 0.285 | 0.313 | 0.275 | 0.287 | 0.294 | 0.264 | 0.272 | 0.297 | 0.299 | 0.290 | 0.302 | 0.305 | 0.321 | 0.301 | 0.319 | 0.309 | 0.311 | 0.313 | 0.311 |
| OmniGen | 0.278 | 0.239 | 0.253 | 0.263 | 0.237 | 0.240 | 0.251 | 0.242 | 0.265 | 0.286 | 0.253 | 0.251 | 0.271 | 0.243 | 0.266 | 0.228 | 0.237 | 0.264 | 0.249 | 0.259 | 0.247 | 0.254 | 0.247 | 0.235 | 0.237 | 0.238 | | | |
| Custom Diffusion | 0.255 | 0.234 | 0.236 | 0.235 | 0.223 | 0.221 | 0.238 | 0.230 | 0.245 | 0.249 | 0.226 | 0.239 | 0.219 | 0.234 | 0.246 | 0.228 | 0.231 | 0.241 | 0.237 | 0.234 | 0.241 | 0.249 | 0.259 | 0.247 | 0.254 | 0.247 | 0.235 | 0.237 | 0.238 |
| DreamBooth | 0.266 | 0.236 | 0.236 | 0.239 | 0.215 | 0.226 | 0.233 | 0.229 | 0.222 | 0.251 | 0.228 | 0.250 | 0.222 | 0.236 | 0.253 | 0.226 | 0.223 | 0.252 | 0.240 | 0.237 | 0.247 | 0.266 | 0.234 | 0.239 | 0.253 | 0.252 | 0.251 | 0.255 | 0.257 |
| Textual Inversion | 0.253 | 0.234 | 0.219 | 0.226 | 0.217 | 0.178 | 0.208 | 0.214 | 0.261 | 0.248 | 0.205 | 0.233 | 0.205 | 0.234 | 0.207 | 0.211 | 0.233 | 0.218 | 0.217 | 0.228 | 0.231 | 0.252 | 0.234 | 0.249 | 0.236 | 0.237 | 0.232 | 0.244 | |
| λ-Eclipse | 0.276 | 0.220 | 0.228 | 0.235 | 0.206 | 0.222 | 0.231 | 0.222 | 0.277 | 0.242 | 0.234 | 0.248 | 0.229 | 0.236 | 0.242 | 0.214 | 0.227 | 0.247 | 0.238 | 0.234 | 0.237 | 0.247 | 0.266 | 0.245 | 0.266 | 0.267 | 0.263 | 0.256 | 0.245 |
| HiPer | 0.269 | 0.238 | 0.242 | 0.237 | 0.218 | 0.211 | 0.232 | 0.222 | 0.287 | 0.253 | 0.229 | 0.256 | 0.219 | 0.230 | 0.253 | 0.225 | 0.224 | 0.239 | 0.243 | 0.242 | 0.251 | 0.255 | 0.277 | 0.264 | 0.271 | 0.267 | 0.257 | 0.255 | 0.254 |
| NeTI | 0.262 | 0.224 | 0.234 | 0.233 | 0.213 | 0.186 | 0.215 | 0.218 | 0.249 | 0.248 | 0.211 | 0.246 | 0.214 | 0.221 | 0.247 | 0.214 | 0.224 | 0.246 | 0.228 | 0.234 | 0.241 | 0.245 | 0.266 | 0.250 | 0.257 | 0.260 | 0.243 | 0.240 | 0.250 |

**MLLM Projects** *(1) Qwen2.5-VL-7B:* https://huggingface.co/Qwen/Qwen2.5-VL-7B-Instruct *(2) Implementation Framework:* https://github.com/hiyouga/LLaMA-Factory

**Why we use *Kendall's τ value* and *Spearman correlation coefficient value*:** The scenario is as follows: We have multiple metrics scoring the same dataset. These metrics may have different value ranges. The ground truth scores are provided by human annotators. We want to measure the correlation between each metric and the human scores. The following are some commonly used correlation evaluation metrics:

Table 14: We evaluated the performance of various methods on DSH-Bench dataset, specifically analyzing their effectiveness across **the third-level categories (under non-photorealistic)**.

| Method | | | | | | | | | | | | | | | Subject Preservation | | | | | | | | | | | | | | |
|---|---|---|---|---|---|---|---|---|---|---|---|---|---|---|---|---|---|---|---|---|---|---|---|---|---|---|---|---|---|
| | VE | MI | PUF | FB | MS | BO | FUR | HA | AM | BU | DP | IN | ST | DN | PL | JE | BS | AR | CL | SE | SBA | TO | MA | RE | BI | FI | HF | FA | AC |
| BLIP-Diffusion | 0.227 | 0.170 | 0.175 | 0.208 | 0.146 | 0.153 | 0.210 | 0.180 | 0.133 | 0.185 | 0.139 | 0.175 | 0.108 | 0.208 | 0.183 | 0.171 | 0.181 | 0.125 | 0.188 | 0.375 | 0.221 | 0.177 | 0.219 | 0.247 | 0.192 | 0.183 | 0.180 | 0.175 | 0.194 |
| IP-Adapter | 0.240 | 0.248 | 0.175 | 0.283 | 0.212 | 0.281 | 0.178 | 0.210 | 0.208 | 0.223 | 0.164 | 0.188 | 0.192 | 0.248 | 0.183 | 0.217 | 0.227 | 0.175 | 0.250 | 0.683 | 0.243 | 0.144 | 0.250 | 0.233 | 0.245 | 0.208 | 0.177 | 0.200 | 0.201 |
| MS-Diffusion | 0.362 | 0.410 | 0.387 | 0.375 | 0.396 | 0.361 | 0.363 | 0.345 | 0.342 | 0.292 | 0.272 | 0.292 | 0.233 | 0.438 | 0.250 | 0.310 | 0.367 | 0.438 | 0.742 | 0.426 | 0.235 | 0.325 | 0.342 | 0.363 | 0.300 | 0.302 | 0.233 | 0.300 | |
| OminiControl | 0.290 | 0.370 | 0.367 | 0.333 | 0.377 | 0.275 | 0.181 | 0.252 | 0.292 | 0.265 | 0.269 | 0.213 | 0.237 | 0.342 | 0.075 | 0.254 | 0.362 | 0.217 | 0.300 | 0.633 | 0.342 | 0.217 | 0.273 | 0.261 | 0.190 | 0.283 | 0.158 | 0.092 | 0.202 |
| SSR-Encoder | 0.187 | 0.168 | 0.221 | 0.260 | 0.183 | 0.219 | 0.179 | 0.198 | 0.208 | 0.201 | 0.103 | 0.079 | 0.146 | 0.173 | 0.179 | 0.133 | 0.154 | 0.175 | 0.170 | 0.667 | 0.260 | 0.131 | 0.219 | 0.156 | 0.177 | 0.208 | 0.145 | 0.208 | 0.176 |
| UNO | 0.440 | 0.448 | 0.537 | 0.365 | 0.467 | 0.428 | 0.415 | 0.399 | 0.433 | 0.377 | 0.367 | 0.308 | 0.342 | 0.471 | 0.317 | 0.462 | 0.481 | 0.458 | 0.530 | 0.675 | 0.438 | 0.335 | 0.350 | 0.431 | 0.358 | 0.375 | 0.315 | 0.208 | 0.338 |
| Emu2 | 0.350 | 0.310 | 0.337 | 0.315 | 0.412 | 0.303 | 0.303 | 0.257 | 0.192 | 0.267 | 0.186 | 0.212 | 0.379 | 0.448 | 0.100 | 0.333 | 0.333 | 0.158 | 0.305 | 0.433 | 0.350 | 0.187 | 0.294 | 0.314 | 0.280 | 0.375 | 0.251 | 0.250 | 0.288 |
| RealCustom++ | 0.435 | 0.425 | 0.763 | 0.431 | 0.508 | 0.472 | 0.436 | 0.380 | 0.525 | 0.300 | 0.331 | 0.404 | 0.208 | 0.469 | 0.337 | 0.646 | 0.398 | 0.433 | 0.317 | 0.450 | 0.493 | 0.342 | 0.419 | 0.406 | 0.392 | 0.367 | 0.265 | 0.179 | 0.271 |
| OmniGen | 0.235 | 0.130 | 0.108 | 0.246 | 0.348 | 0.231 | 0.232 | 0.180 | 0.292 | 0.152 | 0.094 | 0.200 | 0.158 | 0.175 | 0.067 | 0.158 | 0.256 | 0.100 | 0.243 | 0.375 | 0.169 | 0.002 | 0.042 | 0.017 | 0.055 | 0.092 | 0.025 | 0.000 | 0.056 |
| Custom Diffusion | 0.071 | 0.072 | 0.067 | 0.048 | 0.073 | 0.094 | 0.017 | 0.052 | 0.033 | 0.154 | 0.019 | 0.037 | 0.050 | 0.077 | 0.050 | 0.029 | 0.038 | 0.000 | 0.047 | 0.150 | 0.042 | 0.017 | 0.055 | 0.092 | 0.025 | 0.000 | 0.056 | | |
| DreamBooth | 0.171 | 0.245 | 0.179 | 0.215 | 0.181 | 0.211 | 0.111 | 0.170 | 0.233 | 0.208 | 0.089 | 0.096 | 0.163 | 0.231 | 0.050 | 0.196 | 0.154 | 0.050 | 0.172 | 0.308 | 0.112 | 0.040 | 0.120 | 0.089 | 0.118 | 0.233 | 0.065 | 0.017 | 0.087 |
| Textual Inversion | 0.144 | 0.090 | 0.108 | 0.090 | 0.104 | 0.067 | 0.033 | 0.052 | 0.075 | 0.164 | 0.033 | 0.083 | 0.063 | 0.054 | 0.054 | 0.058 | 0.083 | 0.058 | 0.058 | 0.267 | 0.040 | 0.040 | 0.112 | 0.097 | 0.122 | 0.058 | 0.092 | 0.050 | 0.122 |
| λ-Eclipse | 0.279 | 0.237 | 0.225 | 0.317 | 0.244 | 0.261 | 0.194 | 0.218 | 0.250 | 0.226 | 0.214 | 0.192 | 0.092 | 0.235 | 0.221 | 0.192 | 0.219 | 0.450 | 0.227 | 0.517 | 0.250 | 0.196 | 0.270 | 0.294 | 0.232 | 0.267 | 0.215 | 0.196 | 0.234 |
| HiPer | 0.162 | 0.167 | 0.163 | 0.192 | 0.142 | 0.142 | 0.057 | 0.114 | 0.108 | 0.162 | 0.053 | 0.079 | 0.075 | 0.156 | 0.054 | 0.067 | 0.133 | 0.200 | 0.135 | 0.333 | 0.126 | 0.021 | 0.109 | 0.031 | 0.107 | 0.183 | 0.061 | 0.046 | 0.108 |
| NeTI | 0.181 | 0.200 | 0.237 | 0.146 | 0.194 | 0.164 | 0.189 | 0.150 | 0.275 | 0.181 | 0.142 | 0.154 | 0.117 | 0.210 | 0.146 | 0.125 | 0.148 | 0.317 | 0.093 | 0.170 | 0.135 | 0.137 | 0.210 | 0.231 | 0.182 | 0.150 | 0.158 | 0.175 | 0.151 |

| Method | | | | | | | | | | | | | | | Prompt Following | | | | | | | | | | | | | | |
|---|---|---|---|---|---|---|---|---|---|---|---|---|---|---|---|---|---|---|---|---|---|---|---|---|---|---|---|---|---|
| | VE | MI | PUF | FB | MS | BO | FUR | HA | AM | BU | DP | IN | ST | DN | PL | JE | BS | AR | CL | SE | SBA | TO | MA | RE | BI | FI | HF | FA | AC |
| BLIP-Diffusion | 0.290 | 0.287 | 0.291 | 0.293 | 0.291 | 0.279 | 0.275 | 0.284 | 0.281 | 0.280 | 0.292 | 0.277 | 0.278 | 0.296 | 0.259 | 0.282 | 0.287 | 0.259 | 0.286 | 0.285 | 0.292 | 0.277 | 0.283 | 0.291 | 0.274 | 0.289 | 0.256 | 0.222 | 0.272 |
| IP-Adapter | 0.317 | 0.336 | 0.316 | 0.321 | 0.335 | 0.304 | 0.307 | 0.319 | 0.326 | 0.304 | 0.313 | 0.320 | 0.321 | 0.332 | 0.267 | 0.320 | 0.331 | 0.317 | 0.311 | 0.309 | 0.314 | 0.321 | 0.344 | 0.332 | 0.333 | 0.334 | 0.328 | 0.308 | 0.327 |
| MS-Diffusion | 0.337 | 0.355 | 0.334 | 0.339 | 0.354 | 0.326 | 0.335 | 0.340 | 0.332 | 0.326 | 0.332 | 0.329 | 0.328 | 0.348 | 0.310 | 0.341 | 0.344 | 0.322 | 0.346 | 0.342 | 0.344 | 0.321 | 0.344 | 0.332 | 0.333 | 0.334 | 0.328 | 0.324 | 0.338 |
| OminiControl | 0.330 | 0.348 | 0.341 | 0.335 | 0.341 | 0.330 | 0.324 | 0.334 | 0.330 | 0.344 | 0.321 | 0.329 | 0.339 | 0.318 | 0.333 | 0.294 | 0.333 | 0.316 | 0.330 | 0.300 | 0.339 | 0.317 | 0.327 | 0.309 | 0.308 | 0.297 | 0.317 | | |
| SSR-Encoder | 0.289 | 0.317 | 0.287 | 0.304 | 0.305 | 0.287 | 0.291 | 0.305 | 0.301 | 0.285 | 0.293 | 0.302 | 0.296 | 0.305 | 0.270 | 0.292 | 0.308 | 0.291 | 0.304 | 0.298 | 0.299 | 0.293 | 0.300 | 0.303 | 0.302 | 0.298 | 0.288 | 0.251 | 0.294 |
| UNO | 0.316 | 0.343 | 0.314 | 0.337 | 0.334 | 0.305 | 0.321 | 0.324 | 0.314 | 0.309 | 0.313 | 0.327 | 0.302 | 0.338 | 0.289 | 0.313 | 0.333 | 0.294 | 0.333 | 0.316 | 0.330 | 0.300 | 0.339 | 0.317 | 0.327 | 0.309 | 0.308 | 0.297 | 0.317 |
| Emu2 | 0.309 | 0.337 | 0.303 | 0.311 | 0.326 | 0.303 | 0.293 | 0.305 | 0.313 | 0.298 | 0.307 | 0.317 | 0.301 | 0.318 | 0.291 | 0.309 | 0.304 | 0.276 | 0.308 | 0.323 | 0.312 | 0.273 | 0.297 | 0.312 | 0.304 | 0.297 | 0.292 | 0.243 | 0.279 |
| RealCustom++ | 0.319 | 0.351 | 0.326 | 0.328 | 0.340 | 0.328 | 0.331 | 0.331 | 0.317 | 0.326 | 0.309 | 0.332 | 0.320 | 0.350 | 0.301 | 0.325 | 0.329 | 0.298 | 0.346 | 0.336 | 0.338 | 0.317 | 0.338 | 0.319 | 0.339 | 0.333 | 0.330 | 0.321 | 0.338 |
| OmniGen | 0.322 | 0.321 | 0.313 | 0.328 | 0.341 | 0.318 | 0.308 | 0.305 | 0.332 | 0.317 | 0.301 | 0.322 | 0.309 | 0.321 | 0.299 | 0.298 | 0.338 | 0.305 | 0.326 | 0.336 | 0.314 | 0.323 | 0.329 | 0.327 | 0.309 | 0.333 | 0.326 | 0.315 | 0.333 |
| Custom Diffusion | 0.315 | 0.336 | 0.330 | 0.326 | 0.326 | 0.313 | 0.324 | 0.323 | 0.322 | 0.323 | 0.308 | 0.302 | 0.311 | 0.326 | 0.314 | 0.324 | 0.323 | 0.318 | 0.321 | 0.329 | 0.321 | 0.317 | 0.331 | 0.320 | 0.325 | 0.313 | 0.316 | 0.312 | 0.326 |
| DreamBooth | 0.322 | 0.337 | 0.324 | 0.330 | 0.317 | 0.322 | 0.317 | 0.327 | 0.322 | 0.310 | 0.317 | 0.309 | 0.322 | 0.314 | 0.319 | 0.316 | 0.295 | 0.325 | 0.334 | 0.325 | 0.334 | 0.328 | 0.323 | 0.324 | 0.315 | 0.324 | | | |
| Textual Inversion | 0.290 | 0.304 | 0.287 | 0.293 | 0.303 | 0.287 | 0.291 | 0.293 | 0.302 | 0.301 | 0.281 | 0.288 | 0.291 | 0.296 | 0.272 | 0.286 | 0.304 | 0.266 | 0.301 | 0.314 | 0.282 | 0.282 | 0.299 | 0.300 | 0.298 | 0.292 | 0.290 | 0.289 | 0.294 |
| λ-Eclipse | 0.280 | 0.313 | 0.256 | 0.300 | 0.293 | 0.290 | 0.295 | 0.292 | 0.301 | 0.278 | 0.281 | 0.304 | 0.302 | 0.310 | 0.273 | 0.293 | 0.306 | 0.241 | 0.311 | 0.335 | 0.304 | 0.288 | 0.305 | 0.297 | 0.306 | 0.296 | 0.286 | 0.283 | 0.279 |
| HiPer | 0.321 | 0.339 | 0.317 | 0.334 | 0.325 | 0.317 | 0.324 | 0.320 | 0.327 | 0.324 | 0.311 | 0.317 | 0.320 | 0.325 | 0.318 | 0.324 | 0.319 | 0.220 | 0.323 | 0.336 | 0.312 | 0.333 | 0.336 | 0.310 | 0.339 | 0.319 | 0.313 | 0.319 | |
| NeTI | 0.297 | 0.327 | 0.305 | 0.306 | 0.317 | 0.292 | 0.274 | 0.291 | 0.302 | 0.295 | 0.272 | 0.300 | 0.301 | 0.305 | 0.257 | 0.286 | 0.300 | 0.256 | 0.291 | 0.325 | 0.288 | 0.270 | 0.299 | 0.290 | 0.295 | 0.301 | 0.285 | 0.227 | 0.296 |

| Method | | | | | | | | | | | | | | | Image Quality | | | | | | | | | | | | | | |
|---|---|---|---|---|---|---|---|---|---|---|---|---|---|---|---|---|---|---|---|---|---|---|---|---|---|---|---|---|---|
| | VE | MI | PUF | FB | MS | BO | FUR | HA | AM | BU | DP | IN | ST | DN | PL | JE | BS | AR | CL | SE | SBA | TO | MA | RE | BI | FI | HF | FA | AC |
| BLIP-Diffusion | 0.244 | 0.208 | 0.193 | 0.221 | 0.184 | 0.185 | 0.186 | 0.202 | 0.232 | 0.243 | 0.190 | 0.234 | 0.220 | 0.221 | 0.198 | 0.224 | 0.220 | 0.204 | 0.212 | 0.184 | 0.225 | 0.189 | 0.246 | 0.272 | 0.256 | 0.224 | 0.216 | 0.235 | |
| IP-Adapter | 0.285 | 0.255 | 0.239 | 0.263 | 0.230 | 0.231 | 0.227 | 0.245 | 0.309 | 0.279 | 0.245 | 0.271 | 0.259 | 0.258 | 0.220 | 0.278 | 0.264 | 0.242 | 0.240 | 0.229 | 0.251 | 0.227 | 0.294 | 0.310 | 0.265 | 0.311 | 0.283 | 0.293 | 0.305 |
| MS-Diffusion | 0.304 | 0.287 | 0.268 | 0.296 | 0.269 | 0.273 | 0.269 | 0.276 | 0.308 | 0.304 | 0.267 | 0.286 | 0.288 | 0.294 | 0.267 | 0.299 | 0.298 | 0.267 | 0.284 | 0.283 | 0.289 | 0.255 | 0.307 | 0.321 | 0.296 | 0.323 | 0.293 | 0.307 | 0.311 |
| OminiControl | 0.295 | 0.290 | 0.265 | 0.287 | 0.269 | 0.274 | 0.278 | 0.278 | 0.314 | 0.298 | 0.277 | 0.281 | 0.284 | 0.295 | 0.288 | 0.298 | 0.285 | 0.265 | 0.282 | 0.281 | 0.287 | 0.270 | 0.312 | 0.309 | 0.307 | 0.325 | 0.297 | 0.291 | 0.309 |
| SSR-Encoder | 0.258 | 0.237 | 0.216 | 0.241 | 0.204 | 0.206 | 0.213 | 0.239 | 0.288 | 0.260 | 0.214 | 0.232 | 0.234 | 0.217 | 0.245 | 0.237 | 0.227 | 0.229 | 0.210 | 0.241 | 0.212 | 0.271 | 0.287 | 0.266 | 0.293 | 0.268 | 0.248 | 0.268 | |
| UNO | 0.286 | 0.277 | 0.249 | 0.288 | 0.257 | 0.249 | 0.252 | 0.262 | 0.312 | 0.275 | 0.243 | 0.285 | 0.267 | 0.290 | 0.223 | 0.282 | 0.274 | 0.226 | 0.265 | 0.228 | 0.276 | 0.240 | 0.302 | 0.315 | 0.290 | 0.311 | 0.273 | 0.278 | 0.294 |
| Emu2 | 0.278 | 0.261 | 0.239 | 0.259 | 0.236 | 0.239 | 0.227 | 0.235 | 0.294 | 0.269 | 0.242 | 0.278 | 0.246 | 0.265 | 0.254 | 0.252 | 0.277 | 0.254 | 0.255 | 0.254 | 0.223 | 0.269 | 0.296 | 0.275 | 0.302 | 0.268 | 0.256 | 0.264 | |
| RealCustom++ | 0.304 | 0.300 | 0.283 | 0.291 | 0.260 | 0.280 | 0.277 | 0.274 | 0.316 | 0.314 | 0.258 | 0.304 | 0.290 | 0.304 | 0.276 | 0.285 | 0.290 | 0.259 | 0.311 | 0.331 | 0.313 | 0.321 | 0.305 | 0.319 | 0.330 | | | | |
| OmniGen | 0.280 | 0.251 | 0.236 | 0.262 | 0.261 | 0.262 | 0.231 | 0.232 | 0.291 | 0.274 | 0.239 | 0.282 | 0.248 | 0.259 | 0.243 | 0.256 | 0.265 | 0.253 | 0.256 | 0.268 | 0.253 | 0.252 | 0.281 | 0.298 | 0.251 | 0.292 | 0.276 | 0.288 | 0.292 |
| Custom Diffusion | 0.240 | 0.249 | 0.238 | 0.235 | 0.227 | 0.221 | 0.241 | 0.228 | 0.240 | 0.257 | 0.221 | 0.228 | 0.251 | 0.241 | 0.247 | 0.240 | 0.231 | 0.229 | 0.235 | 0.245 | 0.240 | 0.253 | 0.245 | 0.266 | 0.255 | 0.250 | 0.237 | 0.227 | 0.250 |
| DreamBooth | 0.253 | 0.249 | 0.246 | 0.246 | 0.231 | 0.235 | 0.235 | 0.239 | 0.281 | 0.270 | 0.229 | 0.240 | 0.229 | 0.257 | 0.252 | 0.255 | 0.236 | 0.246 | 0.245 | 0.245 | 0.239 | 0.269 | 0.280 | 0.262 | 0.274 | 0.249 | 0.249 | 0.262 | |
| Textual Inversion | 0.227 | 0.220 | 0.210 | 0.212 | 0.204 | 0.198 | 0.212 | 0.206 | 0.240 | 0.251 | 0.204 | 0.217 | 0.204 | 0.209 | 0.212 | 0.222 | 0.219 | 0.181 | 0.217 | 0.211 | 0.217 | 0.198 | 0.205 | 0.230 | 0.235 | 0.193 | 0.248 | 0.254 | |
| λ-Eclipse | 0.250 | 0.236 | 0.212 | 0.239 | 0.214 | 0.232 | 0.215 | 0.227 | 0.264 | 0.249 | 0.211 | 0.252 | 0.250 | 0.238 | 0.211 | 0.235 | 0.238 | 0.208 | 0.227 | 0.250 | 0.234 | 0.211 | 0.263 | 0.282 | 0.249 | 0.281 | 0.265 | 0.289 | 0.252 | 0.260 | 0.254 |
| HiPer | 0.245 | 0.252 | 0.224 | 0.243 | 0.218 | 0.226 | 0.230 | 0.227 | 0.293 | 0.265 | 0.212 | 0.235 | 0.232 | 0.247 | 0.243 | 0.259 | 0.235 | 0.193 | 0.244 | 0.234 | 0.234 | 0.234 | 0.274 | 0.281 | 0.265 | 0.289 | 0.252 | 0.264 | 0.258 |
| NeTI | 0.244 | 0.244 | 0.227 | 0.227 | 0.214 | 0.202 | 0.186 | 0.211 | 0.271 | 0.250 | 0.196 | 0.228 | 0.220 | 0.232 | 0.194 | 0.226 | 0.221 | 0.208 | 0.218 | 0.218 | 0.211 | 0.183 | 0.244 | 0.281 | 0.238 | 0.277 | 0.232 | 0.227 | 0.254 |

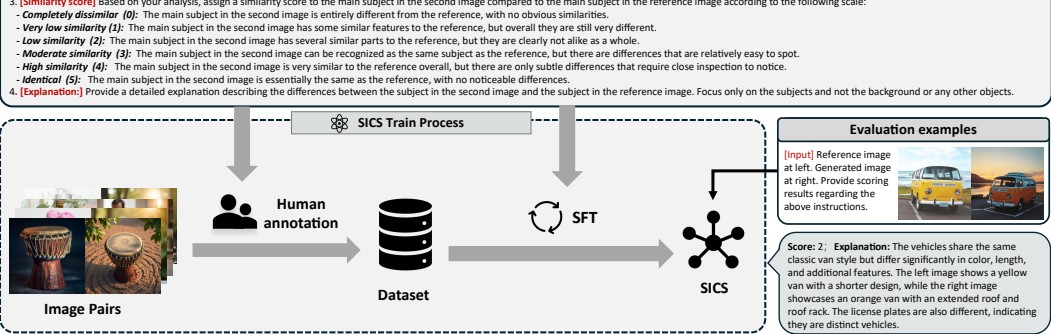

**⚙ Evaluation Instruction**

As an experienced evaluator, you are given two images. The first image is the reference image. Your tasks are as follows:
1. **[Focus only on the main subject of the reference image]** When comparing, consider only the main subject in the first (reference) image. In the second image, look for the corresponding main subject and compare it to the reference. Ignore the background and any other objects or elements, including those interacting with the main subject.
2. **[Criteria for comparison]** Compare the main subject in the second image to the main subject in the reference image based on the following criteria:
   - Shape and structure.     - Color and texture     - Size and proportion     - Distinctive features or markings
3. **[Similarity score]** Based on your analysis, assign a similarity score to the main subject in the second image compared to the main subject in the reference image according to the following scale:
   - **Completely dissimilar (0):** The main subject in the second image is entirely different from the reference, with no obvious similarities.
   - **Very low similarity (1):** The main subject in the second image has some similar features to the reference, but overall they are still very different.
   - **Low similarity (2):** The main subject in the second image has several similar parts to the reference, but they are clearly not alike as a whole.
   - **Moderate similarity (3):** The main subject in the second image can be recognized as the same subject as the reference, but there are differences that are relatively easy to spot.
   - **High similarity (4):** The main subject in the second image is very similar to the reference overall, but there are only subtle differences that require close inspection to notice.
   - **Identical (5):** The main subject in the second image is essentially the same as the reference, with no noticeable differences.
4. **[Explanation:]** Provide a detailed explanation describing the differences between the subject in the second image and the subject in the reference image. Focus only on the subjects and not the background or any other objects.

**⚛ SICS Train Process**

Image Pairs → Human annotation → Dataset → SFT → SICS

**Evaluation examples**

[Input] Reference image at left. Generated image at right. Provide scoring results regarding the above instructions.

Score: 2; Explanation: The vehicles share the same classic van style but differ significantly in color, length, and additional features. The left image shows a yellow van with a shorter design, while the right image showcases an orange van with an extended roof and roof rack. The license plates are also different, indicating they are distinct vehicles.

Figure 12: **The training process of SICS.** We constructed and annotated a dataset specifically tailored for subject consistency determination, and subsequently trained models using this dataset.

1. **Pearson Correlation Coefficient**
   - **Advantages:** Measures linear correlation between two continuous variables; simple to compute.
   - **Disadvantages:** Only suitable for linear relationships; sensitive to outliers; requires interval or ratio data; variables should be on the same scale.
   - **Applicability:** Use if both our metric and human scores are continuous and linearly related. If scales differ, standardize first.

2. **Spearman Rank Correlation Coefficient**
   - **Advantages:** Measures monotonic relationships; does not require linearity; robust to outliers; works with different scales and ordinal data.
   - **Disadvantages:** Only captures monotonic relationships; some information loss due to ranking.

Table 15: We evaluated the performance of various methods on DSH-Bench dataset, specifically analyzing their effectiveness across **prompts with different scenarios**. Subject preservation, prompt following, and image quality are evaluated using SICS, CLIP-T, and HPSv2, respectively.

| Method | Subject Preservation | | | | | |
| | Background Change | Variation in Subject Viewpoint or Size | Interaction with Other Entities | Attribute Change | Style Change | Imagination |
| --- | --- | --- | --- | --- | --- | --- |
| BLIP-Diffusion | 0.204 | 0.207 | 0.201 | 0.182 | 0.189 | 0.195 |
| IP-Adapter | 0.233 | 0.230 | 0.224 | 0.177 | 0.203 | 0.209 |
| MS-Diffusion | 0.361 | 0.359 | 0.337 | 0.266 | 0.294 | 0.308 |
| OminiControl | 0.300 | 0.263 | 0.212 | 0.176 | 0.252 | 0.211 |
| SSR-Encoder | 0.206 | 0.201 | 0.200 | 0.166 | 0.171 | 0.188 |
| UNO | **0.433** | **0.414** | **0.379** | **0.359** | **0.418** | **0.349** |
| Emu2 | 0.393 | 0.316 | 0.315 | 0.326 | 0.224 | 0.239 |
| RealCustom++ | 0.386 | 0.384 | 0.353 | 0.297 | 0.314 | 0.310 |
| OmniGen | 0.238 | 0.167 | 0.159 | 0.125 | 0.155 | 0.133 |
| Custom Diffusion | 0.073 | 0.060 | 0.053 | 0.047 | 0.037 | 0.047 |
| DreamBooth | 0.180 | 0.157 | 0.138 | 0.139 | 0.128 | 0.144 |
| Textual Inversion | 0.121 | 0.102 | 0.104 | 0.109 | 0.074 | 0.098 |
| $\lambda$-Eclipse | 0.262 | 0.257 | 0.249 | 0.246 | 0.244 | 0.230 |
| HiPer | 0.148 | 0.130 | 0.127 | 0.116 | 0.106 | 0.125 |
| NeTI | 0.211 | 0.182 | 0.185 | 0.198 | 0.173 | 0.182 |
| *Aver.* | 0.250 | 0.229 | 0.216 | 0.195 | 0.199 | 0.198 |

| Method | Prompt Following | | | | | |
| | Background Change | Variation in Subject Viewpoint or Size | Interaction with Other Entities | Attribute Change | Style Change | Imagination |
| --- | --- | --- | --- | --- | --- | --- |
| BLIP-Diffusion | 0.297 | 0.275 | 0.264 | 0.285 | 0.272 | 0.271 |
| IP-Adapter | 0.326 | 0.319 | 0.319 | 0.312 | 0.306 | 0.310 |
| MS-Diffusion | 0.342 | **0.339** | **0.341** | 0.324 | **0.338** | 0.341 |
| OminiControl | 0.338 | 0.334 | 0.337 | **0.329** | 0.326 | **0.342** |
| SSR-Encoder | 0.310 | 0.296 | 0.288 | 0.299 | 0.288 | 0.291 |
| UNO | 0.334 | 0.328 | 0.333 | 0.305 | 0.302 | 0.335 |
| Emu2 | 0.308 | 0.305 | 0.297 | 0.295 | 0.313 | 0.308 |
| RealCustom++ | **0.343** | 0.333 | 0.329 | 0.319 | 0.328 | 0.338 |
| OmniGen | 0.327 | 0.325 | 0.328 | 0.311 | 0.315 | 0.327 |
| Custom Diffusion | 0.326 | 0.317 | 0.320 | 0.328 | 0.325 | 0.323 |
| DreamBooth | 0.326 | 0.318 | 0.319 | 0.319 | 0.323 | 0.320 |
| Textual Inversion | 0.303 | 0.299 | 0.300 | 0.296 | 0.298 | 0.296 |
| $\lambda$-Eclipse | 0.302 | 0.292 | 0.289 | 0.285 | 0.286 | 0.299 |
| HiPer | 0.325 | 0.323 | 0.315 | 0.318 | 0.318 | 0.311 |
| NeTI | 0.309 | 0.305 | 0.301 | 0.296 | 0.295 | 0.297 |
| *Aver.* | 0.321 | 0.314 | 0.312 | 0.308 | 0.309 | 0.314 |

| Method | Image Quality | | | | | |
| | Background Change | Variation in Subject Viewpoint or Size | Interaction with Other Entities | Attribute Change | Style Change | Imagination |
| --- | --- | --- | --- | --- | --- | --- |
| BLIP-Diffusion | 0.234 | 0.220 | 0.199 | 0.235 | 0.239 | 0.214 |
| IP-Adapter | 0.269 | 0.263 | 0.258 | 0.272 | 0.276 | 0.259 |
| MS-Diffusion | 0.291 | 0.292 | 0.292 | 0.287 | 0.300 | 0.301 |
| OminiControl | 0.285 | 0.283 | 0.290 | **0.293** | 0.294 | 0.296 |
| SSR-Encoder | 0.256 | 0.246 | 0.231 | 0.256 | 0.256 | 0.238 |
| UNO | 0.282 | 0.281 | 0.283 | 0.268 | 0.275 | 0.276 |
| Emu2 | 0.262 | 0.260 | 0.249 | 0.252 | 0.268 | 0.270 |
| RealCustom++ | **0.300** | **0.298** | **0.295** | 0.284 | **0.301** | **0.307** |
| OmniGen | 0.263 | 0.259 | 0.271 | 0.257 | 0.260 | 0.282 |
| Custom Diffusion | 0.245 | 0.236 | 0.237 | 0.248 | 0.231 | 0.240 |
| DreamBooth | 0.252 | 0.242 | 0.239 | 0.250 | 0.246 | 0.243 |
| Textual Inversion | 0.228 | 0.222 | 0.222 | 0.230 | 0.225 | 0.221 |
| $\lambda$-Eclipse | 0.247 | 0.242 | 0.233 | 0.241 | 0.249 | 0.242 |
| HiPer | 0.250 | 0.247 | 0.241 | 0.255 | 0.247 | 0.240 |
| NeTI | 0.239 | 0.231 | 0.230 | 0.240 | 0.236 | 0.230 |
| *Aver.* | 0.260 | 0.255 | 0.251 | 0.258 | 0.260 | 0.257 |

- **Applicability:** Highly suitable for our scenario, especially when metrics have different scales or are not linearly related.

3. **Kendall Rank Correlation Coefficient**

Table 16: We evaluated the performance of various methods on DSH-Bench dataset, specifically analyzing their effectiveness across **images with different difficulty levels**. Subject preservation, prompt following, and image quality are evaluated using SICS, CLIP-T, and HPSv2, respectively.

| Method | Subject Preservation | | | Prompt Following | | | Image Quality | | |
|---|---|---|---|---|---|---|---|---|---|
| | Easy | Medium | Hard | Easy | Medium | Hard | Easy | Medium | Hard |
| BLIP-Diffusion | 0.221 | 0.209 | 0.190 | 0.284 | 0.278 | 0.273 | 0.198 | 0.227 | 0.232 |
| IP-Adapter | 0.266 | 0.233 | 0.206 | 0.316 | 0.315 | 0.316 | 0.236 | 0.270 | 0.278 |
| MS-Diffusion | 0.410 | 0.362 | 0.312 | **0.340** | **0.339** | **0.335** | 0.278 | 0.297 | 0.299 |
| OminiControl | 0.294 | 0.256 | 0.242 | 0.337 | 0.336 | 0.331 | 0.278 | 0.292 | 0.294 |
| SSR-Encoder | 0.234 | 0.212 | 0.174 | 0.299 | 0.295 | 0.294 | 0.220 | 0.251 | 0.257 |
| UNO | **0.469** | **0.405** | **0.383** | 0.326 | 0.325 | 0.319 | 0.261 | 0.281 | 0.283 |
| Emu2 | 0.349 | 0.346 | 0.332 | 0.308 | 0.306 | 0.301 | 0.239 | 0.263 | 0.268 |
| RealCustom++ | 0.448 | 0.379 | 0.331 | 0.334 | 0.333 | 0.329 | **0.281** | **0.300** | **0.303** |
| OmniGen | 0.224 | 0.188 | 0.170 | 0.321 | 0.324 | 0.321 | 0.249 | 0.267 | 0.272 |
| Custom Diffusion | 0.067 | 0.061 | 0.060 | 0.323 | 0.324 | 0.322 | 0.234 | 0.241 | 0.241 |
| DreamBooth | 0.184 | 0.163 | 0.139 | 0.323 | 0.322 | 0.319 | 0.232 | 0.248 | 0.249 |
| Textual Inversion | 0.092 | 0.112 | 0.115 | 0.295 | 0.300 | 0.299 | 0.206 | 0.226 | 0.233 |
| $\lambda$-Eclipse | 0.286 | 0.260 | 0.235 | 0.302 | 0.293 | 0.286 | 0.228 | 0.244 | 0.248 |
| HiPer | 0.139 | 0.145 | 0.122 | 0.323 | 0.319 | 0.315 | 0.230 | 0.251 | 0.251 |
| NeTI | 0.203 | 0.189 | 0.190 | 0.303 | 0.302 | 0.298 | 0.214 | 0.237 | 0.242 |
| *Aver.* | 0.259 | 0.235 | 0.213 | 0.316 | 0.314 | 0.311 | 0.239 | 0.260 | 0.263 |

Table 17: **Experiment hyperparameters on DSH-Bench.** LR: learning rate, Steps: training steps, GS: guidance scale

| Method | T2I Model | Batch Size | LR | Train Steps | GS | Infer Steps | Additional parameter |
|---|---|---|---|---|---|---|---|
| BLIP-Diffusion | SD v1.5 | N/A | N/A | N/A | 7.5 | 25 | N/A |
| IP-Adapter | SDXL | N/A | N/A | N/A | 7.5 | 30 | ip_adapter_scale: 0.5 |
| MS-Diffusion | SDXL | N/A | N/A | N/A | 7.5 | 30 | scale: 0.6 |
| OminiControl | FLUX.1-schnell | N/A | N/A | N/A | 3.5 | 10 | condition_scale: 1 |
| SSR-Encoder | SD v1.5 | N/A | N/A | N/A | 7.5 | 30 | $\lambda : 0.5$ |
| UNO | FLUX.1-dev | N/A | N/A | N/A | 4 | 25 | N/A |
| Emu2 | SDXL | N/A | N/A | N/A | 3.0 | 50 | N/A |
| RealCustom++ | SDXL | N/A | N/A | N/A | 7.5 | 25 | N/A |
| OmniGen | SDXL | N/A | N/A | N/A | 2.5 | 50 | img_guidance_scale: 1.8 |
| $\lambda$-Eclipse | SDXL | N/A | N/A | N/A | 7.5 | 50 | N/A |
| Textual Inversion | SD v1.5 | 4 | 5e-4 | 3000 | 7.5 | 50 | N/A |
| DreamBooth | SD v1.5 | 1 | 2.5e-6 | 250 | 7.5 | 50 | N/A |
| Custom Diffusion | SD v1.4 | 2 | 1e-5 | 250 | 6.0 | 100 | N/A |
| HiPer | SD v1.4 | 1 | 5e-3 | 1500 | 7.5 | 50 | N/A |
| NeTI | SD v1.4 | 2 | 1e-3 | 250 | 7.5 | 50 | N/A |

- - **Advantages:** Also measures monotonic relationships; robust to outliers; suitable for rank/ordinal data.
  - **Disadvantages:** More computationally intensive than Spearman; only captures monotonic relationships.
  - **Applicability:** Also highly suitable, especially for smaller datasets or when we want a more robust rank-based measure.

4. **Krippendorff's Alpha**
   - **Advantages:** Handles multiple raters and various data types (nominal, ordinal, interval, ratio); can handle missing data.
   - **Disadvantages:** Mainly used for inter-rater reliability, not for correlation; does not indicate the direction of association; computationally complex.
   - **Applicability:** Not suitable for our scenario, as it is designed to measure agreement among multiple raters.

Consequently, we choose *Kendall's $\tau$ value* and *Spearman correlation coefficient value*.

### E.3  DETAILS OF PROMPT GENERATION

We present below a comparative analysis of prompts generated by three state-of-the-art vision-language models: GPT-4o, Gemini 2.5 Pro, and Claude. The specific instruction used to generate the

prompts can be found in Figure 5. A qualitative evaluation of these prompts reveals no significant or discernible differences in their content or structure. Given the comparable performance across the models, we selected GPT-4o for all prompt generation tasks in this study. Moreover, the results generated by GPT-4o include slightly more complex and diverse descriptions of the environment, which makes the model's generation and application more intricate and challenging. This choice also aligns with the methodology established in the Dreambench++, ensuring methodological consistency with prior work. The example is as follows:

**Background Change**

1. **Gemini 2.5 Pro:** A single african drum resting on the red earth of the African savanna at sunset.
2. **Claude:** An african drum standing on a sandy beach with waves in the background
3. **GPT-4o:** A single african drum placed on a sandy beach with gentle waves in the background under a clear sky.
4. **Gemini 2.5 Pro:** An african drum placed on a clean, white studio background.
5. **Claude:** An african drum placed in a lush green rainforest clearing.
6. **GPT-4o:** An african drum positioned on a wooden table inside a cozy rustic cabin with warm lighting and wooden walls.

**Variation in Subject Viewpoint or Size**

1. **Gemini 2.5 Pro:** A low-angle shot of an african drum sitting on a wooden stage, illuminated by a single, warm spotlight from above, with dust motes dancing in the light.
2. **Claude:** A low-angle shot of an african drum on a stage, with spotlights creating dramatic shadows.
3. **GPT-4o:** An african drum viewed from a high-angle perspective, placed on a grassy hilltop overlooking a distant mountain range with scattered wildflowers around it.
4. **Gemini 2.5 Pro:** Bird's-eye view of an african drum next to a crackling campfire at dusk, with the flickering firelight casting long, dancing shadows on the ground.
5. **Claude:** A bird's-eye view of an african drum in the center of a circular stone pattern in a park.
6. **GPT-4o:** A close-up shot of an african drum resting on a cobblestone pathway in a small village, with faint rays of sunlight filtering through nearby trees and a soft shadow cast beneath it.

**Interaction with Other Entities**

1. **Gemini 2.5 Pro:** A close-up shot of a musician's hands, blurred with motion, striking the surface of an african drum, with beads of sweat flying off the drumhead.
2. **Claude:** An african drum being played by multiple hands in a crowded marketplace, with motion blur on the hands.
3. **GPT-4o:** An african drum being played by a young boy surrounded by a group of people dancing in a circle under the moonlight at an outdoor festival, motion blurs capturing the dynamic movements of the dancers.
4. **Gemini 2.5 Pro:** A group of children in a circle, their joyful faces illuminated by the sun, all reaching out to play a large african drum together.
5. **Claude:** An african drum partially submerged in a clear lake, with its reflection visible on the water's surface
6. **GPT-4o:** An african drum partially obscured by tall reeds in a dense wetland, with a heron standing nearby and its reflection visible in the water below.

**Attribute Change**

1. **Gemini 2.5 Pro:** An african drum carved entirely from shimmering, translucent crystal, with glowing runes etched onto its surface.

2. **Claude:** An african drum made entirely of clear, shimmering crystal.

3. **GPT-4o:** An african drum painted entirely in vibrant rainbow colors and decorated with intricate tribal patterns.

4. **Gemini 2.5 Pro:** A futuristic, metallic african drum with a holographic drumhead that displays vibrant sound visualizations with every beat.

5. **Claude:** An african drum with a top made of stained glass, creating colorful patterns when light shines through.

6. **GPT-4o:** An african drum crafted from polished metal instead of wood, with a reflective surface that mirrors its surroundings.

**Style Change**

1. **Gemini 2.5 Pro:** A cubist painting of an african drum, deconstructed into geometric shapes and shown from multiple perspectives simultaneously.

2. **Claude:** An african drum rendered in the style of Vincent van Gogh's 'Starry Night', with swirling brushstrokes and vibrant colors.

3. **GPT-4o:** An african drum depicted in the Cubist art style, with fragmented geometric shapes and abstract distortions.

4. **Gemini 2.5 Pro:** An african drum depicted in the style of Japanese Ukiyo-e woodblock print, with flat colors, bold outlines, and a serene composition.

5. **Claude:** A cubist interpretation of an african drum, broken down into geometric shapes and multiple perspectives.

6. **GPT-4o:** An african drum illustrated in watercolor art, blending soft pastel tones with fluid brushstrokes, surrounded by splashes of color in the background.

**Imagination**

1. **Gemini 2.5 Pro:** A fleet of miniature african drums floating through a cosmic nebula, propelled by rhythmic sound waves that ripple through the stardust.

2. **Claude:** An african drum as a spaceship, with tiny alien creatures using it to explore the galaxy.

3. **GPT-4o:** An african drum floating in mid-air, surrounded by glowing orbs of light that pulse rhythmically as if responding to the drum's silent beat.

4. **Gemini 2.5 Pro:** In an enchanted forest, an ancient african drum is covered in moss and glowing mushrooms; when played, it causes the surrounding trees to grow and bloom instantly.

5. **Claude:** An african drum transformed into a living creature, with eyes and limbs, dancing in a magical forest.

6. **GPT-4o:** An african drum transformed into a magical portal, with swirling galaxies and stars emerging from its open top.

## E.4   Details of Human Annotation

### E.4.1   Annotation Verification Guidelines For Subject Difficulty Level Classification

**Task Objective**

- Verify the difficulty label assigned by the model (Easy / Medium / Hard) for the "subject detail preservation" when an image is used as a reference image.
- Subject: the primary, most prominent object or semantic entity in the image

**Label Definitions (must align strictly)**

- **Easy**: Low surface complexity, homogeneous texture, near-uniform color/material; virtually no high-frequency details. Examples: smooth, solid-colored ceramic mug; smooth sphere; plain object without text/markings.
- **Medium**: Contains discernible high-frequency features while maintaining globally simple, coherent structure; local details are present but not overwhelmingly dense. Examples: cylindrical container with readable text/logo; simple shapes with a few clear markings/scales/brand labels.
- **Hard**: Non-uniform texture distribution with multi-scale geometric/details; dense, fine-grained features across regions that materially affect the subject's appearance. Examples: book cover with fine calligraphy and intricate patterns; woven/engraved/ornamented materials; many lines of small text with layout hierarchy.

**Step-by-Step Decision Process**

1. **Identify the subject**
   - Select the single most salient object (by size, focus sharpness, centrality, semantic importance).
   - Multiple visible parts forming one entity (e.g., front and back of one book) count as one subject.

2. **Assess scale and detail density**
   - High-frequency details: small text, granular textures, dense lines/patterns, fine edges, repetitive microstructures.
   - Multi-scale: presence of large contours plus mid/small-scale text/patterns that meaningfully define appearance.

3. **Evaluate texture uniformity and surface complexity**
   - Uniform materials (solid/near-solid color, smooth, gradual highlights) → tends to *Easy*.
   - Simple shape with a few clear elements (e.g., a single line of text, logo) → tends to *Medium*.
   - Complex surface patterns with non-uniform textures across regions, multiple areas with distinct details → often *Hard*.

4. **Evaluate readable elements**
   - More numerous/smaller/denser text, more font variations, and deeper layout hierarchy increase difficulty.
   - A single large word or single big logo alone does not imply *Hard*; typically *Medium*.

5. **Shape and geometry**
   - Basic primitives (sphere/cylinder/cube) without details → *Easy*.
   - Basic primitives with a few marks/engraved lines → *Medium*.
   - Irregular shapes, folds/pleats, filigree, cutouts, elaborate ornaments → often *Hard*, depending on detail density.

6. **Lighting and material cues (auxiliary only)**
   - Specular highlights or reflections alone do not equal high-frequency detail. If the surface lacks true microstructure/patterns, it can still be *Easy*.
   - Genuine material textures (wood grain, fabric weave, leather grain, brushed metal) may raise difficulty to *Medium/Hard* based on density and multi-scale complexity.

7. **Context and occlusion (consider only if they affect visible details)**
   - Complex backgrounds do not increase subject difficulty unless inseparable from or part of the subject surface.

**Decision Boundaries and Edge Cases**

- *Easy vs. Medium*: Are there local details that must be faithfully reproduced to recognize the subject (e.g., small text, scales, granular texture)? If present and more than minimal, lean *Medium*.

- *Medium vs. Hard*: Are there multi-scale, regionally distributed complex details (e.g., title + subtitle + fine print over a textured background; or patterns + filigree + material grain) that jointly define appearance? If yes, lean *Hard*.
- **Text cases**:
  - Single large brand word: *Medium*.
  - Multiple lines of small text, varied font sizes, complex hierarchy: *Hard*.
- **Texture cases**:
  - Uniform matte/gloss: *Easy*.
  - Discernible but sparse texture (e.g., coarse weave with large spacing): *Medium*.
  - Fine, dense, non-uniform texture (e.g., fine weave, leather grain, wood grain overprinted with graphics): *Hard*.
- **Shape cases**:
  - Smooth ceramic mug (no pattern): *Easy*.
  - Measuring cup with a few scales/digits: *Medium*.
  - Book cover with fine calligraphy plus intricate patterns/material grain: *Hard*.

**Operational Protocol and Output Format** For each image, output:

1. **Difficulty verification**: choose *Easy / Medium / Hard / Uncertain*.
2. **Consistency with model label**: *Consistent / Inconsistent*.

**Quality Control and Consistency**

- Reduce borderline oscillation. If on the boundary, use majority voting over three axes: high-frequency detail density, texture uniformity, multi-scale complexity.
- If image quality is too low (blur/low resolution) to assess detail density, mark *Uncertain* and state the reason.
- Do not score based on aesthetics or unrelated factors (exposure, colorfulness), unless they impair detail discernibility.

E.4.2   ANNOTATION VERIFICATION GUIDELINES FOR PROMPT GENERATION

This guidelines defines the validation criteria for human annotators to assess whether prompts generated by a large model, based on an image subject, comply with task standards and formatting requirements. Please verify each item strictly and record any non-compliance with brief explanations in the annotation tool.

**1. Overall Objective**   Annotators should determine whether the model-generated prompts, given the clearly defined subject `subject_name` (provided by the task input or uniquely identifiable from the image), meet the following:

1. Content centers on the subject and matches the semantic requirements of the corresponding category (background change, variation in subject viewpoint or size, interaction with other entities, attribute changes, style changes, imagination).
2. Quantity and structure meet the required counts.
3. Language is clear, unambiguous, and executable as image-generation prompts.
4. No prohibited or unsafe content (e.g., safety, ethics, copyright issues). Mark as non-compliant if present.

**2. Input and Output Specifications (Strict)**

- Input: an image (with a subject) and the model's JSON output.
- The model output must be valid JSON with the following six top-level fields (names must match exactly):

- – "background change": list with 2 items
- – "variation in subject viewpoint or size": list with 2 items
- – "interaction with other entities": list with 2 items
- – "attribute changes": list with 2 items
- – "style changes": list with 2 items
- – "imagination": list with 2 items

- Each list element is an natural-language prompt (preferably a complete sentence) and must explicitly include the subject name: subject_name.

- JSON must be machine-parseable: paired quotes, correct commas and brackets, and no comments or trailing commas.

**3. Category Definitions and Decision Criteria**   Read each prompt and decide whether it satisfies the semantic requirements of its assigned category. If not, label it "Non-compliant – Category Error."

(1) BACKGROUND CHANGE (2 ITEMS)

- Requirement: Only change the background environment or setting; the subject's default attributes remain unchanged (color/material/shape/appearance unchanged). No interactions or complex effects.

- Acceptable: Simple location/time/weather descriptions (e.g., in a park," by the sea," "at sunset").

- Prohibited: Large viewpoint changes (e.g., bird's-eye view, low-angle close-up), strongly stylized lighting techniques, obstruction/reflection/motion effects, interactions, or attribute changes (e.g., color/material swaps).

(2) VARIATION IN SUBJECT VIEWPOINT OR SIZE (2 ITEMS)

- Requirement: Moderate adjustments to subject location, perspective, and lighting are allowed; add non-interactive environmental elements (buildings, roads, trees, etc.).

- Must include more than lighting changes: at least one clear change in perspective/composition/scene elements.

- Prohibited: Interactions with other animals/people; obstruction, reflection, motion effects; changes to inherent subject attributes.

(3) INTERACTION WITH OTHER ENTITIES (2 ITEMS)

- Requirement: Include one or more of the following:
  - Interactions between the subject and other animals/people or multi-subject interactions.
  - Visual effects such as obstruction, reflection, motion blur/trails.
- May combine perspective, lighting, and environmental changes.

- Prohibited: Changing the subject's inherent attributes (color/shape/material/appearance); if attributes change, it belongs in "attribute changes."

(4) ATTRIBUTE CHANGES (2 ITEMS)

- Requirement: Change the subject's own attributes (color, shape, material, components' appearance, patterns, structural details).

- Changes should focus on the subject itself, not merely environment or lighting.

- Prohibited: Purely stylistic changes (e.g., in the style of Impressionism") belong in style changes"; scene/interaction-only changes belong in background change/variation in subject viewpoint or size/interaction with other entities.

(5) STYLE CHANGES (2 ITEMS)

- Requirement: Change the overall artistic style (art movements, media, periods, processes, rendering methods).

- Example dimensions: Impressionism, Cubism, Baroque, Minimalism, cyberpunk, watercolor, oil painting, pixel art, low-poly, woodcut/printmaking, cel shading, photorealism, etc.

- Prohibited: Only changing inherent subject attributes (belongs in attribute changes"); only changing perspective/interaction/obstruction (belongs in other categories).

- Copyright and sensitivity: Avoid using identifiable living/modern specific artist names or protected signature styles. If present, mark Non-compliant – Copyright/Style."

(6) IMAGINATION (2 ITEMS)

- Requirement: Construct unrealistic or fantastical scenarios (science fiction, fantasy, surrealism), allowing violations of physical laws or common sense.

- Must still center on the subject and include the subject name.

- Prohibited: Vague fantasies unrelated to the subject; avoid illegal, harmful, or clearly unsafe content.

**4. Common Compliance Requirements**

SUBJECT CONSISTENCY

- Every prompt must contain the literal string `subject_name` (e.g., the subject is {subject_name}" or subject_name standing..."). If missing or misspelled, mark Non-compliant – Subject Missing/Error."

- Content must revolve around the subject, with a clear role and visibility. If the subject is ignored or replaced, mark Non-compliant – Subject Irrelevant."

SEMANTIC CLARITY AND NON-CONTRADICTION

- No contradictory descriptions (e.g., midnight with strong sunlight," underwater and on a desert" simultaneously).

- Do not include elements exceeding the category scope (e.g., complex interactions/effects in background change/variation in subject viewpoint or size; attribute/style changes in background change/variation in subject viewpoint or size/interaction with other entities).

- In attribute changes," do not let style terms be the primary change; conversely, in style changes," avoid attribute edits as the main point.

DIVERSITY AND NON-REPETITION

- Prompts within the same field should have substantive differences (location/viewpoint/elements/interaction types/attribute or style dimensions).

- Avoid mere synonym swaps or repeating the same composition.

SAFETY AND ETHICS

- No violent harm, hate, explicit adult content, illegal activity, or instructions for dangerous behavior.

- For people, avoid identifiable personal or sensitive information.

- Avoid generating protected brand logos or named-artist signature styles. Prefer generic style descriptors.

### E.4.3 ANNOTATION GUIDELINES FOR LABELING OF THE SICS TRAINING DATASETS

For detailed annotation guidelines, please refer to the Appendix E.2.

### E.4.4 ANNOTATOR RECRUITMENT, TRAINING, AND CONSENSUS PROTOCOL

We provide the following elaboration on annotator requirements and the specifics of our annotation pipeline:

1. All annotators involved in this work possess extensive experience in domain-relevant annotation. They have previously participated in similar tasks and have a thorough understanding of subject-driven text-to-image (T2I) generation.

2. All annotators hold a bachelor's degree or higher and are between 25 and 30 years old, ensuring the capability to accurately comprehend and implement the annotation guidelines.

3. We provided human annotators with sufficient training to ensure full understanding of the subject-driven T2I task and to promote unbiased, accurate evaluations. Detailed annotation guidelines are provided (see Figure 5 and Figure 12), and we conducted meetings with all annotators to comprehensively align and explain the rules.

4. To ensure consensus on annotation standards, we conducted a calibration process. Annotators first labeled a small pilot dataset; we then reviewed discrepant cases and provided targeted feedback and additional training. This iterative process was repeated until all annotators demonstrated high consistency in their understanding of the differentiation criteria for each annotation task.

5. During the formal annotation phase, each sample was independently labeled by five annotators. To construct a high-confidence training dataset, we applied a consensus-based filtering criterion, retaining only samples for which at least four annotators (i.e., $\geq 80\%$) assigned identical labels.

## F MORE GENERATION EXAMPLES

Figure 13 shows the generation examples of different methods across different difficulty levels. Figure 14 shows the generation examples of different methods across different prompt scenarios. The blue block highlights encoder-based methods, and the green block highlights fine-tuning-based methods.

## G LARGE LANGUAGE MODEL USAGE STATEMENT

In this work, we leverage large language models to assist human annotators during data labeling and to generate prompts. Detailed procedures are provided in the Appendix E.3, Appendix E.4, Section 2.1.1 and Section 2.1.3. We also employed large language models to aid in manuscript preparation.

## H LIMITATIONS

DSH-Bench addresses the limitations of current subject-driven T2I generation benchmarks by providing a comprehensive and diverse dataset with 459 subject images and 5,508 prompts, covering categories such as person, mammal, clothing, and so on. However, the benchmark is constrained to 459 subject images. Increasing the number of test samples could enhance the credibility and complexity of the evaluation. Additionally, we did not conduct a cross-analysis between subject difficulty and prompt scenario. Despite meticulous manual reviews, some unintentional annotation errors may still be present.

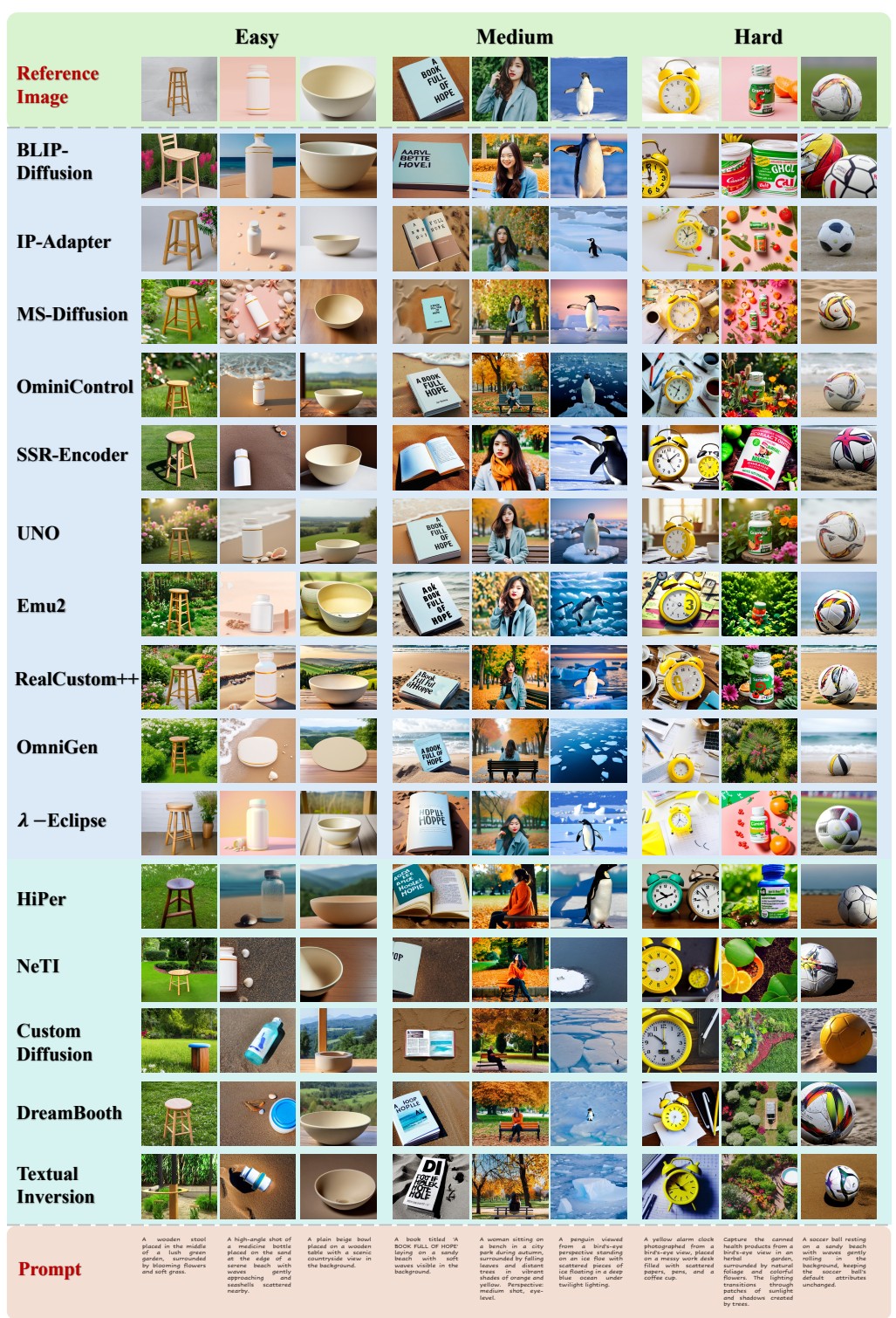

Figure 13: **Examples of images generated by all methods on different subjects difficulty level.**

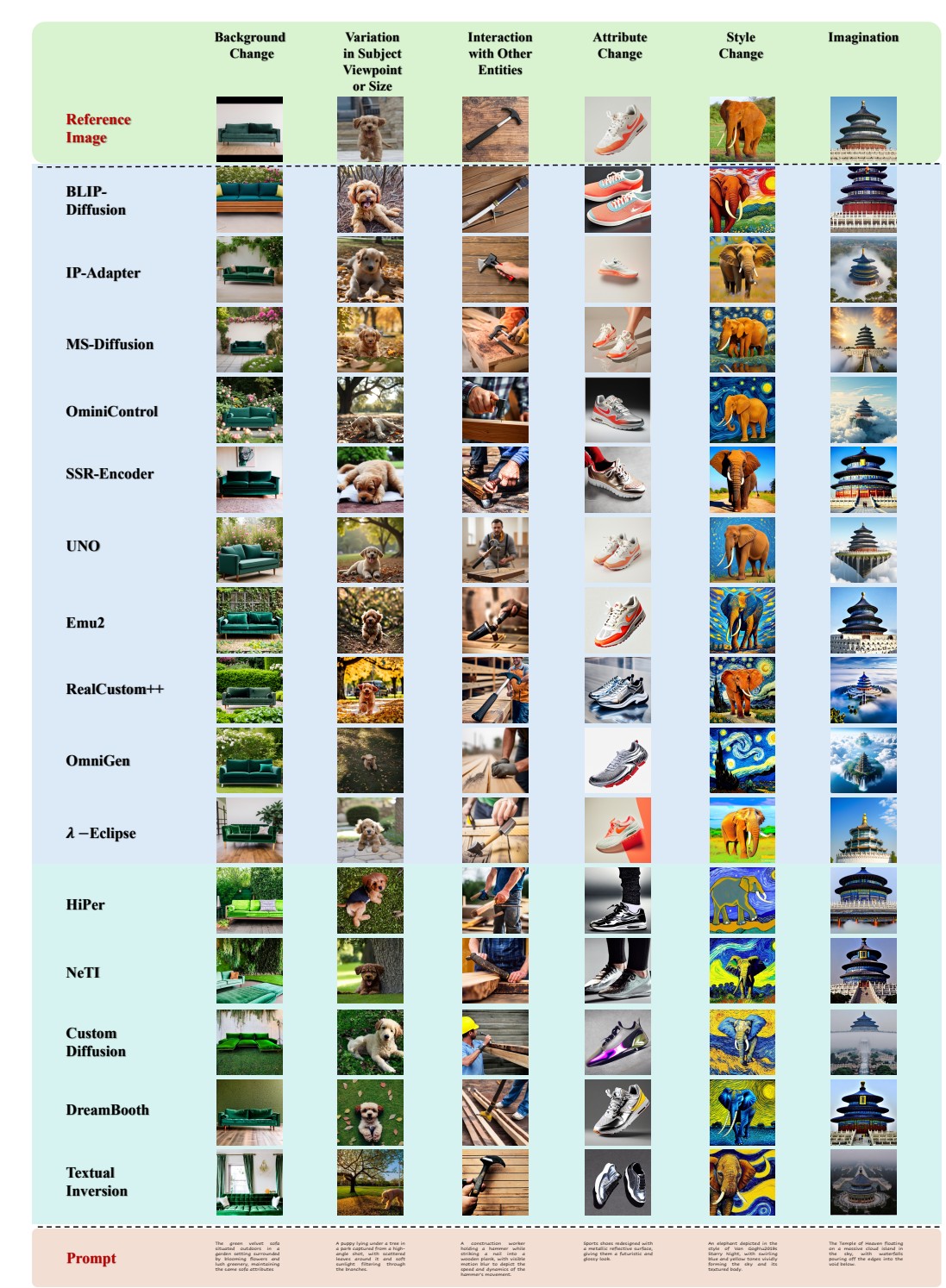

Figure 14: **Examples of images generated by all methods on different prompt scenarios.**

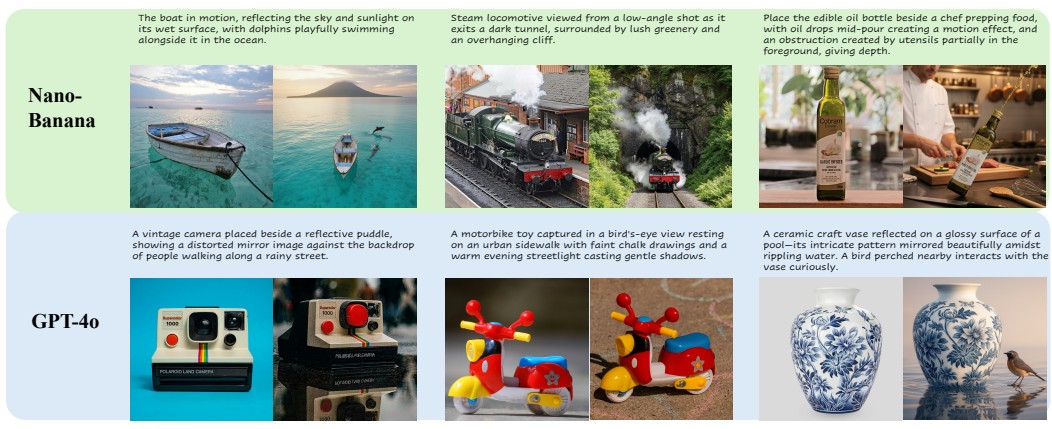

Figure 15: **Examples of badcases generated by GPT-4o and Nano-Banana on hard examples**

