# OpenReview forum: "DSH-Bench: A Difficulty- and Scenario-Aware Benchmark with Hierarchical Subject Taxonomy for Subject-Driven Text-to-Image Generation"
_ICLR.cc/2026/Conference — Submitted to ICLR 2026_

### Official Review · Reviewer_72ca · 2025-10-31

[review text omitted: it was posted to a different submission]

---

> ### Author Response · Authors · 2025-11-24
> **Review Mismatch for Our Paper**
>
> # Response
> Thanks for your valuable time and effort in reviewing our paper.
>
> After carefully reading your comments, we have strong reasons to believe that the comments were intended for a different paper. Could you please verify if the correct review was attached to our submission? I would appreciate it if you could look into this matter and kindly provide us with the correct comments.
>
> Thank you for your time and assistance.
>
> Sincerely,
>
> The Authors

---

> > ### Comment · Area_Chair_oXF2 · 2025-11-24
> >
> > Dear authors,
> >
> > After reading this review, I think reviewer 72ca made a terrible mistake on this paper. This paper has nothing to do with videos and speech, while the reviewer's comments are focusing on these topics. As the area chair, my decision is that this review will not be considered a valid review for this paper, and the authors can ignore this review. I will contact the reviewer for more details.
> >
> > AC

---

### Official Review · Reviewer_SKZe · 2025-10-31

**Soundness:** 3
**Presentation:** 3
**Contribution:** 3
**Rating:** 4
**Confidence:** 4

**Summary:**

This paper introduces DSH-Bench, a comprehensive benchmark designed to evaluate subject-driven text-to-image (T2I) generation models. The authors identify two critical limitations in existing benchmarks like DreamBench and DreamBench++: (1) a lack of diversity in subject images, leading to potential evaluation bias, and (2) insufficient granularity in assessing model performance across different levels of difficulty and types of prompts.

To address these gaps, DSH-Bench uses a hierarchical taxonomy derived from datasets like COCO and ImageNet to sample a diverse set of 459 subject images across 58 categories] It also introduces a novel classification scheme that categorizes subjects by difficulty (Easy, Medium, Hard) and prompts by scenario (e.g., background change, style change, imagination). This allows for a more fine-grained analysis of model capabilities. Finally, the paper proposes a new metric, the Subject Identity Consistency Score (SICS), which is designed to be more aligned with human perception of subject preservation and more cost-effective than evaluations that rely on expensive API calls to large models like GPT-4o.

**Strengths:**

1. The paper presents a valuable benchmark for the community focusing specifically on the issue of subject consistency in generated images. The authors create a principled taxonomy regarding subject difficulty and prompt difficulty and conduct an analysis
2. The authors propose a metric for automatically measuring subject consistency between images. They measure the correlation of their metric and other approaches with human ratings and show significantly stronger correlation.

**Weaknesses:**

1. The classification of subject difficulty is based on criteria that, while thoughtfully defined, are inherently qualitative (e.g., "minimal surface complexity" vs. "non-uniform texture distributions"). While GPT-4o and human annotators were used to ensure consistency, this process is still subjective and may not capture all nuances of what makes a subject difficult for a T2I model to render. Also the difficulty of a subject for a T2I model depends on the model itself and its training data so it is not something consistent across models.
2. The authors did not benchmark the latest state-of-the-art image generation models on their dataset, e.g., GPT, Gemini. I think this is crucial in order to understand how difficult/useful the benchmark is now and to validate whether their observations hold for more powerful models.
3. The constraint of having a single subject with a clean background on the images decreases the usefulness of the benchmark. Since the authors try to create different categories of different difficulty for evaluation, extending to multiple subjects and/or noisy backgrounds with other objects etc would be a more reasonable way to increase the difficulty and make sure that the benchmark will remain challenging as the image generation models are improved.

**Questions:**

1. Why did you constraint the data to not include multiple subjects or noisier environments in the images? I think given the state-of-the-art currently, this would make the benchmark more challenging and more useful for evaluation.

---

> ### Author Response · Authors · 2025-11-23
>
> # Response
> I'm very appreciative of your thorough review of my paper and the numerous suggestions you've provided! I will now clarify each of the concerns you raised regarding the points of confusion in the paper.
>
> &nbsp;
>
> ## **W1: Clarification of Explicit Criteria for Subject Difficulty Classification**
> >  We sincerely appreciate your thorough review of our paper and your valuable feedback. Regarding the reviewer's concern that our classification criteria may be somewhat subjective, we respectfully argue that this characterization is not entirely accurate.
> > 1. **Empirical Derivation of Difficulty Levels:**
>  Our proposed classification criteria are derived from an extensive analysis of generation results across numerous models. We identified that all models consistently perform poorly on specific, universal attributes, such as "minimum surface complexity" and "non-uniform texture distribution." Consequently, the three difficulty levels (Simple/Medium/Hard) are not arbitrary but represent empirical patterns synthesized from the observation of a vast number of samples. Furthermore, in our final experimental analysis, the performance of all models across different tags aligns perfectly with our definitions. This consistency confirms that our subject difficulty classification is both reasonable and effective, serving as a valid metric to assess the true capabilities of a model.
> 2. **Universality Across Different Models:**
>  Addressing the reviewer's comment that "what is difficult or simple may vary across different models," we acknowledge that while model architectures and training data introduce variance, there are significant commonalities among them.
> ● Consistent Failure Modes: Through observing mass generation results from various models using different reference images, we found a universal struggle in preserving fidelity for images with intricate details. This is an empirically observed regularity.
> ● High Sensitivity to Detail: We attribute this phenomenon to two factors: first, the inherent complexity of detailed images means that even minor customization errors lead to noticeable inconsistencies (imposing higher demands on model capability); and second, current state-of-the-art models generally lack the capacity to preserve fine-grained details perfectly.
> ● **Experimental Validation:** Finally, our experiments conclusively demonstrate that every evaluated model performed worse on the "Hard" category compared to the "Simple" category. This empirical evidence fully validates the correctness and universality of our classification criteria, proving that they are robust to model variations.
>
> &nbsp;
>
> ## **W2: Regarding Closed-Source Models (Nano-banana & GPT-4o)**
> > We sincerely appreciate your thorough review of our paper and your valuable feedback.
> > Regarding models like Nona-Banana and GPT-4o, we initially excluded them because they are closed-source, and our paper focuses on open-source models to ensure reproducibility. Additionally, the API costs for a full-scale evaluation are high. However, we acknowledge the reviewer's valid concern regarding whether our benchmark remains challenging for these powerful models. To address this, we conducted a sampled evaluation by randomly selecting 30 cases for each difficulty level. The results are reported in t Table 1 and Table 2. We also provide visualizations of "failure cases" on "Hard" samples in the Appendix (see Figure 15).
>
> > **Conclusion:** The results demonstrate that while these SOTA models perform well on simple samples, they still struggle significantly with preserving fine details in the "Hard" category samples. This confirms that our benchmark remains a challenging and valuable standard for guiding future improvements in subject-driven text-to-image generation, even for the most advanced models currently available.
>
> &nbsp;
> &nbsp;
>
> Table1. DSH-Bench leaderboard.
> | Method | Subject Preservation | Prompt Following | Image Quality | $ S_h $ |
> | :--- | :--- | :--- | :--- | :--- |
> | Nano-Banana | **0.443** | **0.341** | **0.308** | **0.272** |
> | GPT-4o | 0.419 | 0.329 | 0.295 | 0.260 |
>
> &nbsp;
> &nbsp;
>
> Table2. We evaluated the performance of  Nano-Banana and GPT-4o on DSH-Bench dataset, specifically analyzing their effectiveness across images with different difficulty levels.
> | Method | Subject Preservation(Easy) | Subject Preservation(Medium) | Subject Preservation(Hard) | Prompt Following(Easy) | Prompt Following(Medium) | Prompt Following(Hard) | Image Quality(Easy) | Image Quality(Medium) | Image Quality(Hard) |
> | :--- | :--- | :--- | :--- | :--- | :--- | :--- | :--- | :--- | :--- |
> | Nano-Banana | 0.487 | 0.449 | 0.399 | 0.341 | 0.349 | 0.333 | 0.279 | 0.321 | 0.325 |
> | GPT-4o | 0.479 | 0.421 | 0.386 | 0.339 | 0.329 | 0.321 | 0.27 | 0.304 | 0.312 |

---

> ### Author Response · Authors · 2025-11-23
>
> # Response
> I'm very appreciative of your thorough review of my paper and the numerous suggestions you've provided! I will now clarify each of the concerns you raised regarding the points of confusion in the paper.
>
> ## **W3 & Q1: Not include multiple subjects or noisier environments**
> > **A:**  We sincerely appreciate the valuable suggestions provided by the reviewer.
> > 1. Focus on Single-Subject Customization:
>  In this paper, we primarily focus on the domain of single-subject-driven text-to-image generation, consistent with numerous foundational studies [1,2,3,4,5]. Single-subject generation serves as the cornerstone for multi-subject tasks. Effective evaluation in this fundamental setting is a prerequisite for mastering more complex scenarios. Therefore, our current work concentrates solely on single-subject scenarios to provide a rigorous and in-depth analysis.
> 2. Rationale for Clean Backgrounds and Single Subjects:
> >> ● We deliberately constrained the data to single subjects with clean backgrounds for several critical reasons: **Decoupling Extraction from Preservation:** Introducing multiple subjects or noisy backgrounds shifts the evaluation focus toward the model's ability to extract or segment a specific subject from a cluttered scene. In contrast, our benchmark aims to rigorously evaluate the model's ability to preserve subject details. We believe that isolating the "preservation" capability is crucial for a precise evaluation. **Real-world User Behavior**: From an application perspective, users typically provide reference images where the subject is clear, distinct, and centrally positioned to achieve the best customization results. Our benchmark reflects this common usage pattern.
>   &nbsp; ● Existing Challenges in SOTA Models: Crucially, high-fidelity detail preservation remains an unsolved challenge, even with clean backgrounds. Current state-of-the-art (SOTA) models still fail to achieve perfect consistency across all categories and difficulty levels. For instance, consider a hardcover book featuring intricate English calligraphy and fine floral patterns. When customizing based on such an image, even SOTA models struggle to perfectly reconstruct these specific patterns. This limitation hinders real-world adoption—such as in advertising, where maintaining strict product consistency is non-negotiable. Therefore, benchmarking this "detail preservation" capability is highly significant before moving to noisier environments.
>
> > 3. Generalizability and Future Work:
>  Nevertheless, our proposed methodologies—such as the subject difficulty classification and prompt scenario categorization—are highly transferable to multi-subject settings. While our metric, SICS, was trained specifically for single-subject customization, the underlying approach offers a blueprint for evaluating multi-subject tasks. We agree that the reviewer’s suggestion represents an important direction. We are committed to exploring multi-subject and noisy-environment evaluations in future work and will explicitly discuss this scope and limitation in the final version of the paper.
>
> # References
> [1] Yuang Peng, Yuxin Cui, Haomiao Tang, Zekun Qi, Runpei Dong, Jing Bai, Chunrui Han, Zheng Ge, Xiangyu Zhang, and Shu-Tao Xia. Dreambench++: A human-aligned benchmark for personalized image generation. In The Thirteenth International Conference on Learning Representations, 2025.
>
> [2] Rinon Gal, Yuval Alaluf, Yuval Atzmon, Or Patashnik, Amit Haim Bermano, Gal Chechik, and Daniel Cohen-or. An image is worth one word: Personalizing text-to-image generation using textual inversion. In The Eleventh International Conference on Learning Representations, 2023.
>
> [3] Nataniel Ruiz, Yuanzhen Li, Varun Jampani, Yael Pritch, Michael Rubinstein, and Kfir Aberman. Dreambooth: Fine tuning text-to-image diffusion models for subject-driven generation. In Proceedings of the IEEE/CVF conference on computer vision and pattern recognition, pages 22500–22510, 2023.
>
> [4] Hu Ye, Jun Zhang, Sibo Liu, Xiao Han, and Wei Yang. Ip-adapter: Text compatible image prompt adapter for text-to-image diffusion models. arXiv preprint arXiv:2308.06721, 2023.
>
> [5] Zhenxiong Tan, Songhua Liu, Xingyi Yang, Qiaochu Xue, and Xinchao Wang. Ominicontrol: Minimal and universal control for diffusion transformer. arXiv preprint arXiv:2411.15098, 2024.

---

> ### Author Response · Authors · 2025-11-28
> **Sincere Request for Further Discussions**
>
> Dear Reviewer,
>
> We sincerely appreciate the time and effort you have devoted to reviewing our work. Your detailed comments and constructive suggestions have been invaluable in improving the clarity and completeness of our work.
>
> In response to your concerns, we have provided detailed replies and conducted additional experiments to address the issues you raised. All these updates will be incorporated into the final version of the paper. We hope that our responses satisfactorily address your concerns. If there are any remaining concerns or aspects that require clarification, we are ready to address them as soon as possible.
>
> If you find that our responses have resolved your concerns, we would be grateful if you would consider raising your final rating to a higher score. Your feedback is crucial to the improvement of our work and is greatly appreciated.
>
> Thank you once again for your thoughtful review and support.
>
> Best regards,
>
> Authors

---

### Official Review · Reviewer_Szmx · 2025-10-31

**Soundness:** 4
**Presentation:** 2
**Contribution:** 4
**Rating:** 6
**Confidence:** 4

**Summary:**

This paper introduces DSH-Bench, a new benchmark for evaluating subject-driven text-to-image (T2I) generation models. The authors argue that existing benchmarks suffer from insufficient diversity in subject images and lack granular analysis of subject difficulty and prompt scenarios. DSH-Bench addresses these by providing a diverse dataset of 459 subject images across 58 categories.

Subject Identity Consistency Score (SICS), a new metric for subject consistency + prompt following is proposed, which is shown to have a higher correlation with human judgment than existing metrics like DreamBench++.

**Strengths:**

1. DSH-Bench significantly expands subject diversity compared to previous benchmarks like DreamBench, with 459 subjects across 58 fine-grained categories derived from a hierarchical taxonomy.

2. The paper evaluates a large number (15) of diverse T2I models (both fine-tuning and encoder-based), providing a valuable snapshot of the current state of the field

3. Authors shared their DSH-Bench benchmark with the paper and they attend to publish it to the community.

4.

**Weaknesses:**

1. critical weakness is that the SICS metric is only evaluated on the authors' own dataset. To prove it is a generally applicable metric and not overfitted to DSH-Bench, it needs to be evaluated against other human-annotated Text-to-Image datasets.

2. The classification of subject difficulty into Easy/Medium/Hard, while guided by GPT-4o and human review, inherently contains some subjectivity.

3. While larger than DreamBench, 459 images is still relatively small for a "large-scale" benchmark in the era of generative AI.

4. Many figures and plots (e.g., Figure 4, 6, and 7) are extremely small in the main paper, requiring significant zooming to be legible. I strongly recommend increasing their size in the main text where feasible, leave the main point of the figure and move the remaining to the Appendix in larger size.

**Questions:**

1. Will the fine-tuned weights for the SICS model be publicly shared? The paper mentions open-sourcing the benchmark and "related code," but explicitly confirming the release of the SICS model itself is important as it is a primary contribution

2. Could you provide more details on the inter-annotator agreement for the subject difficulty classification? How often did annotators disagree with GPT-4o's initial classification?

3. Have you considered expanding the "interaction with other entities" prompt scenario to include more complex interactions beyond just co-occurrence, such as the subject actively doing something to or with another entity?

4. How does SICS perform on out-of-distribution subject categories that are not well-represented in its training data?

---

> ### Author Response · Authors · 2025-11-23
> **Response Regarding the Weaknesses**
>
> # Response
>
> I'm very appreciative of your thorough review of our paper and the numerous suggestions you've provided! I will now clarify each of the concerns you raised regarding the points of confusion in the paper.
>
> ## **W1: Generalization Analysis of the SICS Metric on External Datasets**
> > **A1:** During the training phase, we constructed the dataset by carefully selecting data that spans a wide range of scenarios, difficulty levels, and categories, thereby ensuring a diverse data distribution. To further enrich the dataset, we also incorporated additional general images, such as commercial product advertisements, for training data generation.
>
> > To further address the reviewers' concerns regarding generalization, we conducted an evaluation on DreamBench, which serves as an out-of-distribution (OOD) test set. We randomly sampled 100 instances generated by three representative models: UNO, BLIP-Diffusion, and IP-Adapter. The evaluation metrics remain consistent with those detailed in the main paper. The experimental results are presented in Table 1. It can be observed that our method demonstrates superior alignment with human evaluations. This performance is primarily attributed to our model's meticulous design and robust training strategy.
>
> **Table 1 : Results of evaluation on a OOD test set**
>
> | Model | Kendallτ-GPT-4o | Kendallτ-SICS(Ours) | Spearmanρ-GPT-4o | Spearmanρ-SICS(Ours) |
> | :--- | :---: | :---: | :---: | :---: |
> | UNO | 0.287 | 0.337 | 0.315 | 0.378 |
> | BLIP-Diffusion | 0.323 | 0.505 | 0.346 | 0.527 |
> | IP-Adapter | 0.428 | 0.434 | 0.465 | 0.451 |
> | Average | 0.346 | 0.425 | 0.375 | 0.452 |
>
> &nbsp;
>
> ## **W2: Clarification of Explicit Criteria for Subject Difficulty Classification**
> > **A2:** We express our gratitude for your meticulous examination of our paper and for providing valuable feedback. However, we respectfully disagree with the concern that our classification criteria are subjective. We would like to clarify that our proposed taxonomy is empirically derived rather than arbitrarily defined (lines 227-230).
>
> > Through extensive observation of generation results across numerous models, we identified consistent performance degradation on specific geometric and textural features—specifically, "minimal surface complexity" and "non-uniform texture distribution." Consequently, the categorization into Easy/Medium/Hard levels represents empirical patterns distilled from analyzing a large-scale sample set.
>
> > Furthermore, our experimental results (See Analysis, lines 427-430) corroborate this taxonomy: the performance of all evaluated models consistently aligns with our difficulty definitions (i.e., performance drops as the defined difficulty increases). This demonstrates that our classification of subject difficulty is both rational and effective, serving as a robust metric for evaluating the true capabilities of generative models.
>
> &nbsp;
>
> ## **W3: While larger than DreamBench, 459 images is still relatively small**
> > **A3:** We thank the reviewer for their thorough review and valuable feedback. We firmly believe that the concerns raised do not compromise the effectiveness of our benchmark in evaluating model capabilities.
>
> > **Representativeness over Quantity:**  For customized text-to-image generation, we argue that representativeness is paramount. Our taxonomy is meticulously constructed to be comprehensive, covering the vast majority of common objects. By retrieving reference images based on this detailed hierarchy, we ensure that the dataset is highly representative of real-world scenarios.
>
> > **Extensive Scale and Robustness:** We have designed diverse prompts across different scenarios. With 12 distinct prompts per reference subject, the benchmark comprises a total of 5,508 prompts. This means a model must generate at least 5,508 samples per evaluation.
>
> > **Statistical Reliability:**  Furthermore, as detailed in Appendix E.1, we generate 4 random samples for each prompt and report the averaged metrics to minimize variance. This results in a total of 22,032 generated images per model evaluation. This substantial volume of data guarantees the robustness and statistical significance of our results.
>
> > In summary, we are confident that our benchmark provides a comprehensive and accurate assessment of model performance.
>
> &nbsp;
>
> ## **W4: Many figures and plots are extremely small in the main paper**
> > **A4:** We sincerely appreciate your valuable suggestion. Due to the current page limit constraints, we acknowledge that some key figures are displayed at a smaller scale than ideal. We will strive to present the results more clearly in the final (camera-ready) version where space permits.

---

> ### Author Response · Authors · 2025-11-23
> **Response Regarding the Questions**
>
> # Response
> I'm very appreciative of your thorough review of my paper and the numerous suggestions you've provided! I will now clarify each of the concerns you raised regarding the points of confusion in the paper.
>
> &nbsp;
>
> ## **Q1: Open-Sourcing the SICS Model and Fine-tuned Weights**
> > **A1:** As explicitly stated in the manuscript (Lines 160-161), we plan to open-source all relevant data, code, and evaluation results. We would like to take this opportunity to reassure you that all related resources (including SICS model and fine-tuned weights) will be made publicly available to facilitate reproducibility.
>
> &nbsp;
>
> ## **Q2: More details on the inter-annotator agreement for the subject difficulty classification**
> > **A2:** We sincerely appreciate your thorough review of our paper and your valuable feedback.
>
> > Regarding the "Inter-Annotator Agreement (IAA) for topic difficulty classification" you mentioned, we have provided detailed documentation in Appendix E.4.4 (Lines 1750-1754). The Inter-Annotator Agreement exceeds 80%.
>
> > Regarding the "frequency of disagreement between annotators and GPT-4o's initial classification," we observed a high level of alignment between human annotators and GPT-4o, particularly for the Easy and Hard categories. Specifically:
>
> > ● For images labeled as Easy by GPT-4o, 96.7% of the labels were consistent with human annotations.
>
> > ● For images labeled as Hard by GPT-4o, 92.4% of the labels were consistent with human annotations.
>
> > ● For images labeled as Medium by GPT-4o, 82.6% of the labels were consistent with human annotations.
>
> &nbsp;
>
> ## **Q3: Expanding the "interaction with other entities" prompt scenario**
> > **A3:** We are very appreciative of your valuable insights. We would like to clarify that our current benchmark dataset already incorporates instances of "more complex interactions beyond just co-occurrence" as suggested.
>
> > For example, consider the following prompt from our dataset: "The shoes worn by a child interacting with a playful golden retriever in a backyard, with a ball rolling between them." In this specific case, the subject to be customized is a pair of shoes. The scenario involves a child actively wearing these shoes while engaging in play, which exemplifies a complex interaction rather than simple static co-occurrence. We have provided a subset of images and prompts from our dataset in the Supplementary Material.
>
> &nbsp;
>
> ## **Q4: SICS's performance on out-of-distribution subject categories**
> > **A4:** We express our gratitude for your meticulous examination of our paper and for providing valuable feedback. We address the specific concerns raised regarding the model generalization from the following three perspectives:
>
> > 1. **Training Set Construction and Balance:**
> When constructing the training set, we utilized various open-source models to generate image pairs based on reference images from our benchmark, followed by manual annotation according to strict standards. We ensured a balanced coverage of diverse categories during this process; therefore, the training set is free from bias toward "atypical" categories. Furthermore, to enhance the generalizability of our method, we also selected external images (outside the primary dataset) as references to augment the training data.
>
> > 2. **Unbiasedness of the Test Set:**
>  To ensure the unbiasedness of our evaluation, the test set was constructed by randomly sampling image pairs generated by over a dozen different models on the benchmark dataset. This random sampling strategy guarantees the statistical reliability and confidence of our test results.
>
> > 3. **Generalization Mechanism of SICS:**
>  We posit that the training methodology of the SICS model inherently guarantees sufficient generalizability. During training, we explicitly emphasized via prompts that the model should prioritize the similarity of the subjects within the two input images (lines 1266 - 1272). Consequently, when encountering unseen data, the model is conditioned to assess similarity based solely on the visual appearance of the subjects, rather than relying on high-level semantics. This distinct focus on subject identity over general semantics constitutes a key advantage of our metric compared to CLIP-I.

---

> ### Author Response · Authors · 2025-11-28
> **Sincere Request for Further Discussions**
>
> Dear Reviewer,
>
> Thank you for your invaluable efforts and constructive feedback on our manuscript.
>
> As the discussion period draws to a close, we eagerly anticipate your thoughts on our response. We sincerely hope that our response meets your expectations. If there are any remaining concerns or aspects that require clarification, we are ready to address them as soon as possible.
>
> Best regards,
>
> The Authors

---

### Official Review · Reviewer_CnDT · 2025-11-01

**Soundness:** 3
**Presentation:** 3
**Contribution:** 2
**Rating:** 4
**Confidence:** 4

**Summary:**

DSH-Bench is a comprehensive benchmark for subject-driven text-to-image (T2I) generation. It addresses the limitations of existing benchmarks through three core innovations: first, it adopts a hierarchical taxonomy sampling mechanism, covering 58 fine-grained categories and 459 subject images. Compared with DreamBench (6 categories, 30 subjects) and DreamBench++ (150 subjects), its diversity is significantly improved (the number of categories is increased by 8 times and the number of subjects by 15 times); second, it proposes an classification scheme involving subject difficulty levels (Easy, Medium, Hard) and prompt scenarios (6 types including background change, viewpoint/size variation, etc.), enabling fine-grained evaluation of model capabilities; third, it introduces the Subject Identity Consistency Score (SICS) metric, which has a 9.4% higher correlation with human evaluation than existing metrics and is far less costly than the GPT-4o-based evaluation in DreamBench++. Through empirical evaluation of 15 subject-driven T2I models, DSH-Bench reveals the limitations of current models in preserving complex subject details and adapting to multiple prompt scenarios.

**Strengths:**

1. The experiments are thorough and involve significant workload, which provides reference value for the evaluation of subject-driven image generation tasks.
2. This benchmark achieves better alignment with human evaluation results.

**Weaknesses:**

1. The methods used for comparison are not up-to-date. For example, some of the latest models like nano-banana, GPT-4o, and DreamO are not included in the evaluation.
2. The contributions of this paper do not meet the acceptance criteria of ICLR. While I acknowledge the substantial workload invested in this research—primarily reflected in making the benchmark more detailed, systematic, and cost-effective—its contribution to the evaluation of subject-driven text-to-image generation remains limited.

**Questions:**

Please refer to the weakness part

---

> ### Author Response · Authors · 2025-11-22
> **Response Regarding the Comparison with Latest Methods**
>
> I'm very appreciative of your thorough review of my paper and the numerous suggestions you've provided! I will now clarify each of the concerns you raised regarding the points of confusion in the paper.
> # 1. Regarding DreamO (Concurrent Work):
> > DreamO is a concurrent work released around the same time as our submission, which is why it was not included in the initial evaluation. However, our original paper already includes a comprehensive analysis of 15 representative open-source models (e.g., UNO), which we believe is sufficient to substantiate the validity of our conclusions.
> To address the reviewer's concern, we have now conducted a full evaluation of DreamO. As shown in Tables 1, 2, 3, 4, 5 and 6, the additional results align perfectly with the trends and conclusions presented in our main paper.
>
> &nbsp;
> &nbsp;
>
>  Table1. Evaluation of Subject-driven T2I generation.
> | Model | Subject Preservation(BD) | Subject Preservation(DB++) | Subject Preservation(HB) | Prompt Following(BD) | Prompt Following(DB++) | Prompt Following(HB) | Image Quality(BD) | Image Quality(DB++) | Image Quality(HB) |
> | :--- | :---: | :---: | :---: | :---: | :---: | :---: | :---: | :---: | :---: |
> | **DreamO** | 0.412 | 0.396 | 0.398 | 0.324 | 0.339 | 0.326 | 0.314 | 0.308 | 0.283 |
> | **BLIP-Diffusion** | 0.229 | 0.216 | 0.204 | 0.291 | 0.278 | 0.277 | 0.267 | 0.254 | 0.223 |
> | **IP-Adapter** | 0.23 | 0.244 | 0.229 | 0.321 | 0.318 | 0.315 | 0.291 | 0.296 | 0.266 |
> | **MS-Diffusion** | 0.316 | 0.346 | 0.352 | 0.332 | 0.339 | 0.338 | 0.311 | 0.314 | 0.294 |
> | **OminiControl** | 0.279 | 0.268 | 0.258 | 0.325 | 0.337 | 0.334 | 0.312 | 0.308 | 0.29 |
> | **SSR-Encoder** | 0.231 | 0.202 | 0.202 | 0.29 | 0.287 | 0.295 | 0.273 | 0.27 | 0.247 |
> | **UNO** | 0.409 | 0.41 | 0.409 | 0.317 | 0.322 | 0.323 | 0.304 | 0.297 | 0.278 |
> | **Emu2** | 0.36 | 0.343 | 0.341 | 0.291 | 0.309 | 0.304 | 0.272 | 0.278 | 0.26 |
> | **RealCustom++** | 0.377 | 0.38 | 0.375 | 0.325 | 0.329 | 0.332 | 0.316 | 0.314 | 0.298 |
>
> &nbsp;
> &nbsp;
>
> Table2. DSH-Bench leaderboard.
> | Method | T2I Model | Subject Preservation | Prompt Following | ImageQuality | $ S_h $ |
> | :--- | :--- | :---: | :---: | :---: | :---: |
> | **UNO** | FLUX.1-dev | **0.409** | 0.323 | 0.278 | **0.252** |
> | **DreamO** | FLUX.1-dev | $ \underline{0.398}$ | 0.326 | 0.283 | **0.252** |
> | **RealCustom++** | SDXL | 0.375 | $ \underline{0.332} $ | **0.294** | $ \\underline{0.251}$ |
> ... ...
>
> &nbsp;
> &nbsp;
>
> Table3. We evaluated the performance of various methods on DSH-Bench dataset, specifically analyzing their effectiveness across images with different difficulty levels.
> | Method | Subject Preservation(Easy) | Subject Preservation(Medium) | Subject Preservation(Hard) | Prompt Following(Easy) | Prompt Following(Medium) | Prompt Following(Hard) | Image Quality(Easy) | Image Quality(Medium) | Image Quality(Hard) |
> | :--- | :---: | :---: | :---: | :---: | :---: | :---: | :---: | :---: | :---: |
> | **DreamO** | 0.473 | 0.412 | 0.375 | 0.336 | 0.321 | 0.322 | 0.265 | 0.281 | 0.291 |
> | **BLIP-Diffusion** | 0.221 | 0.209 | 0.19 | 0.284 | 0.278 | 0.273 | 0.198 | 0.227 | 0.232 |
> | **IP-Adapter** | 0.266 | 0.233 | 0.206 | 0.316 | 0.315 | 0.316 | 0.236 | 0.27 | 0.278 |
> | **MS-Diffusion** | 0.41 | 0.362 | 0.312 | 0.34 | 0.339 | 0.335 | 0.278 | 0.297 | 0.299 |
> ... ...
> | **Average** | 0.272 | 0.246 | 0.224 | 0.317 | 0.315 | 0.311 | 0.241 | 0.261 | 0.265 |
>
> &nbsp;
> &nbsp;
>
> Table 4. We evaluated the subject preservation of various methods on different prompt scenarios.
> | Method | Background Change | Variation in Subject Viewpoint or Size | Interaction with Other Entities | Attribute Change | Style Change | Imagination |
> | :--- | :---: | :---: | :---: | :---: | :---: | :---: |
> | **DreamO** | 0.421 | 0.41 | 0.357 | 0.347 | 0.409 | 0.351 |
> ... ...
> | **Average** | 0.261 | 0.240| 0.225 | 0.205 | 0.212 | 0.207 |
>
> &nbsp;
> &nbsp;
>
> Table 5: We evaluated the prompt following  of various methods on different prompt scenarios.
> | Method | Background Change | Variation in Subject Viewpoint or Size | Interaction with Other Entities | Attribute Change | Style Change | Imagination |
> | :--- | :---: | :---: | :---: | :---: | :---: | :---: |
> | **DreamO** | 0.339 | 0.321 | 0.338 | 0.319 | 0.298 | 0.343 |
> ... ...
> | **Average** | 0.322 | 0.314 | 0.314 | 0.309 | 0.308 | 0.316 |
>
> &nbsp;
> &nbsp;
>
> Table 6: We evaluated the image  quality of various methods on different prompt scenarios.
> | Method | BackgroundChange | Variation in SubjectViewpoint or Size | Interaction withOther Entities | AttributeChange | StyleChange | Imagination |
> | :--- | :---: | :---: | :---: | :---: | :---: | :---: |
> | **DreamO** | 0.276 | 0.291 | 0.294 | 0.289 | 0.261 | 0.289 |
> ... ...
> | **Average** | 0.261 | 0.257 | 0.254 | 0.260 | 0.261 | 0.259 |

---

> ### Author Response · Authors · 2025-11-22
> **Response Regarding the Comparison with Latest Methods**
>
> I'm very appreciative of your thorough review of my paper and the numerous suggestions you've provided! I will now clarify each of the concerns you raised regarding the points of confusion in the paper.
>
> # 2. Regarding Closed-Source Models (Nano-banana & GPT-4o):
>
> > Regarding models like Nona-Banana and GPT-4o, we initially excluded them because they are closed-source, and our paper focuses on open-source models to ensure reproducibility. Additionally, the API costs for a full-scale evaluation are high.
>
> > However, we acknowledge the reviewer's valid concern regarding whether our benchmark remains challenging for these powerful models. To address this, we conducted a sampled evaluation by randomly selecting 30 cases for each difficulty level. The results are reported in the newly added Table 1 and Table 2. We also provide visualizations of "failure cases" on "Hard" samples in the Appendix (see Figure 15).
>
> > Observations: The results show that these closed-source models have indeed made significant progress in customized text-to-image generation. However, while these SOTA models excel at preserving details in "Easy" samples, they still struggle significantly with samples labeled as "Hard." This confirms that our benchmark remains a challenging and valuable standard for guiding future research in this field.
>
> &nbsp;
> &nbsp;
>
> Table1. DSH-Bench leaderboard.
> | Method | Subject Preservation | Prompt Following | Image Quality | $ S_h $ |
> | :--- | :--- | :--- | :--- | :--- |
> | Nano-Banana | **0.443** | **0.341** | **0.308** | **0.272** |
> | GPT-4o | 0.419 | 0.329 | 0.295 | 0.260 |
>
> &nbsp;
> &nbsp;
>
> Table2. We evaluated the performance of  Nano-Banana and GPT-4o on DSH-Bench dataset, specifically analyzing their effectiveness across images with different difficulty levels.
> | Method | Subject Preservation(Easy) | Subject Preservation(Medium) | Subject Preservation(Hard) | Prompt Following(Easy) | Prompt Following(Medium) | Prompt Following(Hard) | Image Quality(Easy) | Image Quality(Medium) | Image Quality(Hard) |
> | :--- | :--- | :--- | :--- | :--- | :--- | :--- | :--- | :--- | :--- |
> | Nano-Banana | 0.487 | 0.449 | 0.399 | 0.341 | 0.349 | 0.333 | 0.279 | 0.321 | 0.325 |
> | GPT-4o | 0.479 | 0.421 | 0.386 | 0.339 | 0.329 | 0.321 | 0.27 | 0.304 | 0.312 |
>
> &nbsp;
> &nbsp;
>
> # 3. Conclusion:
> > The results demonstrate that while these SOTA models perform well on simple samples, they still struggle significantly with preserving fine details in the "Hard" category samples. This confirms that our benchmark remains a challenging and valuable standard for guiding future improvements in subject-driven text-to-image generation, even for the most advanced models currently available.

---

> ### Author Response · Authors · 2025-11-22
> **Addressing the "Limited Contribution" Concern**
>
> We appreciate the reviewer's recognition of the extensive effort behind our work. However, we respectfully disagree with the view that our contribution to "subject-driven text-to-image generation evaluation remains limited."
>
> # Key Insights and Guidelines for Future Research
> As the reviewer noted, our benchmark makes the evaluation process more detailed, systematic, and cost-effective. However, its value extends far beyond these aspects. Crucially, our benchmark provides insights into how models behave under varying subject complexity and prompt scenarios, which is essential for diagnosing weaknesses in current methods and guiding future research. We have elaborated on these specific insights in detail in Section 5 of the main text and in Appendix D.1:
>
> > 1. **The subject-driven T2I capability for different prompt scenarios is not robust (lines 286 - 295):** (1) In BC (Background change), VS (Variation in subject viewpoint or size), and IE (Interaction with other entities) scenarios, the model’s performance consistently declines across all evaluation dimensions. This trend suggests that the difficulty of the scenarios increases progressively from BC to IE. Notably, the finding that the IE scenario is more challenging than the BC scenario aligns with intuitive expectations. Therefore, future research may need to place greater emphasis on IE scenario. (2) For subject preservation, the model’s average performance across the AC, SC, and IM prompt scenarios remains relatively low. This could be because the generated subjects undergo partial modifications relative to the original subjects in these three scenarios.
> > 2. **Model robustness varies considerably among categories (lines 265 - 273, lines 587 - 611):** Figure 6 provides a detailed comparison of the performance of various methods across different third-level categories. For example, performance in categories "artwork" (both photorealistic and non-photorealistic) is substantially lower. None of the current models perform well across all categories. We hypothesize that this may be related to the varying complexity of the subjects within different categories. In Appendix D.1(lines 587 - 611), we conducted a comprehensive analysis of the performance of various methods across both first-level and second-level categories.The results demonstrate that, irrespective of whether the primary category is realistic or non-realistic, the scores for the subject preservation dimension are consistently lower for the human category across nearly all models. This phenomenon can be attributed to the distribution of difficulty levels within the human category (as shown in Table 8).
> > 3. **Implications for technical approaches (lines 612 - 632):** Our analysis indicates that current encoder-based methods still face challenges in accurately reconstructing subjects with high-frequency details in images. This limitation may stem from the characteristics of commonly used image encoders, such as CLIP, which tend to prioritize semantic information over fine-grained details. Consequently, the performance of these methods on tasks classified as "hard level" is suboptimal. Future research should focus on enhancing the restoration of challenging subject details.
>
> #  SICS represents a substantive research contribution.
> Beyond offering profound insights, our proposed SICS metric constitutes a significant contribution to the effective evaluation of customized text-to-image generation tasks. Specifically, SICS serves as an efficient, scalable, low-cost, and fine-grained tool for automatically evaluating subjective consistency:
> > 1. **Focus on Core Visual Attributes over High-Level Semantics:** We observe that GPT-4o's proficiency in high-level semantic understanding can lead it to overlook critical visual details. In a "backpack" case, despite clear discrepancies in the generated image's shape, color, and logo compared to the reference, GPT-4o assigned a maximum score, suggesting a generalized conceptual match. In stark contrast, SICS, aligning with human annotators, astutely identified these inconsistencies and assigned a low score, demonstrating its capacity to focus on the core visual attributes that define the subject.
> > 2. **Superior Discriminative Power and Granularity:** A key advantage of SICS is its scoring granularity, which mitigates the "score saturation" phenomenon common in GPT-4o's upper-range evaluations. For instance, in a clock case, GPT-4o assigned an identical high score of 4 to two images with varying error severity—one with a minor color deviation and another with significant structural differences. In contrast, SICS, consistent with human ratings, successfully differentiated them with scores of 3 and 2, respectively, accurately reflecting the inconsistency. It demonstrates that SICS provides a fine-grained scoring standard align with human perception, a capability that is indispensable for researchers needing to precisely evaluate subtle model improvements.

---

> ### Author Response · Authors · 2025-11-28
> **Sincere Request for Further Discussions**
>
> Dear Reviewer,
>
> We sincerely appreciate the time and effort you have devoted to reviewing our work. Your detailed comments and constructive suggestions have been invaluable in improving the clarity and completeness of our work.
>
> In response to your concerns, we have provided detailed replies and conducted additional experiments to address the issues you raised. All these updates will be incorporated into the final version of the paper. We hope that our responses satisfactorily address your concerns. If there are any remaining concerns or aspects that require clarification, we are ready to address them as soon as possible.
>
> If you find that our responses have resolved your concerns, we would be grateful if you would consider raising your final rating to a higher score. Your feedback is crucial to the improvement of our work and is greatly appreciated.
>
> Thank you once again for your thoughtful review and support.
>
> Best regards,
>
> Authors

---

### Author Response · Authors · 2025-12-02
**Response To All Reviewers**

We sincerely thank the reviewers for their time, detailed feedback, and positive evaluation of our work. We are encouraged that the reviewers unanimously recognize the value of DSH-Bench as a comprehensive and necessary contribution to the community. Their insightful comments have significantly helped us improve the robustness and clarity of our paper.

We are particularly grateful to Reviewer Szmx for rating our soundness and contribution as "excellent." They highlighted that DSH-Bench significantly expands subject diversity compared to previous benchmarks and provides a "valuable snapshot of the current state of the field" by evaluating 15 diverse T2I models. They also commended the proposed SICS metric for demonstrating a higher correlation with human judgment than existing metrics.

We thank Reviewer CnDT for acknowledging the "significant workload" and "thorough experiments" invested in this research, stating that it provides valuable reference value. They recognized our three core innovations—the hierarchical taxonomy, the fine-grained classification scheme (difficulty levels and prompt scenarios), and the SICS metric—noting that our benchmark achieves better alignment with human evaluation results.

We also appreciate Reviewer SKZe for recognizing DSH-Bench as a "valuable benchmark for the community" that addresses critical limitations in existing works. They praised our "principled taxonomy" regarding subject and prompt difficulty and the subsequent analysis. Furthermore, they highlighted the effectiveness of our SICS metric, emphasizing its "significantly stronger correlation" with human ratings compared to other approaches.

We are deeply grateful for the reviewers' recognition of the diversity of our dataset, the rigor of our taxonomy, and the effectiveness of the SICS metric. In the individual responses below, we have carefully addressed the constructive suggestions raised (such as comparisons with the latest SOTA models, generalization of SICS, and difficulty classification criteria) with detailed clarifications and additional experiments. We hope these updates satisfactorily resolve the concerns.

---

### Meta-Review · Area_Chair_1GTV · 2026-01-07

**Summary:**

This paper proposes a new benchmark for single-subject text-to-image generation with cost-friendly metrics. The reviewers were concerned about the lack of results with SOTA T2I models, the subjective choice of difficulty levels, and the lack of insights to advance research in this area.

While most concerns have been addressed, a critical limitation is the dataset size and the fact that it only considers single-subject generation with a clear background. The reviewers acknowledged the tremendous workload of this paper. However, the current version doesn’t really imply broader adoption and impact on the community.

The authors are encouraged to extend the scope to a multi-subject benchmark with more complex background and potentially increase the dataset size. The AC does not recommend accepting this paper and encourages submitting to a future conference with an improved dataset and more insights on research problems & directions.

**Reviewer Concerns:**

Concerns that were addressed by the rebuttal:
- Missing results with latest models, e.g., Nano-banana. (Reviewer CnDT, SKZe) The authors have included more results with SOTA closed-source models in the rebuttal.
- Evaluation of SICS metric on other datasets (Reviewer Szmx)
- The classification of difficulty level into easy, medium, and difficult seems subjective. (Reviewer Szmx, SKZe) The authors have clarified the rational for classification.d

Outstanding concerns:
- Limited contribution as a benchmark paper (Reviewer CnDT) This is partially addressed by more analysis of different metrics provided in the rebuttal to guide future research.
- The benchmark is limited with only 459 images though it comes with many more prompts. (Reviewer Szmx)
- The benchmark is limited to single subject with clean background. (Reviewer SKZe)

**Reviewer Scores:**

Reviewer CnDT is likely to keep rating of 4 or increase to 6.

Reviewer Szmx is likely to keep their initial rating of 6.

Reviewer SKZe is likely to keep their initial rating of 4.

Reviewer 72ca's score is not considered, as the review is intended for another paper.

---

### Decision · Program_Chairs · 2026-01-26

Reject